# Hard-Negative Sampling for Contrastive Learning: Optimal Representation Geometry and Neural- vs Dimensional-Collapse

**Ruijie Jiang**\*  *Ruijie.Jiang@tufts.edu*
*Department of Electrical and Computer Engineering*
*Tufts University, Medford, MA*

**Thuan Nguyen**\*  *nguyent11@etsu.edu*
*Department of Engineering, Engineering Technology*
*East Tennessee State University, Johnson City, TN*

**Shuchin Aeron**  *shuchin@ece.tufts.edu*
*Department of Electrical and Computer Engineering*
*Tufts University, Medford, MA*

**Prakash Ishwar**  *pi@bu.edu*
*Department of Electrical and Computer Engineering*
*Boston University, Boston, MA*

**Reviewed on OpenReview:** *https://openreview.net/forum?id=3cnpZ5SIjU*

## Abstract

For a widely-studied data model and general loss and sample-hardening functions we prove that the losses of Supervised Contrastive Learning (SCL), Hard-SCL (HSCL), and Unsupervised Contrastive Learning (UCL) are minimized by representations that exhibit Neural-Collapse (NC), i.e., the class means form an Equiangular Tight Frame (ETF) and data from the same class are mapped to the same representation. We also prove that for any representation mapping, the HSCL and Hard-UCL (HUCL) losses are lower bounded by the corresponding SCL and UCL losses. In contrast to existing literature, our theoretical results for SCL do not require class-conditional independence of augmented views and work for a general loss function class that includes the widely used InfoNCE loss function. Moreover, our proofs are simpler, compact, and transparent. Similar to existing literature, our theoretical claims also hold for the practical scenario where batching is used for optimization. We empirically demonstrate, for the first time, that Adam optimization (with batching) of HSCL and HUCL losses with random initialization and suitable hardness levels can indeed converge to the NC-geometry if we incorporate unit-ball or unit-sphere feature normalization. Without incorporating hard-negatives or feature normalization, however, the representations learned via Adam suffer from Dimensional-Collapse (DC) and fail to attain the NC-geometry. These results exemplify the role of hard-negative sampling in contrastive representation learning and we conclude with several open theoretical problems for future work. The code can be found at `https://github.com/rjiang03/HCL/tree/main`

***Keywords:*** Contrastive Learning, Hard-Negative Sampling, Neural-Collapse.

\* Authors with equal contribution.

# 1 Introduction

Contrastive representation learning (CL) methods learn a mapping that embeds data into a Euclidean space such that *similar* examples retain close proximity to each other and *dissimilar* examples are pushed apart. CL, and in particular unsupervised CL, has gained prominence in the last decade with notable success in Natural Language Processing (NLP), Computer Vision (CV), time-series, and other applications. Recent surveys Balestriero et al. (2023), Rethmeier & Augenstein (2023) and the references therein provide a comprehensive view of these applications.

The characteristics and utility of the learned representation depend on the joint distribution of similar (positive samples) and dissimilar data points (negative samples) and the downstream learning task. In this paper we are not interested in the downstream analysis, but in characterizing the geometric structure and other properties of global minima of the contrastive learning loss under a general latent data model. In this context, we focus on understanding the impact and utility of hard-negative sampling in the UCL and SCL settings. When carefully designed, hard-negative sampling improves downstream classification performance of representations learned via CL as demonstrated in Robinson et al. (2020), Jiang et al. (2023), and Long et al. (2023). While it is known that hard-negative sampling can be performed implicitly by adjusting what is referred to as the *temperature parameter* in the CL loss, in this paper we set this parameter to unity and explicitly model hard-negative sampling through a general "hardening" function that can tilt the sampling distribution to generate negative samples that are more similar to (and therefore harder to distinguish from) the positive and anchor samples. We also numerically study the impact of feature normalization on the learned representation geometry.

## 1.1 Main contributions

Our *main theoretical contributions* are Theorems 1, 2, and 3, which, under a widely-studied latent data model, hold for any convex, argument-wise non-decreasing contrastive loss function, any non-negative and argument-wise non-decreasing hardening function to generate hard-negative samples, and norm-bounded representations of dimension at least $C - 1$ where $C$ is the number of classes in the dataset.

Theorem 1 establishes that the HSCL loss dominates the SCL loss and similarly the HUCL loss dominates the UCL loss. This is accomplished by creatively utilizing the Harris comparison inequality. In this context we note that Theorem 3.1 in Wu et al. (2020) is a somewhat similar result for the UCL setting, but for a special loss function and it does not address hard-negatives directly and in the generality that we do.

Theorem 2 is a novel result which states that the globally optimal representation geometry for both SCL and HSCL corresponds to Neural-Collapse (NC) (see Definition 6) with the same optimal loss value. In contrast to existing works (see related works section 2), we show that achieving NC in SCL and HSCL does not require class-conditional independence of the positive samples.

Similarly, Theorem 3 establishes the optimality of NC-geometry for UCL if the representation dimension is sufficiently large compared to the number of latent classes which, in turn, is implicitly determined by the joint distribution of the positive examples that corresponds to the augmentation mechanism.

A comprehensive set of experimental results on one synthetic and three real datasets are detailed in Sec. 5 and Appendix C. These include experiments that study effects of two initialization methods, three different feature normalization methods, three different batch sizes, two very different families of hardening functions, and two different CL sampling strategies. Empirical results show that when using the Adam optimizer with random initialization, the matrix of class means for SCL is badly conditioned and effectively low-rank, i.e., it exhibits Dimensional-Collapse (DC). In contrast, the use of hard-negatives at appropriate hardness levels mitigates DC and enables convergence to the global optima. A similar phenomenon is observed in the unsupervised settings. We also show that feature normalization is critical for mitigating DC in these settings. Results are qualitatively similar across different datasets, a range of batch sizes, hardening functions, and CL sampling strategies.

## 2 Related work

### 2.1 Supervised Contrastive Learning

The theoretical results for our SCL setting, where in contrast to Graf et al. (2021) and Khosla et al. (2020) we make use of label information in positive *as well as* negative sampling, are novel. The debiased SCL loss in Chuang et al. (2020) corresponds to our SCL loss, but no analysis pertaining to optimal representation geometry was considered in Chuang et al. (2020). Feeney & Hughes (2023) considers using label information for negative sampling in SCL with InfoNCE loss, calling it the SINCERE loss. Our theoretical results prove that NC is the optimal geometry for the SINCERE loss. Furthermore, our results and analysis also apply to hard-negative sampling as well, a scenario not considered thus far for SCL.

We would like to note that our theoretical set-up for UCL under the sampling mechanism of Figure 1, can be seen to be aligned with the SCL analysis that makes use of label information *only* for positive samples as done in Khosla et al. (2020) and Graf et al. (2021). Therefore, our theoretical results provide an alternative proof of optimality of NC based on simple, compact, and transparent probabilistic arguments complementing the proof of a similar result in Graf et al. (2021). We note that similarly to Graf et al. (2021), our arguments also hold for the case when one approximates the loss using batches.

We want to point out that in all key papers that conduct a theoretical analysis of contrastive learning e.g., Saunshi et al. (2019); Graf et al. (2021); Robinson et al. (2021), the positive samples are assumed to be conditionally i.i.d., conditioned on the label. However, this conditional independence may not hold in practice when using augmentation mechanisms typically considered in CL settings e.g., Oh Song et al. (2016); Oord et al. (2018); Wu et al. (2018); Chen et al. (2020).

Unlike the recent work of Zhou et al. (2022) which shows that the *optimization landscape* of supervised learning with least-squares loss is benign, i.e. all critical points other than the global optima are strict saddle points, in Sec. 5 we demonstrate that the optimization landscape of SCL is more complicated. Specifically, not only may the global optima not be reached by SGD-like methods with random initialization, but also the local optima exhibit the Dimensional-Collapse (DC) phenomenon. However, our experiments demonstrate that these issues are remedied via HSCL whose global optimization landscape may be better. Here we note that Yaras et al. (2022) show that with unit-sphere normalization, Riemannian gradient descent methods can achieve the global optima for SCL, underscoring the importance of optimization methods and constraints for training in CL.

### 2.2 Unsupervised Contrastive Learning

Wang & Isola (2020) argue that asymptotically (in the number of negative samples) the InfoNCE loss for UCL optimizes for a trade-off between the alignment of positive pairs while ensuring uniformity of features on the hypersphere. However, a non-asymptotic and global analysis of the optimal solution is still lacking. In contrast, for UCL in Theorem 3, we show that as long as the embedding dimension is larger than the number of *latent* classes, which in turn is determined by the distribution of the similar samples, the optimal representations in UCL exhibit NC-geometry.

Our results also complement several recent papers, e.g., Parulekar et al. (2023), Wen & Li (2021), Wang et al. (2021), that study the role of augmentations in UCL. Similar to theoretical works analyzing UCL, e.g. Saunshi et al. (2019) and Lee et al. (2021), our results also assume conditional independence of positive pairs given the label. This assumption may or may not be satisfied in practice.

We demonstrate that a recent result, viz., Theorem 4 in Jing et al. (2021), that attempts to explain DC in UCL is limited in that under a suitable initialization, the UCL loss trained with Adam does not exhibit DC (see Sec. 5). Furthermore, we demonstrate empirically, for the first time, that HUCL mitigates DC in UCL at moderate hardness levels. For CL (without hard-negative sampling), Ziyin et al. (2022) characterize local solutions that correspond to DC but leave open the analysis of training dynamics leading to collapsed solutions.

A geometrical analysis of HUCL is carried out in Robinson et al. (2020), but the optimal solutions are only characterized asymptotically (in the number of negative samples) and for the case when hardness also goes to infinity, the analysis seems to require knowledge of supports of class conditional distributions. In contrast, we show that the geometry of the optimal solution for HUCL depends on the hardness level and is, in general, different compared to UCL due to the possibility of class collision.

## 3 Contrastive Learning Framework

### 3.1 Mathematical model

*Notation:* $k, C \in \mathbb{N}, k > 1, C > 1, \mathcal{Y} := \{1, \ldots, C\}, \mathcal{Z} \subseteq \mathbb{R}^{d_{\mathcal{Z}}}$. For $i, j \in \mathbb{Z}$, $i < j$, $i : j := i, i+1, \ldots, j$, and $a_{i:j} := a_i, a_{i+1}, \ldots, a_j$. If $i > j$, $i : j$ and $a_{i:j}$ are "null".

Let $f : \mathcal{X} \to \mathcal{Z}$ denote a (deterministic) representation mapping from data space $\mathcal{X}$ to representation space $\mathcal{Z} \subseteq \mathbb{R}^{d_{\mathcal{Z}}}$. Let $\mathcal{F}$ denote a family of such representation mappings. Contrastive Learning (CL) selects a representation from the family by minimizing an expected loss function that penalizes "misalignment" between the representation of an *anchor* sample $z = f(x)$ and the representation of a *positive* sample $z^+ = f(x^+)$ and simultaneously penalizes "alignment" between $z$ and the representations of $k$ *negative* samples $z_i^- := f(x_i^-), i = 1 : k$.

We consider a CL loss function $\ell_k$ of the following general form.

**Definition 1** (Generalized Contrastive Loss)**.**

$$\ell_k(z, z^+, z_{1:k}^-) := \psi_k(z^\top(z_1^- - z^+), \ldots, z^\top(z_k^- - z^+)) \tag{1}$$

*where $\psi_k : \mathbb{R}^k \to \mathbb{R}$ is a convex function that is also argument-wise non-decreasing (i.e., non-decreasing with respect to each argument when the other arguments are held fixed) throughout $\mathbb{R}^k$.*

This subsumes and generalizes popular CL loss functions such as InfoNCE and triplet-loss with sphere-normalized representations. InfoNCE corresponds to $\psi_k(t_{1:k}) = \log(\alpha + \sum_{i=1}^k e^{t_i})$ with $\alpha > 0$[1] and $\psi_k(t_{1:k}) = \sum_{i=1}^k \max\{t_i + \alpha, 0\}$, $\alpha > 0$, is the triplet-loss with sphere-normalized representations. However, some CL losses such as the spectral contrastive loss of HaoChen et al. (2021) are not of this form.

The CL loss is the expected value of the CL loss function:

$$L_{CL}^{(k)}(f) := \mathbb{E}_{(x, x^+, x_{1:k}^-) \sim p_{CL}}[\ell_k(f(x), f(x^+), f(x_1^-), \ldots, f(x_k^-))]$$

where $p_{CL}(x, x^+, x_{1:k}^-)$ is the joint probability distribution of the anchor, positive, and $k$ negative samples and is designed differently within the supervised and unsupervised settings as described below.

***Supervised CL (SCL):*** Here, all samples have class labels: a *common* class label $y \in \mathcal{Y}$ for the anchor and positive sample and one class label for each negative sample denoted by $y_i^- \in \mathcal{Y}$ for the $i$-th negative sample, $i = 1, \ldots, k$. The joint distribution of all samples and their labels is described in the following equation:

$$p_{SCL}(y, x, x^+, y_{1:k}^-, x_{1:k}^-) := \lambda_y \, q(x, x^+|y) \prod_{i=1}^k r(y_i^-|y) \, s(x_i^-|y_i^-), \tag{2}$$

$$r(y_i^-|y) := \frac{\mathbb{1}(y_i^- \neq y)\lambda_{y_i^-}}{(1 - \lambda_y)} \tag{3}$$

where $\lambda_y \in (0, 1)$ for all $y \in \mathcal{Y}$ is the marginal distribution of the anchor's label and $s(x^-|y^-)$ is the conditional probability distribution of any negative sample $x^-$ given its class $y^-$.

This joint distribution may be interpreted from a sample generation perspective as follows: first, a common class label $y \in \mathcal{Y}$ for the anchor and positive sample is sampled from a class marginal probability distribution $\lambda$.

---

[1]This is the log-sum-exponential function which is convex over $\mathbb{R}^k$ for all $\alpha \geq 0$ and strictly convex over $\mathbb{R}^k$ if $\alpha > 0$.

Then, the anchor and positive samples are generated by sampling from the conditional distribution $q(x, x^+|y)$. Then, given $x, x^+$ and their common class label $y$, the $k$ negative samples and their labels are generated in a conditionally IID manner. The sampling of $y_i^-, x_i^-$, for each $i$, can be interpreted as first sampling a class label $y_i^-$ *different* from $y$ in a manner consistent with the class marginal probability distribution $\lambda$ (sampling from distribution $r(y_i^-|y)$) and then sampling $x_i^-$ from the conditional probability distribution $s(\cdot|\cdot)$ of negative samples given class $y_i^-$. Thus in SCL, the $k$ negative samples are conditionally IID and independent of the anchor and positive sample given the anchor's label.

*In the typical supervised setting, the anchor, positive, and negative samples all share the same common conditional probability distribution $s(\cdot|\cdot)$ within each class given their respective labels.*

We denote the CL loss in the supervised setting by $L_{SCL}^{(k)}(f)$.

*Unsupervised CL (UCL):* Here samples do not have labels or rather they are latent (unobserved) and the $k$ negative samples are IID and independent of the anchor and positive samples.

Latent labels (classes) in UCL can be interpreted as indexing latent clusters. Suppose that there are $C$ latent classes from which the anchor, positive, and the $k$ negative samples can be drawn from. Then the joint distribution of all samples and their latent labels can be described by the following equation:

$$p_{UCL}(y, x, x^+, y_{1:k}^-, x_{1:k}^-) := \lambda_y \, q(x, x^+|y) \prod_{i=1}^{k} r(y_i^-) \, s(x_i^-|y_i^-), \tag{4}$$

where $\lambda$ is the marginal distribution of the anchor's latent label, $r(\cdot)$ the marginal distribution of the latent labels of negative samples, and $s(x^-|y^-)$ is the conditional probability distribution of any negative sample $x^-$ given its latent label $y^-$. We have used the same notation as in SCL (and slightly abused it for $r(\cdot)$) in order to make the similarities and differences between the SCL and UCL distribution structures transparent.

*In the typical UCL setting, $r = \lambda$ and the conditional distribution of $x$ given its label $y$ and the conditional distribution of $x^+$ given its label $y$ are both $s(\cdot|y)$ which is the conditional distribution of any negative sample given its label.*

**Independence relationships between anchor, positive and $k$ negatives in the typical UCL setting:** To preclude potential confusion, we note that the joint distribution given by (4) and the typical UCL setting described above imply the following independence relationships between samples. The anchor-positive-pair is independent of the $k$ negatives. The $k$ negatives are IID. But the anchor sample and the positive sample are *dependent* through their common shared latent class label. In the typical UCL setting, the anchor and positive samples are conditionally IID given their shared label, but they are not (unconditionally) IID. In fact, their dependence through their shared label is crucial. This remains true even in the balanced scenario where all classes are equally likely, i.e., $\lambda_i = 1/C$ for all $i$. Overall, the $(k+2)$ samples (anchor, positive, $k$ negatives) are not IID.

We also note that the number of latent classes, $C$, in the UCL setting, is an unknown intrinsic property of the underlying sampling mechanisms generating the data and is not a tunable parameter of a neural network model used to learn a mapping from data space to representation space.

We denote the CL loss in the unsupervised setting by $L_{UCL}^{(k)}(f)$.

*Anchor and positive samples:* For the SCL scenario, we will consider sampling mechanisms in which the representations of the anchor $x$ and the positive sample $x^+$ have the same conditional probability distribution $s(\cdot|y)$ given their common label $y$ (see (2)). We will not, however, assume that $x$ and $x^+$ are conditionally independent given $y$. This is compatible with settings where $x$ and $x^+$ are generated via IID augmentations of a common reference sample $x^{ref}$ as in SimCLR Chen et al. (2020) (see Fig. 1 for the model and Appendix B for a proof of compatibility of this model).

For UCL, we assume the same mechanism for sampling the positive samples as that for SCL, but the latent label $y$ is unobserved. Further, the negative samples are generated independently of the positive pairs using

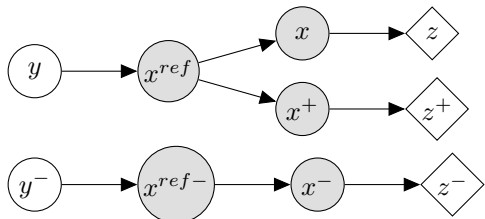

Figure 1: Graphical model for augmentation and negative sampling for SCL and UCL settings used in practical implementations such as Sim-CLR.

the same mechanism, e.g., IID augmentations of an $x^{ref-}$ chosen independently of $x^{ref}$ (see Fig. 1). Thus, the anchor, positive, and negative samples will all have the same marginal distribution given by $\sum_{i=1}^{C} \lambda_i s(\cdot|i)$.

## 3.2 Hard-negative sampling

Hard-negative sampling aims to generate negative samples whose representations are "more aligned" with that of the anchor (making them harder to distinguish from the anchor) compared to a given reference negative sampling distribution (whether unsupervised or supervised). We consider a very general class of "hardening" mechanisms that include several classical approaches as special cases. To this end, we define a ***hardening function*** as follows.

**Definition 2** (Hardening function). *$\eta : \mathbb{R}^k \to \mathbb{R}$ is a hardening function if it is non-negative and argument-wise non-decreasing throughout $\mathbb{R}^k$.*

As an example, $\eta(t_{1:k}) := \prod_{i=1}^{k} e^{\beta t_i}$, $\beta > 0$, is an exponential tilting hardening function employed in Robinson et al. (2021) and Jiang et al. (2023).

***Hard-negative SCL (HSCL)***: From (2) it follows that in SCL, $p(x^-|x, x^+, y) = p(x^-|y) = \sum_{y^- \in \mathcal{Y}} r(y^-|y) s(x^-|y^-) =: p^-_{SCL}(x^-|y)$ is the reference negative sampling distribution for one negative sample and $p(x^-_{1:k}|x, x^+, y) = p(x^-_{1:k}|y) = \prod_{i=1}^{k} p^-_{SCL}(x^-_i|y)$ is the reference negative sampling distribution for $k$ negative samples. Let $\eta$ be a hardening function such that for all $x \in \mathcal{X}$ and all $y \in \mathcal{Y}$,

$$\gamma(x, y, f) := \mathbb{E}_{x^-_{1:k} \sim \text{IID } p^-_{SCL}(\cdot|y)}[\eta(f(x)^\top f(x^-_1), \ldots, f(x)^\top f(x^-_k))] \in (0, \infty).$$

Then we define the $\eta$-harder negative sampling distribution for SCL as follows.

**Definition 3** ($\eta$-harder negatives for SCL).

$$p^-_{HSCL}(x^-_{1:k}|x, x^+, y, f) := \frac{\eta(f(x)^\top f(x^-_1), \ldots, f(x)^\top f(x^-_k))}{\gamma(x, y, f)} \prod_{i=1}^{k} p^-_{SCL}(x^-_i|y). \tag{5}$$

Observe that negative samples which are more aligned with the anchor in the representation space, i.e., $f(x)^\top f(x^-_i)$ is large, are sampled relatively more often in $p^-_{HSCL}$ than in the reference $p^-_{SCL}$ because $\eta$ is argument-wise non-decreasing throughout $\mathbb{R}^k$.

In HSCL, $x^-_{1:k}$ are conditionally independent of $x^+$ given $x$ and $y$ but they are not conditionally independent of $x$ given $y$ (unlike in SCL). Moreover, $x^-_{1:k}$ may not be conditionally IID given $(x, y)$ if the hardening function is not (multiplicatively) separable. We also note that unlike in SCL, $p^-_{HSCL}$ depends on the representation function $f$.

We denote the joint probability distribution of all samples and their labels in the hard-negative SCL setting by $p_{HSCL}$ and the corresponding CL loss by $L^{(k)}_{HSCL}(f)$.

***Hard-negative UCL (HUCL):*** From (4) it follows that in UCL, $p(x^-|x, x^+, y) = p(x^-) = \sum_{y^- \in \mathcal{Y}} r(y^-) s(x^-|y^-) =: p^-_{UCL}(x^-)$ is the reference negative sampling distribution for one negative sample

and $p(x_{1:k}^-|x, x^+, y) = p(x_{1:k}^-) = \prod_{i=1}^k p_{UCL}^-(x_i^-)$ is the reference negative sampling distribution for $k$ negative samples. Let $\eta$ be a hardening function such that for all $x \in \mathcal{X}$,

$$\gamma(x, f) := \mathbb{E}_{x_{1:k}^- \sim \text{IID } p_{UCL}^-}[\eta(f(x)^\top f(x_1^-), \ldots, f(x)^\top f(x_k^-))] \in (0, \infty). \tag{6}$$

Then we define the $\eta$-harder negative sampling probability distribution for UCL as follows.

**Definition 4** ($\eta$-harder negatives for UCL).

$$p_{HUCL}^-(x_{1:k}^-|x, x^+, y, f) := \frac{\eta(f(x)^\top f(x_1^-), \ldots, f(x)^\top f(x_k^-))}{\gamma(x, f)} \cdot \prod_{i=1}^k p_{UCL}^-(x_i^-). \tag{7}$$

Again, observe that negative samples which are more aligned with the anchor in representation space, i.e., $f(x)^\top f(x_i^-)$ is large, are sampled relatively more often in $p_{HUCL}^-$ than in the reference $p_{UCL}^-$ because $\eta$ is argument-wise non-decreasing throughout $\mathbb{R}^k$.

In HUCL, $x_{1:k}^-$ are conditionally independent of $x^+$ given $x$, but they are not independent of $x$ (unlike in UCL). Moreover, $x_{1:k}^-$ may not be conditionally IID given $x$ if the hardening function is not (multiplicatively) separable. We also note that unlike in UCL, $p_{HUCL}^-$ depends on the representation function $f$.

We denote the joint probability distribution of all samples and their (latent labels) in the hard-negative UCL setting by $p_{HUCL}$ and the corresponding CL loss by $L_{HUCL}^{(k)}(f)$.

# 4 Theoretical results

In this section, we present all our theoretical results using the notation and mathematical framework for CL described in the previous section. We will focus on general CL loss in Section 4.1 and Section 4.2 and then extend results to sample-based empirical and batched empirical losses in Section 4.3.

## 4.1 Hard-negative CL loss is not smaller than CL loss

**Theorem 1** (Hard-negative CL versus CL losses). *Let $\psi_k$ in (1) be argument-wise non-decreasing over $\mathbb{R}^k$ and assume that all expectations associated with $L_{UCL}^{(k)}(f)$, $L_{HUCL}^{(k)}(f)$, $L_{SCL}^{(k)}(f)$, $L_{HSCL}^{(k)}(f)$ exist and are finite. Then, for all $f$ and all $k$, $L_{HUCL}^{(k)}(f) \geq L_{UCL}^{(k)}(f)$ and $L_{HSCL}^{(k)}(f) \geq L_{SCL}^{(k)}(f)$.*

We note that convexity of $\psi_k$ is not needed in Theorem 1. The proof of Theorem 1 is based on the generalized (multivariate) association inequality due to Harris, Theorem 2.15 Boucheron et al. (2013) and its corollary which are stated below.

**Lemma 1** (Harris-inequality, Theorem 2.15 in Boucheron et al. (2013)). *Let $g : \mathbb{R}^k \to \mathbb{R}$ and $h : \mathbb{R}^k \to \mathbb{R}$ be argument-wise non-decreasing throughout $\mathbb{R}^k$. If $u_{1:k} \sim \text{IID } p$ then*

$$\mathbb{E}_{u_{1:k} \sim \text{IID } p}[g(u_{1:k})h(u_{1:k})] \geq \mathbb{E}_{u_{1:k} \sim \text{IID } p}[g(u_{1:k})] \cdot \mathbb{E}_{u_{1:k} \sim \text{IID } p}[h(u_{1:k})]$$

*whenever the expectations exist and are finite.*

**Corollary 1.** *Let $\eta : \mathbb{R}^k \to \mathbb{R}$ be non-negative and argument-wise non-decreasing throughout $\mathbb{R}^k$ such that $\gamma := \mathbb{E}_{u_{1:k} \sim \text{IID } p}[\eta(u_{1:k})] \in (0, \infty)$. Let $p_H(u_{1:k}) := \frac{\eta(u_{1:k})}{\gamma} \prod_{i=1}^k p(u_i)$. If $g : \mathbb{R}^k \to \mathbb{R}$ is argument-wise non-decreasing throughout $\mathbb{R}^k$ such that $\mathbb{E}_{u_{1:k} \sim \text{IID } p}[g(u_{1:k})]$ exists and is finite, then*

$$\mathbb{E}_{u_{1:k} \sim p_H}[g(u_{1:k})] \geq \mathbb{E}_{u_{1:k} \sim \text{IID } p}[g(u_{1:k})].$$

**Proof**

$$\mathbb{E}_{u_{1:k} \sim p_H}[g(u_{1:k})]$$

$$= \mathbb{E}_{u_{1:k} \sim \text{IID } p}\left[g(u_{1:k})\frac{\eta(u_{1:k})}{\gamma}\right]$$

$$\geq \mathbb{E}_{u_{1:k} \sim \text{IID } p}[g(u_{1:k})]\frac{\mathbb{E}_{u_{1:k} \sim \text{IID } p}[\eta(u_{1:k})]}{\gamma}$$

$$= \mathbb{E}_{u_{1:k} \sim \text{IID } p}[g(u_{1:k})]$$

where the inequality in the second step follows from the Harris-inequality (see Lemma 1). ∎

*Proof of Theorem 1.* The proof essentially follows from Corollary 1 by defining $u_i := f(x)^\top f(x_i^-)$ for $i = 1 : k$, defining $g_{x,x^+}(u_{1:k}) := \psi_k(u_1 - f(x)^\top f(x^+), \ldots, u_k - f(x)^\top f(x^+))$, noting that $u_{1:k}$ are conditionally IID given $(x, x^+)$ in the UCL setting and conditionally IID given $(x, x^+, y)$ in the SCL setting, and verifying that the conditions of Corollary 1 hold.

For clarity, we provide a detailed proof of the inequality $L_{HSCL}^{(k)}(f) \geq L_{SCL}^{(k)}(f)$. The detailed proof of the inequality $L_{HUCL}^{(k)}(f) \geq L_{UCL}^{(k)}(f)$ parallels that for the (more intricate) supervised setting and is omitted.

$$L_{HSCL}^{(k)}(f) =$$

$$= \mathbb{E}_{(x,x^+,y)}\left[\mathbb{E}_{x_{1:k}^- \sim p_{HSCL}^-(x_{1:k}^-|x,x^+,y,f)}\left[\psi_k(f(x)^\top(f(x_1^-) - f(x^+)), \ldots, f(x)^\top(f(x_k^-) - f(x^+)))\right]\right]$$

$$= \mathbb{E}_{(x,x^+,y)}\left[\mathbb{E}_{x_{1:k}^- \sim \text{IID } p_{SCL}^-(\cdot|y)}\left[\psi_k(f(x)^\top(f(x_1^-) - f(x^+)), \ldots, f(x)^\top(f(x_k^-) - f(x^+))) \cdot \right.\right.$$

$$\left.\left.\frac{\eta(f(x)^\top f(x_1^-), \ldots, f(x)^\top f(x_k^-))}{\gamma(x,y,f)}\right]\right] \tag{8}$$

$$\geq \mathbb{E}_{(x,x^+,y)}\left[\mathbb{E}_{x_{1:k}^- \sim \text{IID } p_{SCL}^-(\cdot|y)}\left[\psi_k(f(x)^\top(f(x_1^-) - f(x^+)), \ldots, f(x)^\top(f(x_k^-) - f(x^+)))\right] \cdot\right.$$

$$\left.\frac{\mathbb{E}_{x_{1:k}^- \sim \text{IID } p_{SCL}^-(\cdot|y)}\left[\eta(f(x)^\top f(x_1^-), \ldots, f(x)^\top f(x_k^-))\right]}{\gamma(x,y,f)}\right] \tag{9}$$

$$= \mathbb{E}_{(x,x^+,y)}\left[\mathbb{E}_{x_{1:k}^- \sim \text{IID } p_{SCL}^-(\cdot|y)}\left[\psi_k(f(x)^\top(f(x_1^-) - f(x^+)), \ldots, f(x)^\top(f(x_k^-) - f(x^+)))\right]\right] \tag{10}$$

$$= L_{SCL}^{(k)}(f)$$

where (8) follows from (5) which defines $p_{HSCL}^-$, (9) follows from the application of the Harris-inequality (see Lemma 1) to the inner expectation where $x$ and $x^+$ are held fixed, and (10) follows from the definition of $\gamma(x,y,f)$ in (6). □

Theorem 1 is used in Theorem 2 to show that both SCL and HSCL achieve NC.

## 4.2 Lower bound for SCL loss and Neural-Collapse

Consider the SCL model with anchor, positive, and $k$ negative samples generated as described in Sec. 3.1. Within this setting, we have the following lower bound for the SCL loss and conditions for equality.

**Theorem 2** (Lower bound for SCL loss and conditions for equality with unit-ball representations and equiprobable classes). *In the SCL model with anchor, positive, and negative samples generated as described in (2) and (3), let (a) $\lambda_y = \frac{1}{C}$ for all $y \in \mathcal{Y}$ (equiprobable classes), (b) $\mathcal{Z} = \{z \in \mathbb{R}^{d_Z} : ||z|| \leq 1\}$ (unit-ball representations), and (c) the anchor, positive and negative samples have a common conditional probability distribution $s(\cdot|\cdot)$ within each class given their respective labels. If $\psi_k$ is a convex function that is also argument-wise non-decreasing throughout $\mathbb{R}^k$, then for all $f : \mathcal{X} \to \mathcal{Z}$,*

$$L_{SCL}^{(k)}(f) \geq \psi_k\left(\tfrac{-C}{(C-1)}, \ldots, \tfrac{-C}{(C-1)}\right). \tag{11}$$

*For a given $f \in \mathcal{F}$ and all $y \in \mathcal{Y}$, let*

$$\mu_y := \mathbb{E}_{x \sim s(x|y)}[f(x)]. \tag{12}$$

*If a given $f \in \mathcal{F}$ satisfies the following additional condition:*

$$\text{Equal inner-product class means: } \forall j, \ell \in \mathcal{Y} : j \neq \ell, \quad \mu_j^\top \mu_\ell = \tfrac{-1}{C-1}, \tag{13}$$

*then equality will hold in (11), i.e., additional condition (13) is **sufficient** for equality in (11). Additional condition (13) also implies the following properties:*

*(i) zero-sum class means: $\sum_{j \in \mathcal{Y}} \mu_j = 0$,*

*(ii) unit-norm class means: $\forall j \in \mathcal{Y}, \|\mu_j\| = 1$,*

*(iii) $d_{\mathcal{Z}} \geq C - 1$.*

*(iv) zero within-class variance: for all $j \in \mathcal{Y}$ and all $i = 1 : k$, $\Pr(f(x) = f(x^+) = \mu_j|y = j) = 1 = \Pr(f(x_i^-) = \mu_j|y_i^- = j)$, and*

*(v) The support sets of $s(\cdot|y)$ for all $y \in \mathcal{Y}$ must be disjoint and the anchor, positive and negative samples must share a common deterministic labeling function defined by the support sets.*

*(vi) equality of HSCL and SCL losses:*

$$L_{HSCL}^{(k)}(f) = L_{SCL}^{(k)}(f) = \psi_k\left(\tfrac{-C}{(C-1)}, \ldots, \tfrac{-C}{(C-1)}\right).$$

*If $\psi_k$ is a **strictly** convex function that is also argument-wise **strictly** increasing throughout $\mathbb{R}^k$, then additional condition (13) is also **necessary** for equality to hold in (11).*

Theorem 2 has a necessary part and a sufficient part. The sufficient part states that if there is an $f$ such that the equal angles condition (13) holds (and we note that this is a property of the family of representation maps and the dataset, but not the loss function), then the lower bound (11) will be achieved for any general $\psi_k$ (convex and argument-wise non-decreasing) and we will have other properties (i)–(vi) as well. The necessary part of Theorem 2 (the last sentence) states that the only way we can get equality in the lower bound is to have the equal angles condition (13).

We note that the results of Theorem 2 hold for any loss function that is convex and argument-wise non-decreasing (see Definition 1), which includes the triplet loss, and is not confined to just the InfoNCE and similar loss functions. In the "Empirical SCL loss" paragraph of Section 4.3, we discuss how condition (13) can be guaranteed provided the family of representation mappings (defined by a neural network) has sufficiently high capacity. This holds irrespective of the loss function.

**Proof** We have

$$L_{SCL}^{(k)}(f) = \mathbb{E}_{x,x^+,x_{1:k}^-}[\psi_k(f(x)^\top(f(x_1^-) - f(x^+)), \ldots, f(x)^\top(f(x_k^-) - f(x^+)))]$$

$$\geq \mathbb{E}_{x,x^+,x_{1:k}^-}[\psi_k(f(x)^\top f(x_1^-) - 1, \ldots, f(x)^\top f(x_k^-) - 1)] \tag{14}$$

$$\geq \psi_k(\mathbb{E}_{x,x_1^-}[f(x)^\top f(x_1^-)] - 1, \ldots, \mathbb{E}_{x,x_k^-}[f(x)^\top f(x_k^-)] - 1) \tag{15}$$

$$= \psi_k(\mathbb{E}_{y,y_1^-}[\mathbb{E}_{x,x_1^-}[f(x)^\top f(x_1^-)|y, y_1^-]] - 1, \ldots, \mathbb{E}_{y,y_k^-}[\mathbb{E}_{x,x_k^-}[f(x)^\top f(x_k^-)|y, y_k^-]] - 1) \tag{16}$$

$$= \psi_k\Big(\sum_{j,\ell \in \mathcal{Y}, j \neq \ell} \tfrac{\mu_j^\top \mu_\ell}{C(C-1)} - 1, \ldots, \sum_{j,\ell \in \mathcal{Y}, j \neq \ell} \tfrac{\mu_j^\top \mu_\ell}{C(C-1)} - 1\Big) \tag{17}$$

$$= \psi_k\Big(\tfrac{\|\sum_{j \in \mathcal{Y}} \mu_j\|^2 - \sum_{j \in \mathcal{Y}} \|\mu_j\|^2}{C(C-1)} - 1, \ldots, \tfrac{\|\sum_{j \in \mathcal{Y}} \mu_j\|^2 - \sum_{j \in \mathcal{Y}} \|\mu_j\|^2}{C(C-1)} - 1\Big) \tag{18}$$

$$\geq \psi_k\Big(\tfrac{0-C}{C(C-1)} - 1, \ldots, \tfrac{0-C}{C(C-1)} - 1\Big) \tag{19}$$

$$= \psi_k\Big(\tfrac{-C}{(C-1)}, \ldots, \tfrac{-C}{(C-1)}\Big) \tag{20}$$

which is the lower bound in (11). Inequality (14) is because $\psi_k$ is argument-wise non-decreasing and $f(x)^\top f(x^+) \leq 1$ by the Cauchy-Schwartz inequality since $\|f(x)\|, \|f(x^+)\| \leq 1$ (unit-ball representations). Inequality (15) is Jensen's inequality applied to the convex function $\psi_k$. Equality (16) is due to the law of iterated expectations. Equality (17) follows from (2), (3), and the assumption of equiprobable classes. Equality (18) is because $\sum_{j,\ell \in \mathcal{Y}} \mu_j^\top \mu_\ell = \sum_{j,\ell \in \mathcal{Y}, j \neq \ell} \mu_j^\top \mu_\ell + \sum_{j \in \mathcal{Y}} \|\mu_j\|^2$. Inequality (19) is because $\psi_k$ is argument-wise non-decreasing and the smallest possible value of $\|\sum_{j \in \mathcal{Y}} \mu_j\|^2$ is zero and the largest possible value of $\|\mu_j\|^2$ is one (unit-ball representations): Jensen's inequality for the *strictly* convex function $\|\cdot\|^2$ together with $\|f(x)\|^2 \leq 1$ (unit-ball representations) imply that for all $y \in \mathcal{Y}$, we have

$$\|\mu_y\|^2 = \|\mathbb{E}_{x \sim s(x|y)}[f(x)]\|^2 \leq \mathbb{E}_{x \sim s(x|y)}[\|f(x)\|^2] \leq 1. \tag{21}$$

Finally the equality (20) is because $\frac{0-C}{C(C-1)} - 1 = \frac{-C}{(C-1)}$.

(*i*) and (*ii*) *Proof that additional condition (13) implies zero-sum and unit-norm class means:* Inequality (21) together with condition (13) implies that

$$0 \leq \|\sum_{j \in \mathcal{Y}} \mu_y\|^2 = \sum_{j,\ell \in \mathcal{Y}} \mu_j^\top \mu_\ell = \underbrace{\sum_{j,\ell \in \mathcal{Y}, j \neq \ell} \underbrace{\mu_j^\top \mu_\ell}_{=\frac{-1}{C-1}}}_{} + \sum_{j \in \mathcal{Y}} \underbrace{\|\mu_j\|^2}_{\leq 1} \leq -\frac{C(C-1)}{C-1} + \sum_{j \in \mathcal{Y}} 1 = -C + C = 0.$$

Thus $\|\sum_{j \in \mathcal{Y}} \mu_y\|^2 = 0$ and for all $j \in \mathcal{Y}$, $\|\mu_j\|^2 = 1$.

(*iii*) Let $M := [\mu_1, \ldots, \mu_C] \in \mathbb{R}^{d_z \times C}$. Then from (13) and (*ii*), the gram matrix $M^\top M = \frac{C}{C-1} I_C - \frac{1}{C-1} \mathbb{1}_C \mathbb{1}_C^\top$ where $I_C$ is the $C \times C$ identity matrix and $\mathbb{1}_C$ is the $C \times 1$ column vector of all ones. From this it follows that $M^\top M$ has one eigenvalue of zero corresponding to eigenvector $\mathbb{1}_C$ and $C-1$ eigenvalues all equal to $\frac{C}{C-1}$ corresponding to $(C-1)$ orthogonal eigenvectors spanning the orthogonal complement of $\mathbb{1}_C$. Thus, $M$ has $C-1$ nonzero singular values all equal to $\sqrt{\frac{C}{C-1}}$ and a rank equal to $C - 1 \leq d_z$.

(*iv*) *Proof that additional condition (13) implies zero within-class variance:* We just proved that additional condition (13) together with the unit-ball representation constraint implies unit-norm class means. This, together with (21) implies that for all $y \in \mathcal{Y}$,

$$1 = \|\mu_y\|^2 = \|\mathbb{E}_{x \sim s(x|y)}[f(x)]\|^2 \leq \mathbb{E}_{x \sim s(x|y)}[\|f(x)\|^2] \leq 1. \tag{22}$$

This implies that we have equality in Jensen's inequality applied to the strictly convex function $g(z) = \|z\|^2$, with $z = f(x)$, and the push-forward conditional probability measure $f_\#(s(x|y))$. Equality can occur iff, with probability one given $y$, we have $f(x) = \mu_y$ (since $\|\cdot\|^2$ is strictly convex).

Since the anchor, positive and negative samples all have a common conditional probability distribution $s(\cdot|\cdot)$ within each class given their respective labels, it follows that for all $j \in \mathcal{Y}$ and all $i = 1 : k$, $\Pr(f(x) = \mu_j|y = j) = \Pr(f(x^+) = \mu_j|y = j) = \Pr(f(x_i^-) = \mu_j|y_i^- = j) = 1$. Moreover, since the anchor and positive samples have the same label, for all $j \in \mathcal{Y}$, with probability one given $y = j$, we have $f(x) = f(x^+) = \mu_j$.

*Proof that additional condition (13) is sufficient for equality to hold in (11):* From the proofs of (*i*), (*ii*), and (*iv*) above, if additional condition (13) holds, then we showed that with probability one given $y = j$ we have $f(x) = f(x^+) = \mu_j$ (see the para below (22)). This equality of $f(x)$ and $f(x^+)$ is a conditional equality given the class. Since this is true for all classes, it implies equality (with probability one) of $f(x)$ and $f(x^+)$ without conditioning on the class:

$$\Pr(f(x) = f(x^+)) = \sum_{j \in \mathcal{Y}} \frac{\Pr(f(x) = f(x^+)|y = j)}{C} = 1. \tag{23}$$

From (23) we get

$$\Pr(f(x)^\top f(x^+) = 1) = \Pr(\|f(x)\|^2 = 1) = \sum_{j \in \mathcal{Y}} \frac{\Pr(\|f(x)\|^2 = 1|y = j)}{C} = 1, \tag{24}$$

since $f(x) = f(x^+) = \mu_j$ with probability one given $y = j$ and $\|\mu_j\|^2 = 1$. Equality in (14) then follows from (23) and (24). Moreover, due to zero within-class variance we will have

$$\text{with probability one, for all } i = 1 : k, f(x)^\top f(x_i^-) = \mu_y^\top \mu_{y_i^-} = \tfrac{-1}{C-1}, \tag{25}$$

and then we will have equality in (15) and (19). Therefore additional condition (13) is a sufficient condition for equality to hold in (11).

($v$) *Proof that support sets of $s(\cdot|y), y \in \mathcal{Y}$ are disjoint:* From part ($iv$), all samples belonging to the support set of $s(\cdot|y), y \in \mathcal{Y}$, are mapped to $\mu_y$ by $f$. From part ($ii$) and condition (13), distinct labels have distinct representation means: for all $y, y' \in \mathcal{Y}$, if $y' \neq y$, then $\mu_y \neq \mu_{y'}$. Therefore the support sets of $s(\cdot|y)$ for all $y \in \mathcal{Y}$ must be disjoint. Since the anchor, positive, and negative samples all share a common conditional probability distribution $s(\cdot|\cdot)$ and the same marginal label distribution $\lambda$, it follows that they share a common conditional distribution of label given sample (labeling function). Since the support sets of $s(\cdot|y)$ for all $y \in \mathcal{Y}$ are disjoint, the labeling function is deterministic and is defined by the support set to which a sample belongs.

($vi$) *Proof of equality of HSCL and SCL losses under additional condition (13):* under the equal inner-product class means condition, with probability one $f(x)^\top f(x_i^-) = \tfrac{-1}{C-1}$ simultaneously for all $i = 1 : k$ and $\eta(f(x)^\top f(x_1^-), \dots, f(x)^\top f(x_k^-)) = \eta(\tfrac{-1}{C-1}, \dots, \tfrac{-1}{C-1})$, a constant. Consequently, for all $x, y$ and the given $f$, we must have $\gamma(x, y, f) = \eta(\tfrac{-1}{C-1}, \dots, \tfrac{-1}{C-1})$ which would imply that (see Equation 5) $p_{HSCL}^-(x_{1:k}^-|x, x^+, y, f) = \prod_{i=1}^k p_{SCL}^-(x_i^-|y)$ and therefore $L_{HSCL}^{(k)}(f) = L_{SCL}^{(k)}(f) = \psi_k\big(\tfrac{-C}{(C-1)}, \dots, \tfrac{-C}{(C-1)}\big)$ where the last equality is because additional condition (13) is sufficient for equality to hold in (11).

*Proof that additional condition (13) is necessary for equality in (11) if $\psi_k$ is strictly convex and argument-wise strictly increasing over $\mathbb{R}^k$:* If equality holds in (11), then it must also hold in (14), (15), and (19). If $\psi_k$ is argument-wise strictly increasing, then equality in (19) can only occur if all class means have unit norms. Then, from (22) and the reasoning in the paragraph below it, we would have zero within-class variance and equations (23) and (24). This would imply equality in (14). If $\psi_k$ is strictly convex then equality in (15), which is Jensen's inequality, can only occur if for all $i = 1 : k$, $\Pr(f^\top(x)f(x_i^-) = \beta_i) = 1$ for some constants $\beta_{1:k}$. Since $(x, x_i^-)$ has the same distribution for all $i$, it follows that for all $i$, $\beta_i = \beta$ for some constant $\beta$. Since we have already proved zero within-class variance and the labels of negative samples are always distinct from that of the anchor, it follows that for all $j \neq \ell$, we must have $\mu_j^\top \mu_\ell = \beta$. Since we have equality in (14) and $\psi_k$ is argument-wise strictly increasing, we must have $\beta = \tfrac{-1}{C-1}$ which implies that additional condition (13) must hold (it is a necessary condition). ∎

**Remark 1.** *We note that Theorem 2 also holds if we have unit-sphere representations (a stronger constraint) as opposed to unit-ball representations, i.e., if $\mathcal{Z} = \{z \in \mathbb{R}^{d_\mathcal{Z}} : \|z\| = 1\}$: the lower bound (11) holds since the unit sphere is a subset of the unit ball and equality can be attained with unit-sphere representations in Theorem 2.*

**Remark 2.** *Interestingly, we note that inequality (19) and therefore the lower bound of Theorem 2 also holds if we replace the unit-ball constraint on representations $\|f(x)\| \leq 1$ with the weaker requirement $\tfrac{1}{C} \sum_{j=1}^C \|\mu_j\|^2 \leq 1$.*

**Definition 5** (ETF)**.** *The equal inner-product, zero-sum, and unit-norm conditions on class means in Theorem 2 define a (normalized) Equiangular Tight Frame (ETF) (see Malozemov & Pevnyi (2009)).*

**Definition 6** (CL Neural-Collapse (NC))**.** *We will say representation map $f(\cdot)$ exhibits CL Neural-Collapse if it has zero within-class variance as in condition ($iv$) of Theorem 2 and the class means in representation space form a normalized ETF as in Definition 5.*

**Remark 3.** *The term "Neural-Collapse" was originally used for representation mappings implemented by deep classifier neural networks (see Papyan et al. (2020)). However, here we use the term more broadly for any family of representation mappings and within the context of CL instead of classifier training.*

The following corollary is a partial restatement of Theorem 2 in terms of CL Neural-Collapse:

**Corollary 2.** *Under the conditions of Theorem 2, equality in (11) is attained by any representation map $f$ that exhibits CL Neural-Collapse. Moreover, if $\psi_k$ is strictly convex and argument-wise strictly increasing over $\mathbb{R}^k$, then equality in (11) is attained by a representation map $f$, if, and only if, it exhibits CL Neural-Collapse.*

### 4.3 Empirical and batched empirical SCL losses

All results presented till this point pertain to the expected loss and not sample-based empirical loss or batched empirical loss. In this section we demonstrate that results that were established in previous sections for expected loss also hold for the empirical and batched empirical losses.

*Empirical SCL loss:* Theorem 2 also holds for **empirical** SCL loss because simple averages over samples can be expressed as expectations with suitable uniform distributions over the samples. If the family of representation mappings $\mathcal{F}$ has sufficiently high capacity (e.g., the family of mappings implemented by a sufficiently deep and wide feed-forward neural network) and $\forall y \in \mathcal{Y}$, $s(\cdot|y)$ is a discrete probability mass function (pmf) over a finite set (e.g., uniform pmf over training samples within each class) with support-sets that are disjoint across different classes, then the equal inner-product condition (13) in Theorem 2 can be satisfied for a suitable $f$ in the family. If either convexity or monotonicity of $\psi_k$ is not strict, e.g., $\psi_k(t_{1:k}) = \sum_{i=1}^k \max\{t_i + \alpha, 0\}$, $\alpha > 0$, then it may be possible for a representation map $f$ to attain the lower bound without exhibiting CL Neural-Collapse.

*Batched empirical SCL loss:* We will now show that representations that exhibit CL Neural-Collapse will also minimize batched empirical SCL loss under the conditions of Theorem 2. Here, the full data set has balanced classes (equal number of samples in each class) but is partitioned into $B$ **disjoint** nonempty batches of possibly unequal size. Let $b$ denote the batch index and $n_b$ the number of samples in batch $b$. Let $\mathbb{E}_{x,x^+,x^-_{1:k}|b}[\cdot]$ denote the empirical SCL loss in batch $b$ and

$$L_{SCL}^{(k,b)}(f) := \sum_{b=1}^{B} \frac{1}{n_b} \mathbb{E}_{x,x^+,x^-_{1:k}|b}\Big[\psi_k(f(x)^\top(f(x^-_1) - f(x^+)), \ldots, f(x)^\top(f(x^-_k) - f(x^+)))\Big]$$

the overall batched empirical SCL loss. Note that in a given batch the data may not be balanced across classes and therefore we cannot simply use Theorem 2, which assumes balanced classes, to deduce the optimality of CL Neural-Collapse representations.

We lower bound the batched empirical SCL loss as follows:

$$L_{SCL}^{(k,b)}(f) = \sum_{b=1}^{B} \frac{1}{n_b} \mathbb{E}_{x,x^+,x^-_{1:k}|b}\Big[\psi_k(f(x)^\top(f(x^-_1) - f(x^+)), \ldots, f(x)^\top(f(x^-_k) - f(x^+)))\Big]$$

$$\geq \sum_{b=1}^{B} \frac{1}{n_b} \mathbb{E}_{x,x^+,x^-_{1:k}|b}\Big[\psi_k(f(x)^\top f(x^-_1) - 1, \ldots, f(x)^\top f(x^-_k) - 1)\Big] \tag{26}$$

$$\geq \psi_k\bigg(\sum_{b=1}^{B} \frac{1}{n_b} \mathbb{E}_{x,x^-_1|b}[f(x)^\top f(x^-_1)] - 1, \ldots, \sum_{b=1}^{B} \frac{1}{n_b} \mathbb{E}_{x,x^-_k}[f(x)^\top f(x^-_k)] - 1\bigg) \tag{27}$$

$$= \psi_k\Big(\mathbb{E}_{x,x^-_1}[f(x)^\top f(x^-_1)] - 1, \ldots, \mathbb{E}_{x,x^-_k}[f(x)^\top f(x^-_k)] - 1\Big) \tag{28}$$

$$\geq \psi_k\Big(\tfrac{-C}{(C-1)}, \ldots, \tfrac{-C}{(C-1)}\Big) \tag{29}$$

where inequality (26) holds for the same reason as in (14), inequality (27) is Jensen's inequality applied to $\psi_k$ which is convex, and equality (28) is due to the law of iterated (empirical) expectation. The right side of (27) is precisely the right side of (15) and therefore (29) follows from (15) – (20). From the above analysis it follows that the arguments used to prove Theorem 2 can be applied again to prove that the conclusions of Theorem 2 and Corollary 2 also hold for the batched empirical SCL loss.

### 4.4 Lower bound for UCL loss with latent labels and Neural-Collapse

Consider the UCL model with anchor, positive, and $k$ negative samples generated as described in Sec. 3.1. Within this setting, we have the following lower bound for the UCL loss and conditions for equality.

**Theorem 3** (Lower bound for UCL loss with latent labels and conditions for equality with unit-ball representations and equiprobable classes)**.** *In the UCL model with anchor, positive, and negative samples generated as described in (4), let (a) $\lambda_y = \frac{1}{C}$ for all $y \in \mathcal{Y}$ (equiprobable classes), (b) $\mathcal{Z} = \{z \in \mathbb{R}^{d_\mathcal{Z}} : ||z|| \leq 1\}$ (unit-ball representations), (c) the anchor, positive and negative samples have a common conditional probability distribution $s(\cdot|\cdot)$ within each latent class given their respective labels, (d) $r = \lambda$ in (4), and (e) the anchor and positive samples are conditionally independent given their common label, i.e., $q(x, x^+|y) = s(x|y)s(x^+|y)$ in (4).[2] If $\psi_k$ is a convex function that is also argument-wise non-decreasing throughout $\mathbb{R}^k$, then for all $f : \mathcal{X} \to \mathcal{Z}$,*

$$L_{UCL}^{(k)}(f) \geq \frac{1}{C^{k+1}} \sum_{y, y_{1:k}^- \in \mathcal{Y}} \psi_k \left( \frac{-C\,1(y_1^- \neq y)}{(C-1)}, \dots, \frac{-C\,1(y_k^- \neq y)}{(C-1)} \right) \tag{30}$$

*where $1(\cdot)$ is the indicator function. For a given $f \in \mathcal{F}$ and all $y \in \mathcal{Y}$, let $\mu_y$ be as defined in (12). If a given $f \in \mathcal{F}$ satisfies additional condition (13), then equality will hold in (30), i.e., additional condition (13) is **sufficient** for equality in (30). Additional condition (13) also implies the following properties:*

*(i) zero-sum class means: $\sum_{j \in \mathcal{Y}} \mu_j = 0$,*

*(ii) unit-norm class means: $\forall j \in \mathcal{Y}, \|\mu_j\| = 1$,*

*(iii) $d_\mathcal{Z} \geq C - 1$.*

*(iv) zero within-class variance: for all $j \in \mathcal{Y}$ and all $i = 1 : k$, $\Pr(f(x) = f(x^+) = \mu_j|y = j) = 1 = \Pr(f(x_i^-) = \mu_j|y_i^- = j)$, and*

*(v) The support sets of $s(\cdot|y)$ for all $y \in \mathcal{Y}$ must be disjoint and the anchor, positive and negative samples must share a common deterministic (latent) labeling function defined by the support sets.*

*If $\psi_k$ is a **strictly** convex function that is also argument-wise **strictly** increasing throughout $\mathbb{R}^k$, then additional condition (13) is also **necessary** for equality to hold in (30).*

**Proof** For $i = 1 : k$, we define the following indicator random variables $b_i := 1(y_i^- \neq y)$ and note that for all $i = 1 : k$, $b_i$ is a deterministic function of $(y, y_i^-)$. Since $y \perp\!\!\!\perp \{y_{1:k}^-\}$ and $y_{1:k}^- \sim$ IID Uniform$(\mathcal{Y})$, it follows that $b_{1:k} \sim$ IID and independent of $y$. We then have the following sequence of inequalities:

$$L_{UCL}^{(k)}(f) = \mathbb{E}_{x,x^+,x_{1:k}^-} \left[ \psi_k \big( f(x)^\top (f(x_1^-) - f(x^+)), \dots, f(x)^\top (f(x_k^-) - f(x^+)) \big) \right]$$

$$\geq \mathbb{E}_{y,y_{1:k}^-} \big[ \psi_k ( \mathbb{E}_{x,x_1^- \sim s(x|y)s(x_1^-|y_1^-)}[f(x)^\top f(x_1^-)] - \mathbb{E}_{x,x^+ \sim s(x|y)s(x^+|y)}[f(x)^\top f(x^+)], \dots,$$

$$\dots, \mathbb{E}_{x,x_k^- \sim s(x|y)s(x_k^-|y_k^-)} \big[ f(x)^\top f(x_k^-) \big] - \mathbb{E}_{x,x^+ \sim s(x|y)s(x^+|y)} \big[ f(x)^\top f(x^+) \big] ) \big] \tag{31}$$

$$= \mathbb{E}_{y,y_{1:k}^-} \left[ \psi_k \big( \mu_y^\top \mu_{y_1^-} - \mu_y^\top \mu_y, \dots, \mu_y^\top \mu_{y_k^-} - \mu_y^\top \mu_y \big) \right] \tag{32}$$

$$\geq \mathbb{E}_{b_{1:k}} \left[ \psi_k \big( \mathbb{E}[\mu_y^\top \mu_{y_1^-} - ||\mu_y||^2 | b_{1:k}], \dots, \mathbb{E}[\mu_y^\top \mu_{y_k^-} - ||\mu_y||^2 | b_{1:k}] \big) \right] \tag{33}$$

$$= \mathbb{E}_{b_{1:k}} \left[ \psi_k \big( \mathbb{E}[\mu_y^\top \mu_{y_1^-} - ||\mu_y||^2 | b_1], \dots, \mathbb{E}[\mu_y^\top \mu_{y_k^-} - ||\mu_y||^2 | b_k] \big) \right] \tag{34}$$

$$= \mathbb{E}_{b_{1:k}} \left[ \psi_k \big( b_1 \mathbb{E}[\mu_y^\top \mu_{y_1^-} - ||\mu_y||^2 | b_1 = 1], \dots, b_k \mathbb{E}[\mu_y^\top \mu_{y_k^-} - ||\mu_y||^2 | b_k = 1] \big) \right] \tag{35}$$

$$\geq \mathbb{E}_{b_{1:k}} \left[ \psi_k \big( b_1 \mathbb{E}[\mu_y^\top \mu_{y_1^-} - 1 | b_1 = 1], \dots, b_k \mathbb{E}[\mu_y^\top \mu_{y_k^-} - 1 | b_k = 1] \big) \right] \tag{36}$$

$$= \mathbb{E}_{b_{1:k}} \left[ \psi_k \big( \tfrac{b_1 \sum_{\ell \neq j} (\mu_j^\top \mu_\ell - 1)}{C(C-1)}, \dots, \tfrac{b_k \sum_{\ell \neq j} (\mu_j^\top \mu_\ell - 1)}{C(C-1)} \big) \right] \tag{37}$$

$$= \mathbb{E}_{b_{1:k}} \left[ \psi_k \big( \tfrac{b_1 (\| \sum_j \mu_j \|^2 - \sum_j ||\mu_j||^2 - C(C-1))}{C(C-1)}, \dots, \tfrac{b_k (\| \sum_j \mu_j \|^2 - \sum_j ||\mu_j||^2 - C(C-1))}{C(C-1)} \big) \right] \tag{38}$$

---

[2]As discussed in Sec. 2, all existing works that conduct a theoretical analysis of UCL make this assumption.

$$\geq \mathbb{E}_{b_{1:k}}\left[\psi_k\left(\tfrac{b_1(0-C-C(C-1))}{C(C-1)}, \ldots, \tfrac{b_k(0-C-C(C-1))}{C(C-1)}\right)\right] \tag{39}$$

$$= \mathbb{E}_{b_{1:k}}\left[\psi_k\left(\tfrac{b_1(0-C\cdot C)}{C(C-1)}, \ldots, \tfrac{b_k(0-C\cdot C)}{C(C-1)}\right)\right]$$

$$= \mathbb{E}_{b_{1:k}}\left[\psi_k\left(\tfrac{-Cb_1}{(C-1)}, \ldots, \tfrac{-Cb_k}{(C-1)}\right)\right] \tag{40}$$

$$= \frac{1}{C^{k+1}} \sum_{y, y_{1:k}^- \in \mathcal{Y}} \psi_k\left(\frac{-C\mathbb{1}(y_1^- \neq y)}{(C-1)}, \ldots, \frac{-C\mathbb{1}(y_k^- \neq y)}{(C-1)}\right) \tag{41}$$

where the validity of each numbered step in the above sequence of inequalities is explained below.

Inequality (31) is Jensen's inequality conditioned on $y, y_{1:k}$ applied to the convex function $\psi_k$. Equality (32) holds because for every $i$, we have $x$ and $x_i^-$ are conditionally independent given $y$ and $y_i^-$ (per the UCL model (4)), $x$ and $x^+$ are conditionally independent given their common label (assumption (d) in the theorem statement), and the class means in representation space are as defined in (12). Inequality (33) is Jensen's inequality conditioned on $b_{1:k}$ applied to the convex function $\psi_k$. Equality (34) holds because for all $i = 1:k$, $(y, y_i^-) \perp\!\!\!\perp \{b_\ell, \ell \neq i\}|b_i$. Equality (35) holds because $b_i$ only takes values 0, 1 and if $b_i = 0$, then $\mu_{y_i^-} = \mu_y$ and $\mu_y^\top \mu_{y_i^-} - \|\mu_y\|^2 = 0$. Therefore the expressions to the right of the equality symbols in (34) and (35) match when $b_i = 0$ and when $b_i = 1$. Inequality (36) is because $\psi_k$ is non-decreasing and $\|\mu_y\| \leq 1$ for all $y$ because all representations are in the unit closed ball. Equality (37) holds because for each $i = 1:k$, $y, y_i^- \sim$ IID Uniform($\mathcal{Y}$) and $y \neq y_i^-$ when $b_i = 1$. Equality (38) follows from elementary linear algebraic operations. Inequality (39) holds because $\psi_k$ is argument-wise non-decreasing, the smallest possible value for $\|\sum_j \mu_j\|^2$ is zero and the largest possible value for $\|\mu_j\|^2$, for all $j$, is one. Equality (41) follows from the definition of the indicator variables in terms of $y, y_{1:k}^-$ and because $y, y_{1:k}^-$ are IID Uniform($\mathcal{Y}$).

Similarly to the proof of Theorem 2, if additional condition (13) holds for some $f$, then properties $(i)$–$(v)$ in Theorem 3 hold. Moreover, then (23), (24), and (25) also hold and then we will have equality in (31), (33), (36) and (39). Thus additional condition (13) is sufficient for equality to hold in (30).

*Proof that additional condition (13) is necessary for equality in (30) if $\psi_k$ is strictly convex and argument-wise strictly increasing over $\mathbb{R}^k$:* If equality holds in (30), then it must also hold in (31), (33), (36), and (39). If $\psi_k$ is argument-wise strictly increasing, then equality in (39) can only occur if all class means have unit norms (which would also imply equality in (36)). Then, from (22) and the reasoning in the paragraph below it, we would have zero within-class variance and (24), which would imply that with probability one for all $i$, $f(x)^\top f(x_i^-) = \mu_y^\top \mu_{y_i^-}$ and $f(x)^\top f(x^+) = \|\mu_y\|^2$ which would imply equality in (31) as well. Equality in (33) together with strict convexity of $\psi_k$ and $\|\mu_y\|^2 = 1$ would imply that with probability one, for all $i$, given $b_i$ we must have $\mu_y^\top \mu_{y_i^-} =$ some deterministic function of $b_i$ and if $\psi_k$ is also argument-wise strictly increasing, then this function must be such that $\mu_y^\top \mu_{y_i^-} - 1 = \frac{-Cb_i}{(C-1)}$ due to (40). This would imply that for all $i$, we must have $\mu_y^\top \mu_{y_i^-} = 1 - \frac{Cb_i}{(C-1)} = 1 - \frac{C\mathbb{1}(y_i^- \neq y)}{(C-1)}$. Thus for all $y \neq y_i^-$, i.e., $b_i = 1$ (and this has nonzero probability), $\mu_y^\top \mu_{y_i^-} = 1 - \frac{C}{C-1} = \frac{-1}{C-1}$ which is the additional condition (13). Thus the additional condition (13) must hold (it is a necessary condition). ∎

Counterparts of Corollary 2 and results for empirical SCL loss and batched empirical SCL loss can be stated and derived for UCL.

A novel perspective offered by Theorem 3, vis-à-vis existing theoretical works on UCL, is that it relates the structure of the underlying sampling mechanism to the optimal value of the contrastive loss, albeit under certain restrictive assumptions which may not be possible to verify in practice for a given sampling mechanism (especially the latent-class-conditional independence of positive pairs).

One important difference between results for SCL and UCL is that Theorem 3 is missing the counterpart of property $(vi)$ in Theorem 2. **Unlike in Theorem 2, we cannot assert that if the lower bound is attained, then we will have** $L_{HUCL}^{(k)}(f) = L_{UCL}^{(k)}(f)$**.** This is because in the UCL and HUCL settings, the

negative sample can come from the same latent class as the anchor (latent class collision) with a positive probability ($\frac{1}{C^2}$). Then for a representation $f$ that exhibits Neural-Collapse, we cannot conclude that with probability one we must have $\eta(f(x)^\top f(x_1^-), \ldots, f(x)^\top f(x_k^-)) = \eta(\beta, \ldots, \beta)$, for some constant $\beta$. ***Deriving a tight lower bound for HUCL and determining whether it can be attained iff there is Neural-Collapse in UCL (under suitable conditions), are open problems.***

Neural-Collapse in SCL or UCL requires that the representation space dimension $d_{\mathcal{Z}} \geq C - 1$ (see part (*iii*) of Theorems 2 and 3). This can be ensured in practical implementations of SCL since labels are available and the number of classes is known. In UCL, however, not just latent labels, but even the number of latent classes is unknown. Thus even if it was possible to attain the global minimum of the empirical UCL loss with an $f$ exhibiting Neural-Collapse, ***we may not observe Neural-Collapse with an argument-wise strictly increasing and strictly convex $\psi_k$ unless $d_{\mathcal{Z}}$ is chosen to be sufficiently large.***

In practice, even without knowledge of latent labels, it is possible to design a sampling distribution having a structure that is compatible with (4) and conditions (c) and (d) of Theorem 3, e.g, via IID augmentations of a reference sample as in SimCLR illustrated in Fig. 1. However, it is impossible to ensure that the equiprobable latent class condition (a) or the anchor-positive conditional independence condition (e) in Theorem 3 hold or that the supports of $s(\cdot|\cdot)$ determined by the sample augmentation mechanism will be disjoint across all latent classes. Thus a representation minimizing UCL loss may not exhibit Neural-Collapse even if $\psi_k$ is strictly convex and argument-wise strictly increasing or it might exhibit zero within-class variance, but the class means may not form an ETF (see Nguyen et al. (2024)).

A second important difference between theoretical results for SCL and UCL is that unlike in Theorem 2, conditional independence of the anchor and positive samples given their common label is assumed in Theorem 3 (condition (d) in the theorem). It is unclear whether the results of Theorem 3 for the UCL setting will continue to hold in entirety without conditional independence and we leave this as an open problem.

Finally, unlike the SCL loss lower bound given by Equation (11) which is a non-decreasing function of the number of classes $C$ (since $\psi_k$ is argument-wise non-decreasing), the UCL loss lower bound given by Equation (30) is a non-increasing function of the number of latent classes $C$. We prove this monotonicity property of the UCL loss lower bound in Appendix A for the case when the number of negative samples is $k = 1$. We also provide empirical evidence of the monotonicity property for $k = 1, 2, 3, 4, 5$.

## 5 Practical achievability of global optima

We first verify our theoretical results for UCL, SCL, HUCL, and HSCL using a synthetic data and then investigate the achievability of global-optima on three real-world image datasets.

**Datasets:**

- **Synthetic Dataset:** This comprises three classes with 100 data points per class. The points within each class are IID with a 3072-dimensional Gaussian distribution having an identity covariance matrix and a randomly generated mean vector having IID components that are uniformly distributed over $[-1, 1]$. The data dimension of 3072 allows for reshaping the vector into a $32 \times 32 \times 3$ tensor which can be processed by ResNet.

- **Real-world Image Datasets:** This comprises CIFAR10, CIFAR100 (Krizhevsky et al., 2009), and TinyImageNet (Le & Yang, 2015). These datasets consist of $32 \times 32 \times 3$ images across 10 classes (CIFAR10), 100 classes (CIFAR100), and 200 classes (TinyImageNet), respectively. Similar phenomena are observed in all three datasets. We present results for CIFAR100 here and results for CIFAR10 and TinyImageNet in Appendix C.

**Anchor-positive pairs:** We explored the following three strategies for constructing anchor-positive pairs:

- **Using Label Information:** For each anchor sample, a positive sample is chosen uniformly from among all samples having the same label as the anchor (including the anchor). All samples that

share the same label with the anchor are used to form the positive pairs. Note that the anchor and the positive can be identical.

- **Additive Gaussian Noise Augmentation Mechanism:** For each (reference) sample, we generate the anchor sample by adding IID zero-mean Gaussian noise of variance 0.01 to all dimensions of the reference sample. The corresponding positive sample is generated in the same way from the reference sample using noise that is independent of that used to generate the anchor.

- **SimCLR framework:** We also report additional results with the SimCLR framework in Appendix C.5 where a positive pair is generated using two independent augmentations from one reference sample.

We note that anchor-positive pair sampling using label information satisfies the assumptions on the sampling mechanism described in Theorem 3. In particular, the positive samples are conditionally IID given their latent class label (but not unconditionally IID). In sampling via independent additive Gaussian noise augmentations of a reference sample, if the noise level is not high, the anchor and positive samples are likely to have the same latent class label as the reference, but they are not guaranteed to be conditionally IID given the label. In the SimCLR framework, the assumptions of Theorem 3 cannot be guaranteed.

**Negatives:** In all the supervised settings, for a given positive pair, we select $k$ negative samples independently and uniformly at random from all the data within a mini-batch and all their augmentations (if used), but having labels different from that of the positve pair. In all the unsupervised settings, for a given positive pair, we select $k$ negative samples independently and uniformly at random from all the data within a mini-batch and all their augmentations (if used), including possibly the anchor and/or the positive sample. Thus, we do not use label information to sample negatives in all the unsupervised settings. We call this *random negative sampling.*

**Hard Negatives:** We utilize the InfoNCE loss with the exponential tilting hardening function described in Sec. 3.1. Within each minibatch, when we sample negatives, we use the hardening function to increase the probability of selecting negative samples that have a higher value of $f(x)^\top f(x^-)$ by a multiplicative factor equal to the hardening function value at $f(x)^\top f(x^-)$. In Appendix C.4 we report additional results for a family of polynomial hardening functions.

**Representation family, normalization, $k$ values:** We used ResNet-50, He et al. (2016), as the family of representation mappings $\mathcal{F}$ and set the representation dimension to $d = C - 1$ to observe Neural-Collapse (Definition 5). We normalized representations to be within a unit ball as detailed in Algorithm 1, lines 5-12, in Appendix C. We only report results for $k = 256$ negative samples, but observed that results change only slightly for all $k \in [32, 512]$.

**Other parameters:** We chose hyper-parameter $\beta$ of the hardening function from the set $\{0, 10, 30, 50\}$ for synthetic data and the set $\{0, 2, 5, 10, 30\}$ for real data and trained each model for $E = 400$ epochs with a batch size of $B = 512$ using the Adam optimizer at a learning rate of $10^{-3}$. We did not use weight decay. Computations were performed on an NVIDIA A100 32 GB GPU.

## 5.1 Results for synthetic data

Figure 2 summarizes the results for synthetic data. For a representation function $f^*$ that achieves Neural-Collapse, the values of $L_{SCL}^{(256)}(f^*)$ and $L_{HSCL}^{(256)}(f^*)$ across all $\beta$ values and the value of $L_{UCL}^{(256)}(f^*)$ are 0.2014, 0.2014, and 0.3935, respectively. These values are obtained by numerically evaluating the lower bounds in Theorem 2 and Theorem 3.

The first row in Figure 2 shows the result using label information for positive pair construction.

We note that label information is used for sampling positive pairs even in the unsupervised settings to empirically validate the results of Theorem 3. Sampling positive pairs with label information guarantees the conditional independence assumption (e) in Theorem 3.

The values of the minimum loss in different settings are displayed at the top of Fig. 2. Our simulation results are consistent with our theoretical results. After 200 training epochs, we observed that $L_{SCL}^{(256)}(f)$ and $L_{HSCL}^{(256)}(f)$ across all $\beta$ values and $L_{UCL}^{(256)}(f)$ converged to their minimum values. From the figure, we can visually confirm that the representations exhibit Neural-Collapse in $SCL$, $HSCL$, and $UCL$. However, Neural-Collapse was not observed in HUCL as the class means deviate significantly from the ETF geometry.

The second row in Figure 2 shows the results using the additive Gaussian noise augmentation mechanism, where label information is not used in constructing positive examples.

We note that the conditional independence assumption (e) in Theorem 3 is not guaranteed in this method for sampling positive pairs. In accordance with our theoretical results, NC is observed in $SCL$ and $HSCL$ (recall that the conditional independence assumption is not needed in Theorem 2). However, NC is not observed in $UCL$ and $HUCL$, likely due to the lack of conditional independence. Moreover, the degree of deviation from NC increases progressively with increasing hardness levels.

Contrary to a widely held belief that hard-negative sampling is beneficial in both supervised and unsupervised contrastive learning, results on this simple synthetic dataset suggest that not only may SCL not benefit from hard negatives, but UCL may suffer from it. To investigate whether these conclusions hold true more generally or whether there are practical benefits of hard-negatives for SCL and UCL, we turn to real-world datasets next.

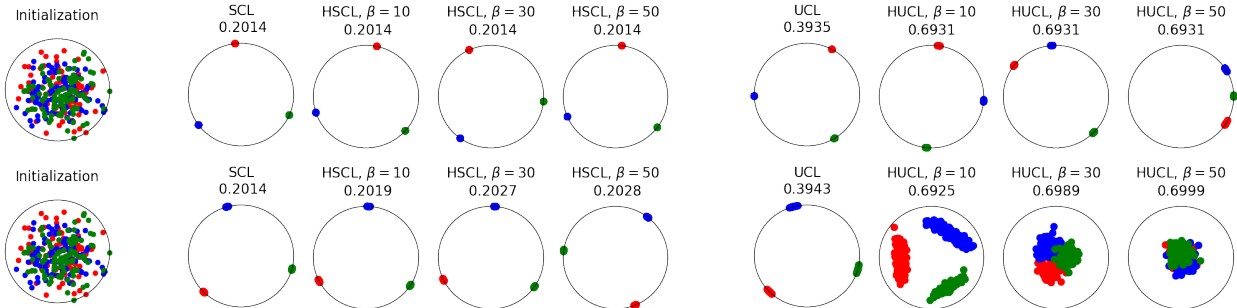

Figure 2: Synthetic dataset results using label information (top row figures) or additive Gaussian noise augmentation mechanism (bottom row figures) for generating anchor-positive pairs: Initial two-dimensional representations (left), post-training SCL and HSCL representations and losses at different hardness levels (middle), post-training UCL and HUCL representations and losses at different hardness levels (right).

## 5.2  Results for real data

To validate the theoretical results on real data, in all four CL settings (UCL, SCL, HUCL, and HSCL), for a given anchor $x$, we randomly sample (without augmentation) the positive sample $x^+$ from the class conditional distribution corresponding to the class of $x$.

We first show that **hard-negative sampling improves downstream classification performance**. From the first row of Fig. 3, we see that with negative sampling at moderate hardness levels ($\beta = 5, 10$), the classification accuracies of HSCL and HUCL are more than 40% points higher than those of SCL and UCL respectively.

**Achievability of Neural-Collapse:** We investigate whether SCL and UCL losses trained using Adam can attain the globally optimal solution exhibiting NC. To test this, in line with properties $(i), (ii)$ and condition (13) in Theorems 2 and 3, we employ the following metrics which are plotted in Fig. 3 in rows two through four.

1. **Zero-sum metric**: $\left\| \sum_{j \in \mathcal{Y}} \mu_j \right\|$
2. **Unit-norm metric**: $\frac{1}{C} \sum_{j \in \mathcal{Y}} \left| \|\mu_j\| - 1 \right|$

3. **Equal inner-product metric:** $\frac{1}{C(C-1)} \sum\limits_{\substack{j,k \in \mathcal{Y}, \\ k \neq j}} \left| \mu_j^\top \mu_k + \frac{1}{C-1} \right|$

We note that even though the equal inner-product class means condition together with unit-ball normalization implies the zero-sum and unit-norm conditions, we report these three metrics separately to gain more insight.

According to Theorems 2 and 3, the optimal solutions for UCL, SCL, and HSCL are anticipated to manifest NC. However, our experimental findings reveal a gap between the theoretical expectations and the observed outcomes. Specifically, in both supervised and unsupervised settings, when leveraging the random negative sampling method, i.e., when the negative samples are sampled uniformly from the whole data in a mini-batch which may include the anchor and positive samples, NC is not exactly achieved: the zero-sum and equal inner-product metrics do not approach zero for all hardness levels (rows 2 and 4 in Fig. 3). This is also supported by the results in Table 1 which shows the theoretical minimum SCL loss value and the practically observed SCL loss values after 400 epochs for different hardness levels. While our theoretical results posit that both SCL and HSCL should have the same minimum loss value for all values of $\beta > 0$, the practically observed loss value for SCL deviates noticeably from theory. On the other hand, increased hardness in HSCL, especially at $\beta = 5, 10$, brings the observed loss value close to the theoretical minimum.

| Theory | Empirical | | | | |
|---|---|---|---|---|---|
| | SCL | HSCL | | | |
| 0.3105 | $\beta = 0$ | $\beta = 2$ | $\beta = 5$ | $\beta = 10$ | $\beta = 30$ |
| | 0.3384 | 0.3603 | **0.3106** | **0.3107** | 0.3222 |

Table 1: Comparison of minimum theoretical loss and practically observed loss values after 400 epochs for CIFAR100.

In addition, the manner in which the final values of NC metrics change with increasing hardness levels is qualitatively different in the supervised and unsupervised settings. Specifically, in the supervised settings, increased hardness invariably leads to improved results, and the model tends to approach NC, notably at $\beta = 5, 10, 30$. However, in the unsupervised settings, there seems to be just a single optimal hardness level ($\beta = 5$ is best among the choices tested).

### 5.3 Dimensional-Collapse

To gain further insights, we investigate the phenomenon of Dimensional-Collapse (DC) that is known to occur in contrastive learning (see Jing et al. (2021)).

**Definition 7.** *[Dimensional-Collapse (DC)] We say that the class means $\mu_1, \ldots, \mu_C$ suffer from DC if their empirical covariance matrix has one or more singular values that are zero or orders of magnitude smaller than the largest singular value.*

If $d_{\mathcal{Z}} = C - 1$, then under Neural-Collapse (NC), the class mean vectors would have full rank $C - 1$ in representation space since they form an ETF (see Definition 5). Thus when $d_{\mathcal{Z}} = C - 1$, $NC \Rightarrow \neg DC$. However, we note that $\neg DC \nRightarrow NC$ because, for example, the class means could have full rank and satisfy the equal inner-product and unit-norm conditions in Theorem 2 without satisfying the zero-sum condition.

We numerically assess DC by plotting the singular values of the empirical covariance matrix of the class means (at the end of training) *normalized by the largest singular value* in decreasing order on a log-scale. Results for UCL, SCL, HUCL, and HSCL are shown in Fig. 4. In the supervised settings, (first row and first column of Fig. 4), the results align with our previous observations from Fig. 3. However, in the unsupervised settings (first row and second column of Fig. 4), while HUCL with high hardness values deviates more from NC compared to UCL in Fig. 3, in Fig. 4 we see that HUCL suffers less from DC.

We must emphasize that observing NC on the training data does not necessarily imply better performance in downstream tasks or improved generalization ability. Our experiments show that using hard-negative

sampling aids the attainment of NC when using iterative methods starting at a random initialization. But more significantly, even if hard-negative sampling may not completely ensure the attainment of the NC solution (which is the optimal solution minimizing the training loss), it could prevent or reduce DC.

### 5.4 Role of initialization

To gain further insights into the DC phenomenon, we trained a model using HSCL with $\beta = 10$ for 400 epochs until it nearly attains NC as measured by the three NC metrics (zero-sum, unit-norm, and equal inner-product) shown in the second row of Table 2. We call this representation mapping (or pre-trained model) the "near-NC" representation mapping (pre-trained model).

Next, with the near-NC representation mapping as initialization, we continue to train the model for an additional 400 epochs under 10 different settings corresponding to hard supervised and hard unsupervised contrastive learning with different hardness levels. Rows 3-12 in Table 2 show the final values of the three NC metrics for the 10 settings. The resulting normalized singular value plots are shown in the second row of Fig. 4.

| Setting | Zero-sum | Unit-norm | Equal inner-product |
|---|---|---|---|
| near-NC model | 0.012 | $1.8 \times 10^{-8}$ | 0.004 |
| SCL | 0.006 | $2.0 \times 10^{-8}$ | 0.007 |
| HSCL, $\beta = 2$ | 0.005 | $2.2 \times 10^{-8}$ | 0.004 |
| HSCL, $\beta = 5$ | 0.007 | $1.9 \times 10^{-8}$ | 0.002 |
| HSCL, $\beta = 10$ | 0.007 | $2.0 \times 10^{-8}$ | 0.002 |
| HSCL, $\beta = 30$ | 0.004 | $1.7 \times 10^{-8}$ | 0.001 |
| UCL | 0.005 | $3.9 \times 10^{-4}$ | 0.005 |
| HUCL, $\beta = 2$ | 0.005 | $6.1 \times 10^{-4}$ | 0.003 |
| HUCL, $\beta = 5$ | 0.004 | $1.4 \times 10^{-3}$ | 0.002 |
| HUCL, $\beta = 10$ | 0.093 | $1.5 \times 10^{-3}$ | 0.047 |
| HUCL, $\beta = 30$ | 0.761 | $7.9 \times 10^{-4}$ | 0.586 |

Table 2: Post-training NC metrics for near-NC initialization in different settings.

From Table 2 we note that in all 5 supervised settings, the final representation mappings have NC metrics that are very similar to those of initial near-NC mapping. However, in the unsupervised settings, especially HUCL for $\beta = 10, 30$, the unit-norm and equal inner-product metrics of the final representation mappings are significantly larger than those of the initial near-NC mapping. This shows that mini-batch Adam optimization of CL losses exhibit dynamics that are different in the supervised and unsupervised settings and are impacted by the hardness level of the negative samples.

From the second row of Fig. 4 we make the following observations:

- SCL and HSCL trained with near-NC initialization and Adam do not exhibit DC or DC is negligible (second row and first column of Fig. 4).

- UCL trained with near-NC initialization and Adam also does not exhibit DC, but the behavior of HUCL depends on the hardness level $\beta$. A larger $\beta$ value appears to make DC more pronounced. This could be explained by the fact that a higher $\beta$ value increases the odds of latent-class collision.

### 5.5 Role of normalization

Feature normalization also plays an important role in alleviating DC. To demonstrate this, we test three normalization conditions during training: (1) unit-ball normalization, (2) unit-sphere normalization, and (3) no normalization. The resulting normalized singular value plots are shown in Fig. 4 (rows 1, 3, and 4). As can be observed, the behavior of unit-sphere normalization is close to that of unit-ball normalization, and

with hard-negative sampling, both SCL and UCL can achieve NC (for suitable hardness levels). Without normalization, neither random-negative nor hard-negative training methods attain NC and they suffer from DC. We also observe that for SCL and UCL, absence of normalization leads to less DC (compare blue curves in rows 1 and 4 of Fig. 4). However, feature normalization could potentially reduce DC in hard-negative sampling for a range of hardness levels.

As we discussed in the paragraph following Definition 7, one way to avoid DC is to achieve NC. As shown in Theorem 2 and Theorem 3 (and Remarks 1 and 2 following Theorem 2), one condition for NC is unit-ball or unit-sphere normalization of feature vectors. This may partially explain why feature normalization helps prevent or mitigate DC. On the other hand, as discussed in the paragraph following Definition 7, it is not necessary to achieve NC in order to avoid DC. We leave it to future work to study more general conditions that help avoid DC.

## 5.6  Impact of batch size

In Appendix C.3, we report results using different batch sizes.

In Section 4.3 we showed that NC can occur in Contrastive Learning with any value of batch size if certain conditions are satisfied. Thus, in theory, NC could be observed regardless of the value of the batch size. In practice, however, it may be harder to achieve NC with a very small or a very large batch size as we explain below.

**Large batch size effects:** When the batch size is too large, the neural network may require a greater representation capacity to map a larger number of samples from the same class into the same point (class-collapse). This could be solved by increasing the capacity of the neural network by expanding its size. In our experimental results, however, we only used standard architectures and observed that NC is consistently achieved for all batch sizes ranging from 64 to 512.

**Small batch size effects:** When the batch size is decreased to 64, NC is still evident in HSCL and HUCL for certain values of $\beta$. However, when the batch size is further reduced to 32, NC is no longer observed. One potential reason for this is the presence of 100 distinct classes in our dataset. With a batch size of 32, the anchor can be compared with samples only from a very limited number of classes, reducing the diversity of negative pairs. This suggests that using larger batch sizes can contribute to more stable and robust results in the context of high class-cardinality.

## 5.7  Role of hardening function

We investigated the impact of hardening functions, using a family of polynomial functions detailed in Appendix C.4. Results shown in Figs. 12 and 13 of Appendix C.4 confirm that hard-negative sampling helps prevent DC and achieve NC.

## 5.8  Experiments using the SimCLR framework

In Appendix C.5 we report results using the state-of-the-art SimCLR framework for contrastive learning which uses augmentations to generate samples. The results of Figs. 14 and 15 in Appendix C.5 are somewhat similar to those in Figs. 3 and 4, respectively, but a key difference is the failure to attain NC in the SimCLR sampling framework for both supervised and unsupervised settings at all hardness levels. This is primarily because the SimCLR sampling framework does not utilize label information and cannot guarantee property $(v)$ in Theorem 2 and Theorem 3 nor guarantee (for UCL) the conditional independence of anchor and positive samples given their label.

## 6  Conclusion and open questions

We proved the theoretical optimality of the NC-geometry for SCL, UCL, and notably (for the first time) HSCL losses for a very general family of CL losses and hardening functions that subsume popular choices. We empirically demonstrated the ability of hard-negative sampling to achieve global optima for CL and mitigate

dimensional-collapse, in both supervised and unsupervised settings. Our theoretical and empirical results motivate a number of open questions. Firstly, a tight lower bound for HUCL remains open due to latent-class collision. It is also unclear whether the HUCL loss is minimized iff there is Neural-Collapse.

If the number of classes is more than the dimension of the latent space plus one, then ETF is no longer the optimal solution, and the lower bound in Theorem 3 is not achievable. In this case, we may need other approaches based on convex optimization as described in Nguyen et al. (2024).

Our theoretical results for the SCL setting did not require conditional independence of anchor and positive samples given their label, but our results for the UCL setting did. A theoretical characterization of the geometry of optimal solutions for UCL in the absence of conditional independence remains open. A difficulty with empirically observing NC in UCL and HUCL is that the number of latent classes is not known because it is, in general, implicitly tied to the properties of the sampling distribution, and this requires one to choose a sufficiently large representation dimension. Another open question is to unravel precisely how and why hard-negatives alter the optimization landscape enabling the training dynamics of Adam with random initialization to converge to the global optimum for suitable hardness levels and what are optimum hardness levels.

## 7 Acknowledgements

This research was supported by NSF 1931978.

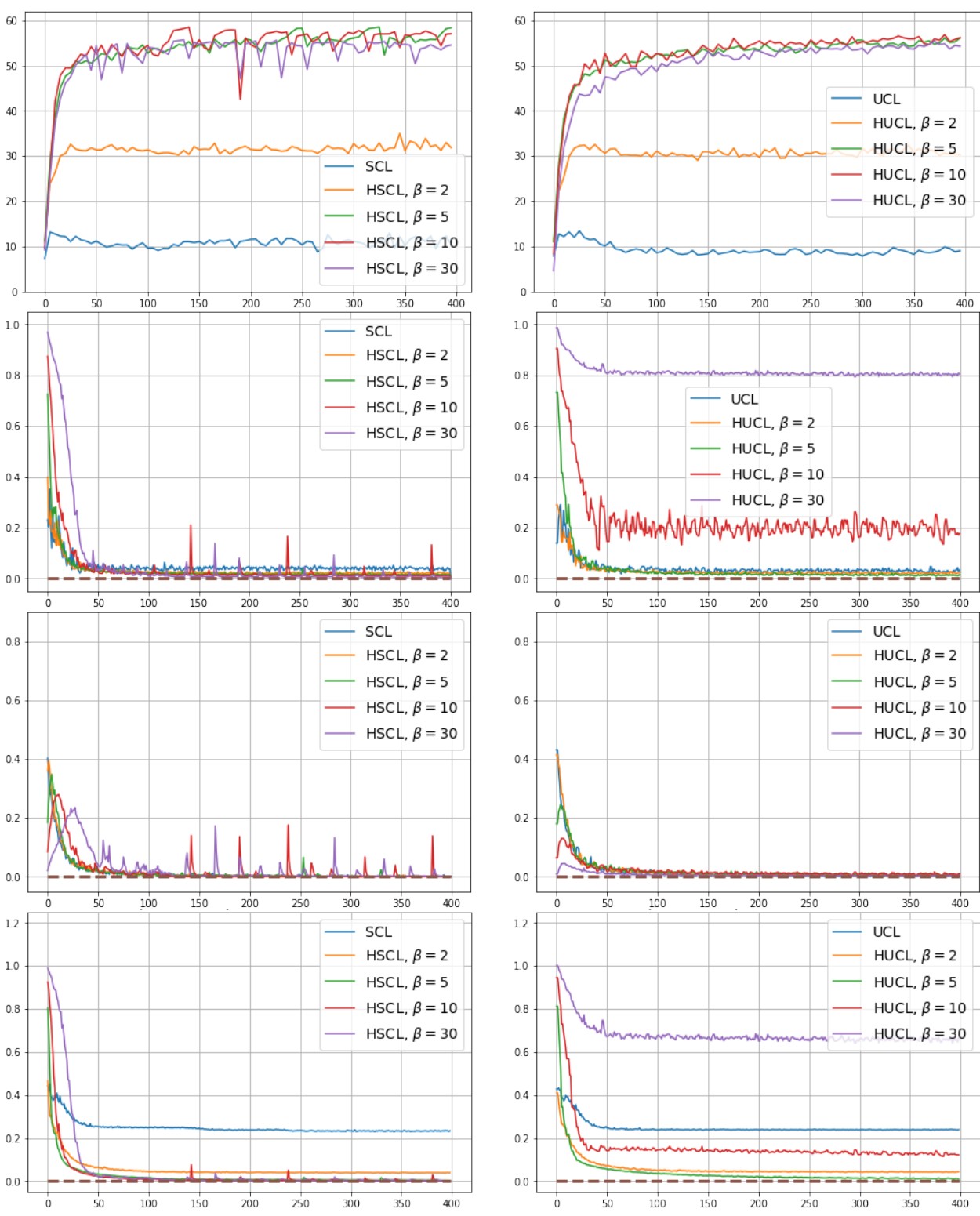

Figure 3: Results for CIFAR100 under supervised settings (SCL, HSCL, left column) and unsupervised settings (UCL, HUCL, right column) with unit-ball normalization and random initialization. From top to bottom: Downstream Test Accuracy, Zero-sum metric, Unit-norm metric, and Equal inner-product metric, all plotted against the number of epochs.

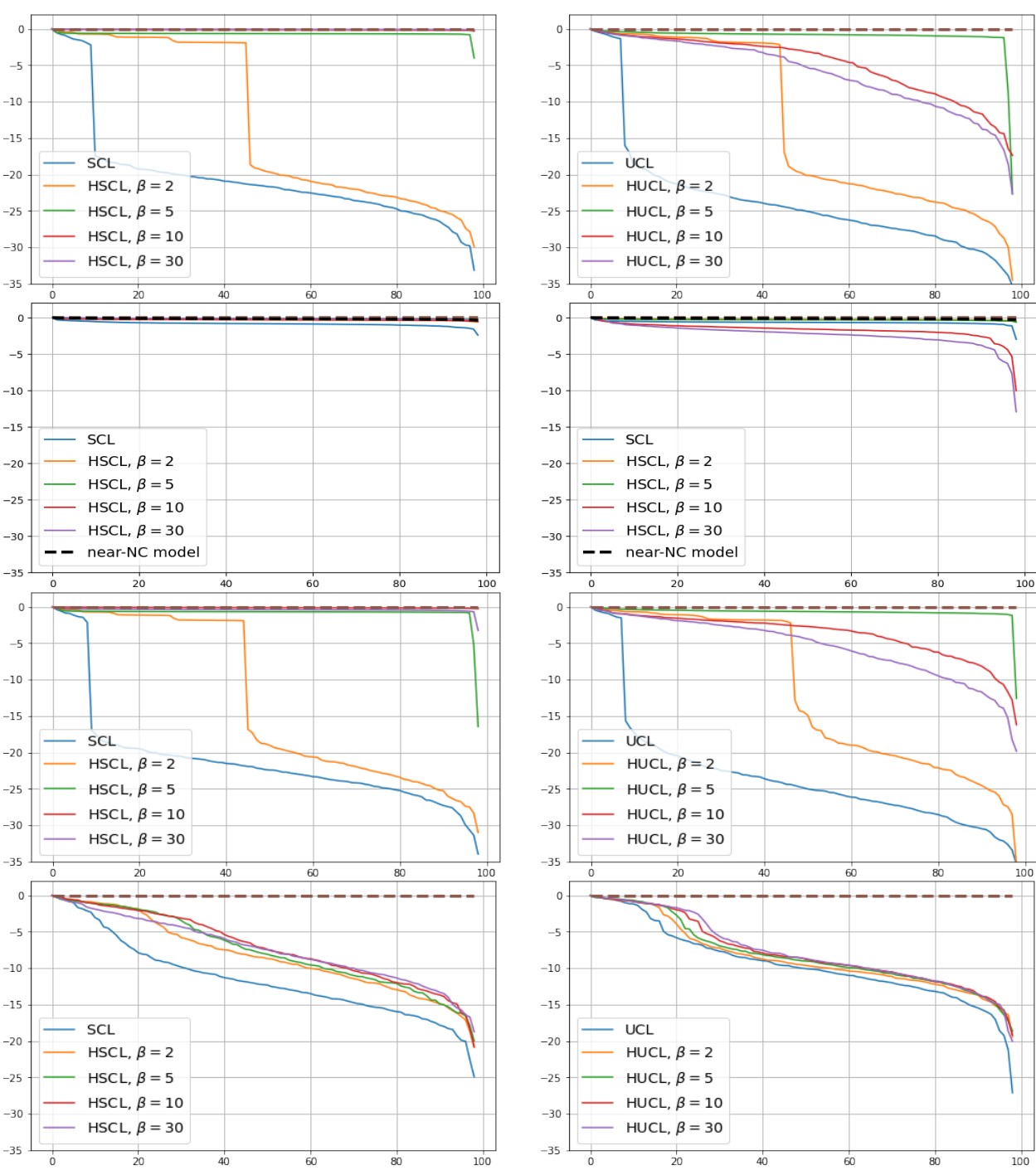

Figure 4: Normalized singular values of the empirical covariance matrix of class means (in representation space) plotted in log-scale in decreasing order for CIFAR100 under supervised (left column) and unsupervised (right column) settings. The horizontal axis is the sorted index of the singular values. From top to bottom: Unit-ball normalization with random initialization, Unit-ball normalization with near-NC initialization, Unit-sphere normalization with random initialization, and un-normalized representation with random initialization.

# A  Monotonicity of SCL and UCL loss lower bounds

In this section, we discuss certain monotonicity properties of the SCL and UCL loss lower bounds in Theorem 2 and Theorem 3, respectively.

The SCL loss lower bound is given by (11), specifically,

$$L_{SCL}^{(k)}(f) \geq \psi_k \left( \tfrac{-C}{(C-1)}, \ldots, \tfrac{-C}{(C-1)} \right). \tag{42}$$

Since $\psi_k(.)$ is argument-wise non-decreasing, the SCL loss lower bound is a non-decreasing function of the number of classes $C$.

Interestingly, the UCL loss lower bound given by (30), i.e.,

$$L_{UCL}^{(k)}(f) \geq \frac{1}{C^{k+1}} \sum_{y,y_{1:k}^- \in \mathcal{Y}} \psi_k \left( \tfrac{-C\,1(y_1^- \neq y)}{(C-1)}, \ldots, \tfrac{-C\,1(y_k^- \neq y)}{(C-1)} \right), \tag{43}$$

is also monotonic in $C$, but in contrast to the SCL loss lower bound, it is a monotonically non-increasing function of $C$. This conclusion is not transparent since although the outer term $\frac{1}{C^{k+1}}$ is decreasing in $C$, the inner term $\sum_{y,y_{1:k}^- \in \mathcal{Y}} \psi_k \left( \tfrac{-C\,1(y_1^- \neq y)}{(C-1)}, \ldots, \tfrac{-C\,1(y_k^- \neq y)}{(C-1)} \right)$ is a non-decreasing function in $C$.

For the case $k = 1$, *i.e.*, when there is a single negative sample, we prove below that the UCL loss lower bound is, indeed, a non-increasing function of $C$. We then provide numerical evidence to support our conjecture that this result is true for all $k > 1$.

When $k = 1$, we can rewrite the UCL loss lower bound as follows:

$$\text{Lower-bound}_{\text{UCL}}(C) \quad = \quad \frac{1}{C^{k+1}} \sum_{y,y_{1:k}^- \in \mathcal{Y}} \psi_k \left( \tfrac{-C\,1(y_1^- \neq y)}{(C-1)}, \ldots, \tfrac{-C\,1(y_k^- \neq y)}{(C-1)} \right) \tag{44}$$

$$= \quad \tfrac{1}{C^2} \sum_{y,y_1^- \in \mathcal{Y}} \psi_1 \left( \tfrac{-C1(y_1^- \neq y)}{C-1} \right) \tag{45}$$

$$= \quad \tfrac{1}{C^2} \sum_{y \in \mathcal{Y}} \sum_{y_1^- \in \mathcal{Y}} \psi_1 \left( \tfrac{-C1(y_1^- \neq y)}{C-1} \right) \tag{46}$$

$$= \quad \tfrac{1}{C^2} C \left( \sum_{y_1^- \neq y} \psi_1 \left( \tfrac{-C1(y_1^- \neq y)}{C-1} \right) + \sum_{y_1^- = y} \psi_1 \left( \tfrac{-C1(y_1^- \neq y)}{C-1} \right) \right) \tag{47}$$

$$= \quad \tfrac{1}{C} \left( (C-1)\psi_1 \left( \tfrac{-C}{C-1} \right) + \psi_1 (0) \right). \tag{48}$$

Next, we show that $\text{Lower-bound}_{\text{UCL}}(C)$ is non-increasing in $C$ by showing that $\text{Lower-bound}_{\text{UCL}}(C) - \text{Lower-bound}_{\text{UCL}}(C+1) \geq 0$ for all $C > 0$. Indeed,

$$\text{Lower-bound}_{\text{UCL}}(C) - \text{Lower-bound}_{\text{UCL}}(C+1) \tag{49}$$

$$= \quad \tfrac{1}{C} \left( (C-1)\psi_1 \left( \tfrac{-C}{C-1} \right) + \psi_1 (0) \right) - \tfrac{1}{C+1} \left( C\psi_1 \left( \tfrac{-(C+1)}{C} \right) + \psi_1 (0) \right) \tag{50}$$

$$= \quad \tfrac{1}{C(C+1)} \left[ (C+1)(C-1)\psi_1 \left( \tfrac{-C}{C-1} \right) + (C+1)\psi_1 (0) - C^2 \psi_1 \left( \tfrac{-(C+1)}{C} \right) - C\psi_1 (0) \right] \tag{51}$$

$$= \quad \tfrac{1}{C(C+1)} \left[ (C^2 - 1)\psi_1 \left( \tfrac{-C}{C-1} \right) + \psi_1 (0) - C^2 \psi_1 \left( \tfrac{-(C+1)}{C} \right) \right] \tag{52}$$

$$= \quad \tfrac{C}{C+1} \left[ \tfrac{C^2-1}{C^2} \psi_1 \left( \tfrac{-C}{C-1} \right) + \tfrac{1}{C^2} \psi_1 (0) - \psi_1 \left( \tfrac{-(C+1)}{C} \right) \right] \tag{53}$$

$$\geq \quad \tfrac{C}{C+1} \left[ \psi_1 \left( \tfrac{C^2-1}{C^2} \times \tfrac{-C}{C-1} + \tfrac{1}{C^2} \times 0 \right) - \psi_1 \left( \tfrac{-(C+1)}{C} \right) \right] \tag{54}$$

$$= \quad \tfrac{C}{C+1} \left[ \psi_1 \left( \tfrac{-(C+1)}{C} \right) - \psi_1 \left( \tfrac{-(C+1)}{C} \right) \right] \tag{55}$$

$$= 0 \tag{56}$$

where the inequality in (54) is due to Jensen's inequality applied to the convex function $\psi_1(.)$, and other rows are just algebraic transformations. Since Lower-bound$_{\text{UCL}}(C) \geq$ Lower-bound$_{\text{UCL}}(C+1)$, it follows that Lower-bound$_{\text{UCL}}(C)$ is a non-increasing function of $C$.

Extending this result to larger values of $k$ is algebraically more complicated. Therefore, instead of proving it in full generality, we numerically plot the UCL loss lower bound for values of $C$ ranging from 2 through 20 and values of $k$ ranging from 1 through 5. Figure 5 shows plots of the UCL loss lower bound for the InfoNCE loss function whereas Fig. 6 shows plots for the triplet loss function.

Recall that the InfoNCE loss function corresponds to $\psi_k(t_{1:k}) = \log(\alpha + \sum_{i=1}^{k} e^{t_i})$, with $\alpha > 0$, and the triplet loss function (with sphere-normalized representations) is given by $\psi_k(t_{1:k}) = \sum_{i=1}^{k} \max\{t_i + \alpha, \, 0\}$, $\alpha > 0$ . We set $\alpha = 1$ in both the InfoNCE and the triplet loss functions.

These plots show that the UCL loss lower bound always decreases with $C$ for any given value of $k$ and for any given value of $C$ it always increases with $k$.

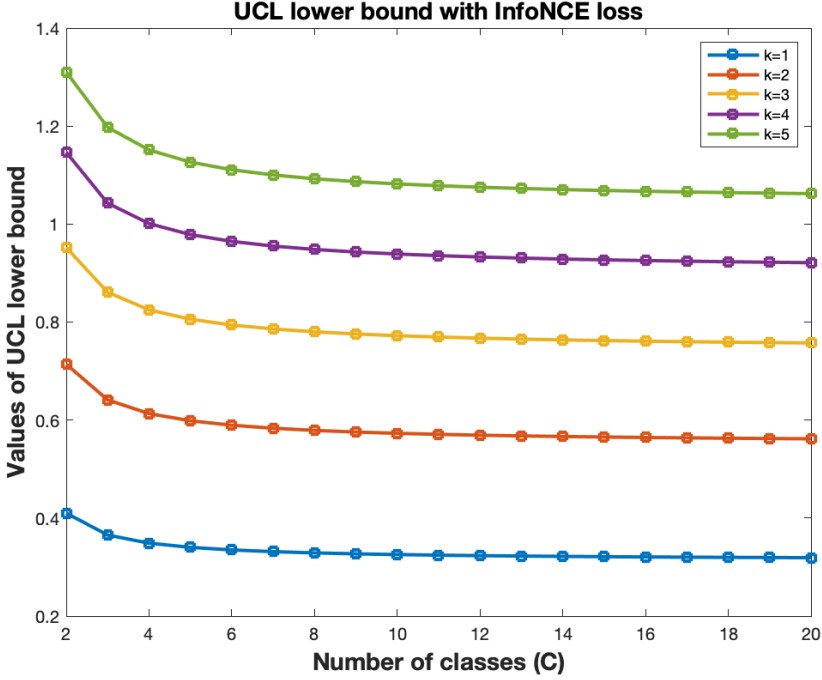

Figure 5: The values of the UCL loss lower-bound in Theorem 3 for the InfoNCE loss function, for $k = 1, 2, 3, 4, 5$, negative samples and $C = 2, 3, \ldots, 20$, latent classes.

## B  Compatibility of sampling model with SimCLR-like augmentations

The following generative model captures the manner in which many data augmentation mechanisms, including SimCLR, generate a pair of anchor and positive samples. First, a label $y$ is sampled. The label may represent an observed class in the supervised setting or a latent (implicit, unobserved) class index in the unsupervised setting. Then given $y$, a reference sample $x^{ref}$ is sampled with conditional distribution $p_{ref}(\cdot|y)$. Then a pair of samples $(x, x^+)$ are generated given $(x^{ref}, y)$ via two independent calls to an augmentation mechanism whose behavior can be statistically described by a conditional probability distribution $p_{aug}(\cdot|x^{ref}, y)$, i.e., $p(x, x^+|x^{ref}, y) = p_{aug}(x|x^{ref}, y) \cdot p_{aug}(x^+|x^{ref}, y)$. Finally, the representations $z = f(x)$ and $z^+ = f(x^+)$ are computed via a mapping $f(\cdot)$, e.g., a neural network. Under the setting just described, it follows that

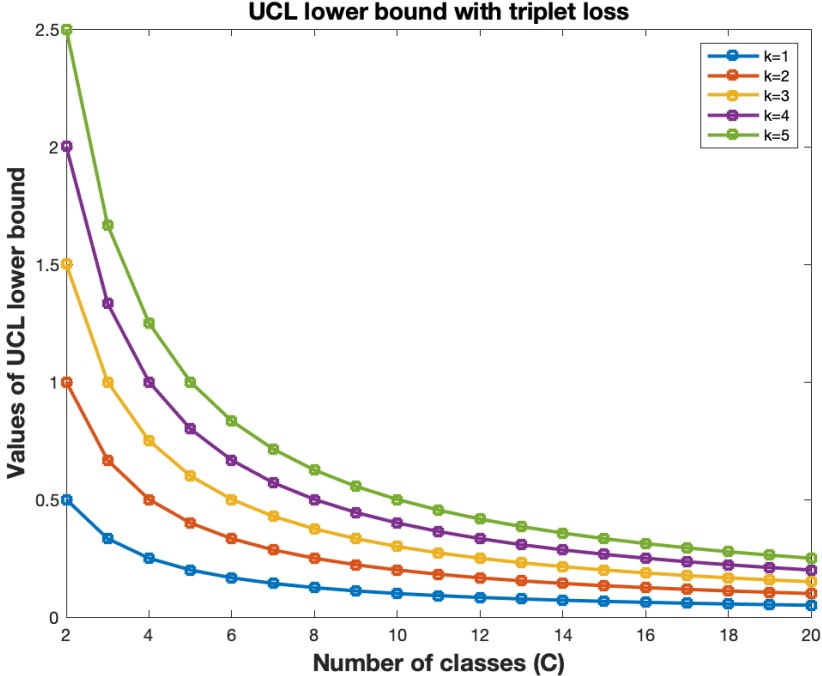

Figure 6: The values of the UCL loss lower-bound in Theorem 3 for the triplet loss function, for $k = 1, 2, 3, 4, 5$, negative samples and $C = 2, 3, \ldots, 20$, latent classes.

both $z|y$ and $z^+|y$ have identical conditional distributions given $y$ which we denote by $s(\cdot|y)$. This can be verified by checking that both $x|y$ and $x^+|y$ have the same conditional distribution given $y$ equal to

$$p(\cdot|y) = \int_{x^{ref}} p_{aug}(\cdot|x^{ref}, y)p_{ref}(x^{ref}|y)dx^{ref}$$

where the integrals will be sums in the discrete (probability mass function) setting. Note that although $x, x^+$ are conditionally IID given $(x^{ref}, y)$, they need not be conditionally IID given just $y$.

## C  Additional experiments

We replicated the same experiments conducted in Sec. 5 on CIFAR10. The results are plotted in Fig. 7 and Fig. 8. In contrast to CIFAR10 and CIFAR100 where the numerical results are provided under three different settings, namely, unit-ball normalization with random initialization, unit-ball normalization with Neural-Collapse initialization, and unit-sphere normalization with random initialization, for Tiny-ImageNet, we only conduct experiments under unit-ball normalization with random initialization. This is because the size of the Tiny-ImageNet dataset (120000 images) is much larger than the sizes of both CIFAR10 and CIFAR100 datasets (50000 images per dataset) which results in a significantly longer processing time. The results for Tiny-ImageNet are plotted in Fig. 9.

### C.1  Neural-Collapse and Dimensional-Collapse

For CIFAR10, from Fig. 7 and Fig. 8, we observe similar phenomena as those for CIFAR100. As before, we note that while Theorems 2 and 3 suggest that Neural-Collapse should occur in both the supervised and unsupervised settings when using the random negative sampling method, one may not be able to observe Neural-Collapse in unsupervised settings in practice. For the supervised case in CIFAR10, any degree of hardness propels the representation towards Neural-Collapse. This may be due to the small number of classes in CIFAR10.

---

**Algorithm 1** Contrastive Learning Algorithm

---

**Require:** Batch size $N$, data $\mathcal{X}$, label $\mathcal{Y}$, neural-net parameters of representation function $f$, Algorithm: SCL/ UCL/HSCL/HUCL, normalization type: unit-ball/unit-sphere/no-normalization, the hardening function $\eta(t_{1:k}) := \prod_{i=1}^{k} e^{\beta t_i}, \beta > 0$.
1: Define negative distribution $p^-(z_{1:k}^-|z, z^+)$ based on the chosen Algorithm, see Sec. 3 for details.
2: **for** each sampled minibatch $\{x_i\}_{i=1}^N$ **do**
3:      **for** all $i \in \{1, \ldots, N\}$ **do**
4:         Compute $f(x_i)$
5:         **if** unit-ball normalization **then**
6:            **if** $\|f(x_i)\| \leq 1$ **then**
7:               $z_i = f(x_i)$
8:            **else**
9:               $z_i = \frac{f(x_i)}{\|f(x_i)\|}$
10:            **end if**
11:         **else if** unit-sphere normalization **then**
12:            $z_i = \frac{f(x_i)}{\|f(x_i)\|}$
13:         **else if** no-normalization **then**
14:            $z_i = \frac{f(x_i)}{\sqrt{d}}$
15:         **end if**
16:      **end for**
17:      **for** all $i \in \{1, \ldots, N\}$ **do**
18:         **for** all $j \in \{1, \ldots, N\}$ **do**
19:            **if** $y(x_i) = y(x_j)$ **then**
20:               Draw $\{z_{1:k}^-\}$ from $p^-(z_{1:k}^-|z_i, z_j)$
21:               $\{v_{i,j,m}\}_{m=1}^k = \{z_i^\top z_m^- - z_i^\top z_j\}_{m=1}^k$
22:               $\ell_{i,j} = \log\left(1 + \frac{1}{k}\sum_{m=1}^k e^{v_{i,j,m}}\right)$
23:            **else**
24:               $\ell_{i,j} = 0$
25:            **end if**
26:         **end for**
27:         Compute the average loss of sample $x_i$: $\ell_i = \frac{1}{|\{\ell_{ij} \neq 0\}|}\sum_{j=1}^N \ell_{i,j}$
28:      **end for**
29:      Compute the average loss of minibatch: $L = \frac{1}{N}\sum_{i=1}^N \ell_i$
30:      Take one stochastic gradient step using Adam
31: **end for**
     **return** Encoder network $f(\cdot)$

---

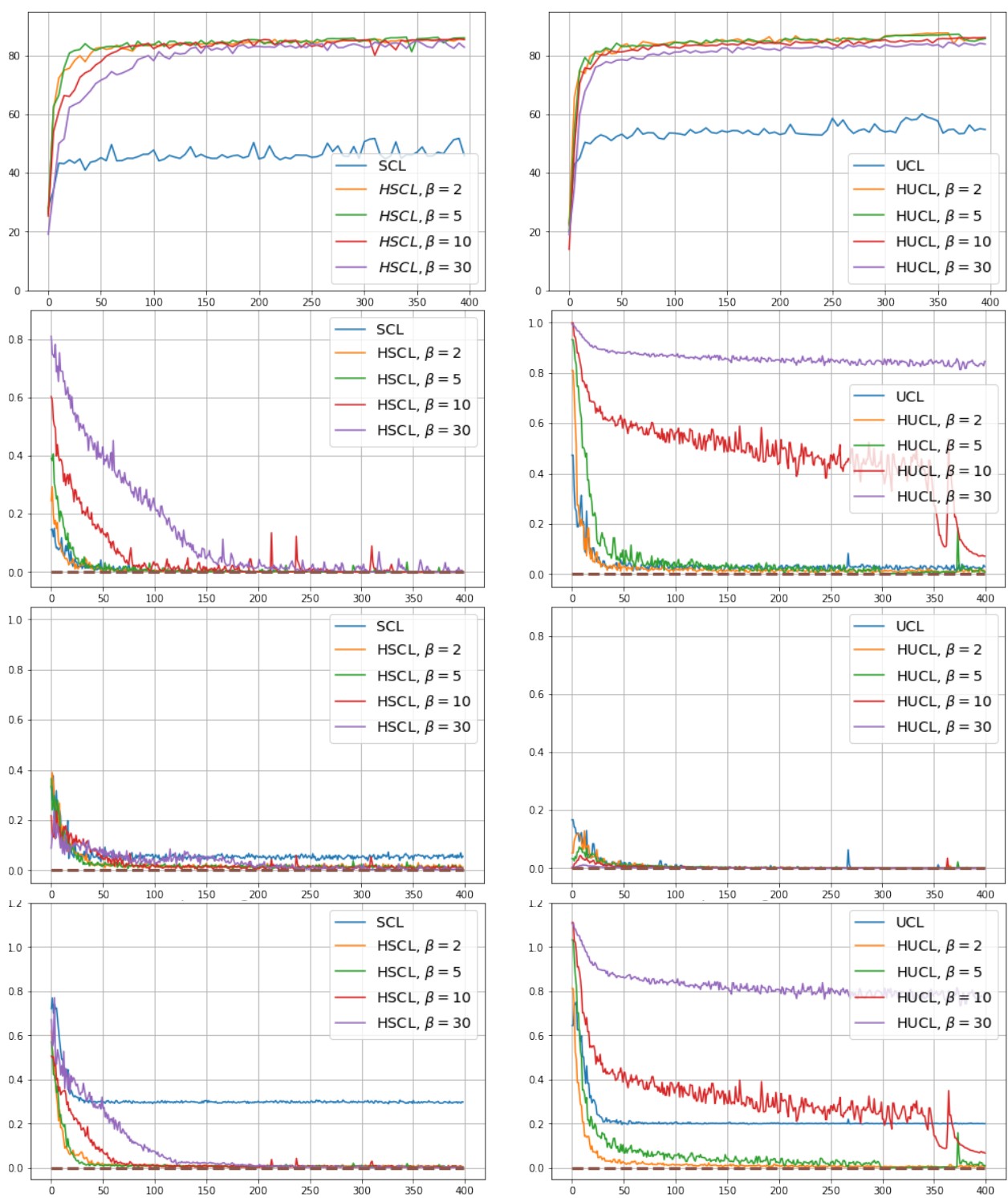

Figure 7: Results for CIFAR10 under supervised settings (SCL, HSCL, left column) and unsupervised settings (UCL, HUCL, right column) with unit-ball normalization and random initialization. From top to bottom: Downstream Test Accuracy, Zero-sum metric, Unit-norm metric, and Equal inner-product metric, all plotted against the number of epochs.

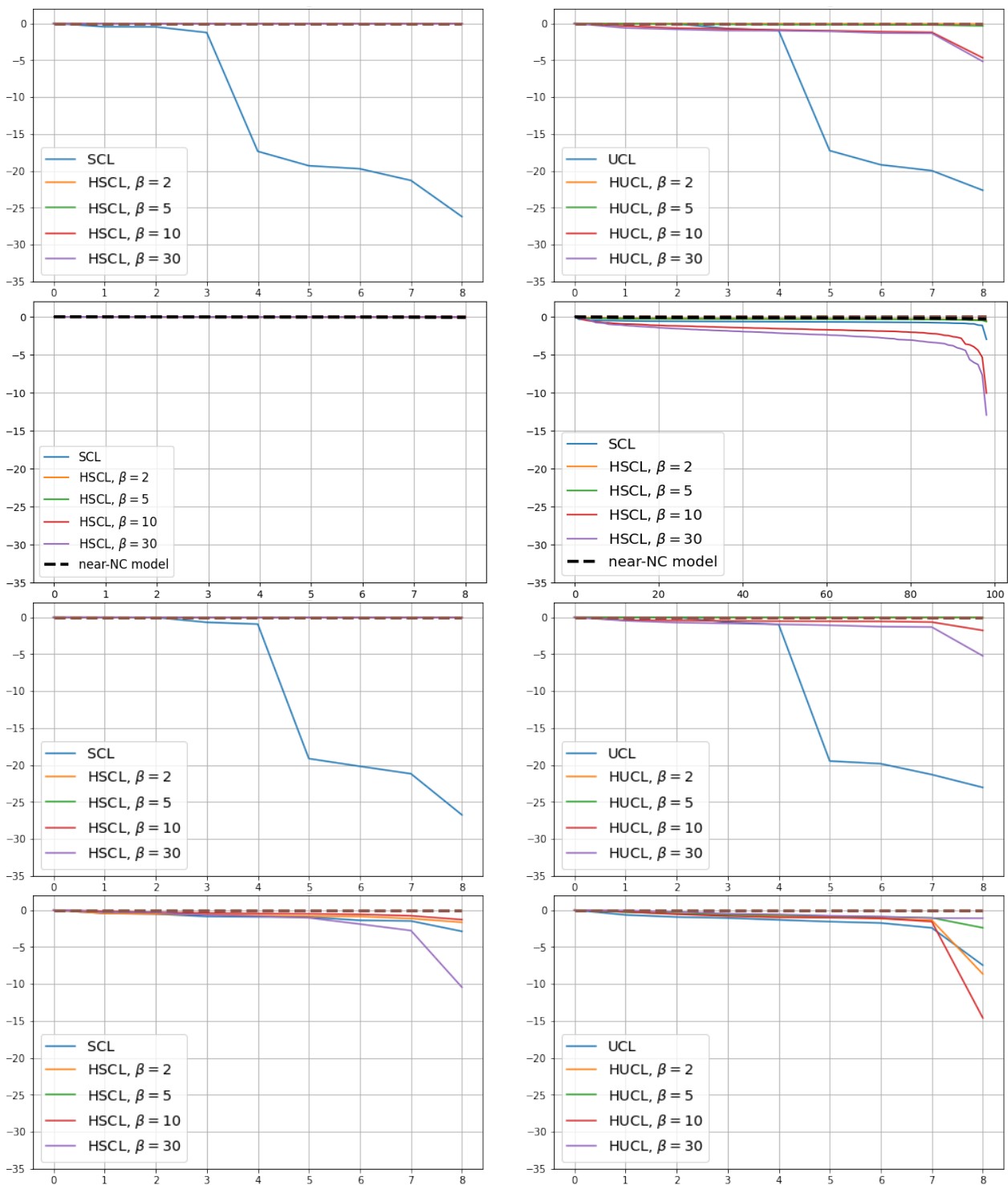

Figure 8: Sorted normalized singular values of the empirical covariance matrix of class means (in representation space) in the last epoch plotted in log-scale for CIFAR10 under supervised (left column) and unsupervised (right column) settings. From top to bottom: Unit-ball normalization with random initialization, Unit-ball normalization with near-NC initialization, Unit-sphere normalization with random initialization, and unnormalized representation with random initialization.

For Tiny-ImageNet, from Fig. 9, we observe that when $\beta = 5, 10, 30$, the geometry of the learned representation closely aligns with that of Neural-Collapse. However, in HUCL, a high degree of hardness can be harmful. At $\beta = 5$, the geometry most closely approximates Neural-Collapse for both CIFAR10 and Tiny-ImageNet. However, increasing the degree of hardness further, for example at $\beta = 30$, causes the center of class means to deviate from the origin and the equal inner-product condition is heavily violated.

Furthermore, from the the normalized singular values for CIFAR10 in Fig. 8 and for Tiny-ImageNet in Fig. 9, we observe that random negative sampling without any hardening ($\beta = 0$) suffers from DC whereas hard-negative sampling consistently mitigates DC. The supervised case benefits more from a higher degree of hardness, since in the unsupervised cases there are higher chances of (latent) class collisions.

## C.2 Effects of initialization and normalization

The normalized singular value plots for CIFAR10 are shown in Fig. 8. Compared to CIFAR 100 (see Fig. 4), the phenomenon of DC in CIFAR10 is far less pronounced. This may be because CIFAR10 has a smaller number of classes compared to CIFAR100 (10 *vs.* 100). However, the effects of initialization and normalization on the learned representation geometry are similar to that for CIFAR100:

**Effects of initialization:** 1) SCL and HSCL trained with near-NC initialization and Adam do not exhibit DC and 2) UCL trained with near-NC initialization and Adam also does not exhibit DC, but the behavior of HUCL depends on the hardness level $\beta$.

**Effects of normalization:** The behavior of unit-sphere normalization is close to that of unit-ball normalization, and with hard-negative sampling (and suitable hardness levels), both SCL and UCL can achieve NC. Without normalization, neither regular nor hard-negative training methods attain NC and they suffer from DC. We also observe that with random-negative sampling, un-normalized representations lead to reduced DC in both SCL and UCL. However, hard-negative sampling benefits more from feature normalization and its absence leads to more severe DC.

## C.3 Experiments with different batch sizes

To investigate the effect of batch size on the outcomes, we conducted experiments with varying batch sizes. All previous experiments were performed with a batch size of 512. In this section, we present results for batch sizes of 64 and 32 in Fig. 10 and Fig. 11, respectively.

We observe that when the batch size is reduced to 64, Neural-Collapse is still nearly achieved in both HSCL ($\beta = 5, 10, 30$) and HUCL ($\beta = 5$). However, with a further reduction of batch size to 32, Neural-Collapse is only achieved in HSCL ($\beta = 30$), and it fails to occur in HUCL for any value of $\beta$.

## C.4 Experiments with a different hardening function

To explore whether similar results can be achieved with a different hardening function, we investigated the impact that changing the hardening function has on NC and DC under unit-ball normalization. We conducted experiments using the CIFAR10 and CIFAR100 datasets adopting the setup consistent with previous experiments but used a new family of hardening functions having the following polynomial form: $\eta(t_{1:k}) := \prod_{i=1}^{k} (\max\{t_i + 1, 0\})^\epsilon, \epsilon > 0$, for the following set of hardness values $\epsilon = 3, 5, 10, 20$. We note that this family of hardening functions decays at a significantly slower rate compared to the exponential hardening function we used in all our previous experiments.

Results for CIFAR100 and CIFAR10 are plotted in Figs. 12 and 13, respectively. We observe phenomena similar to those in Figs. 3 and 7. By selecting an appropriate hardening parameter $\epsilon$, we can achieve, or nearly achieve, NC in both supervised and unsupervised settings. Consequently, we can draw conclusions that are qualitatively similar to those in Sec. C.1. Specifically, hard-negative sampling in both supervised and unsupervised settings can mitigate DC while achieving NC.

## C.5 Experiments with the SimCLR framework

We conducted additional experiments using the state-of-the-art SimCLR framework for Contrastive Learning. Instead of sampling positive samples from the same class directly, we follow the setting in SimCLR which uses two independent augmentations of a reference sample to create the positive pair. No label information is used to generate the anchor-positive pair.

Figure 14 shows results for SimCLR sampling using the loss function proposed in SimCLR which is the large-$k$ asymptotic form of the InfoNCE loss:

$$L^{SimCLR}(f) = \mathbb{E}_{p(x,x^+)}\left[\log\left(1 + \frac{Q\,\mathbb{E}_{p(x^-|x)}[e^{f(x)^\top f(x^-)}]}{e^{f(x)^\top f(x^+)}}\right)\right] \tag{57}$$

where $Q$ is a weighting hyper-parameter that is set to batch size minus two. Compared to our previous experiments, all the NC metrics deviate significantly away from zero in both supervised and unsupervised settings and all hardness levels. Still, the high-level conclusions for DC are qualitatively similar to those from our previous experiments, specifically that hard-negative sampling can mitigate DC in both supervised and unsupervised settings.

Figure 15 shows results for SimCLR sampling using the InfoNCE loss with $k = 256$ and a positive distribution that only relies on the augmentation method. The results are very similar to those in Fig. 14. Since the results in Figs. 14 and 15 share the same SimCLR sampling framework but different losses, it follows that the failure to attain NC is not due to the particular loss function used, but the SimCLR sampling framework itself which does not utilize label information to generate samples. From the unit-norm metric plots in both figures it is clear that the final representations are mostly distributed *within* the unit-ball than on its surface.

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

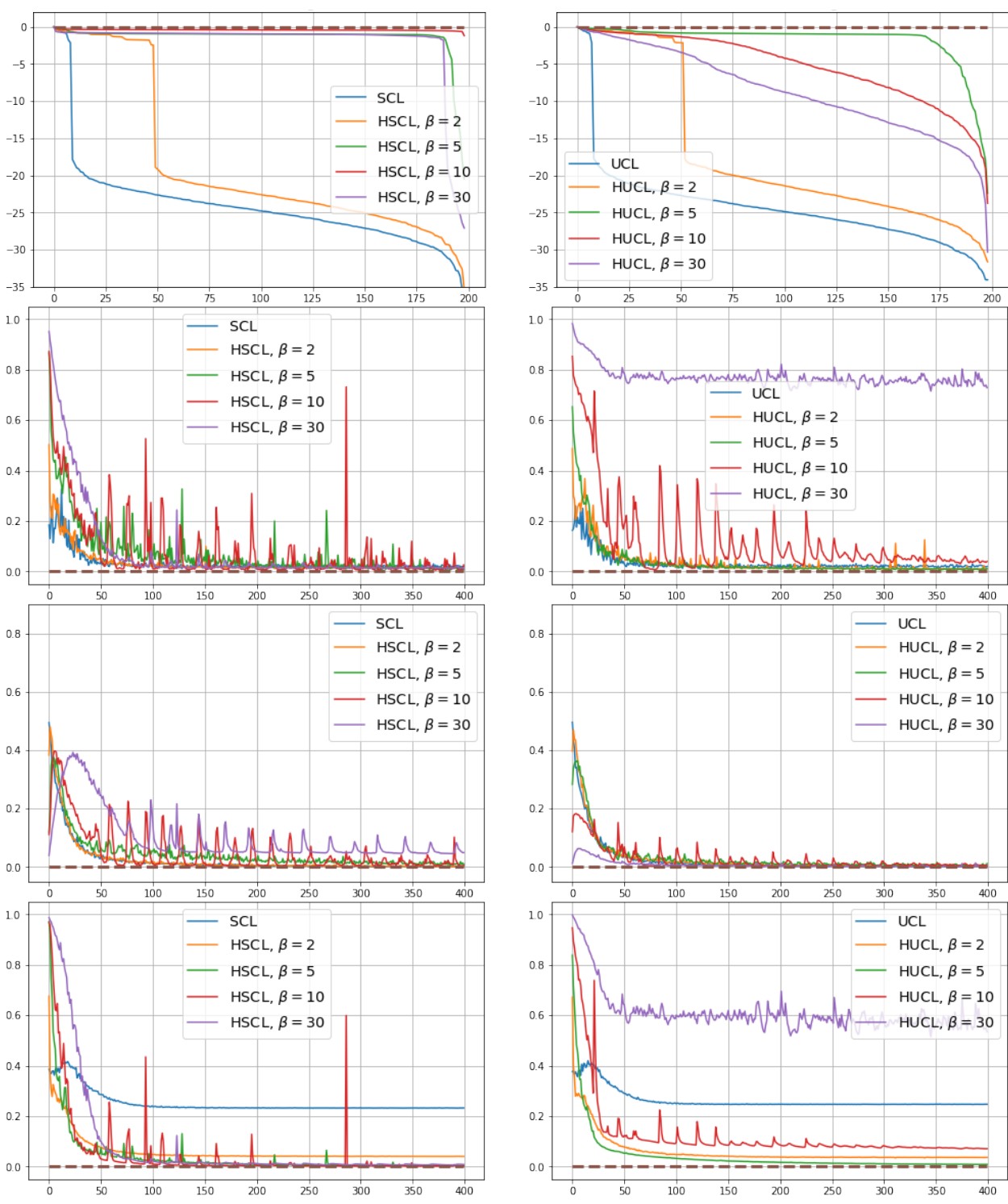

Figure 9: Results for Tiny-ImageNet under supervised settings (SCL, HSCL, left column) and unsupervised settings (UCL, HUCL, right column) with unit-ball normalization and random initialization. Top row: sorted normalized singular values of the empirical covariance matrix of class means (in representation space) in the last epoch plotted in log-scale. Rows 2–4: Zero-sum metric, Unit-norm metric, and Equal inner-product metric, all plotted against the number of epochs.

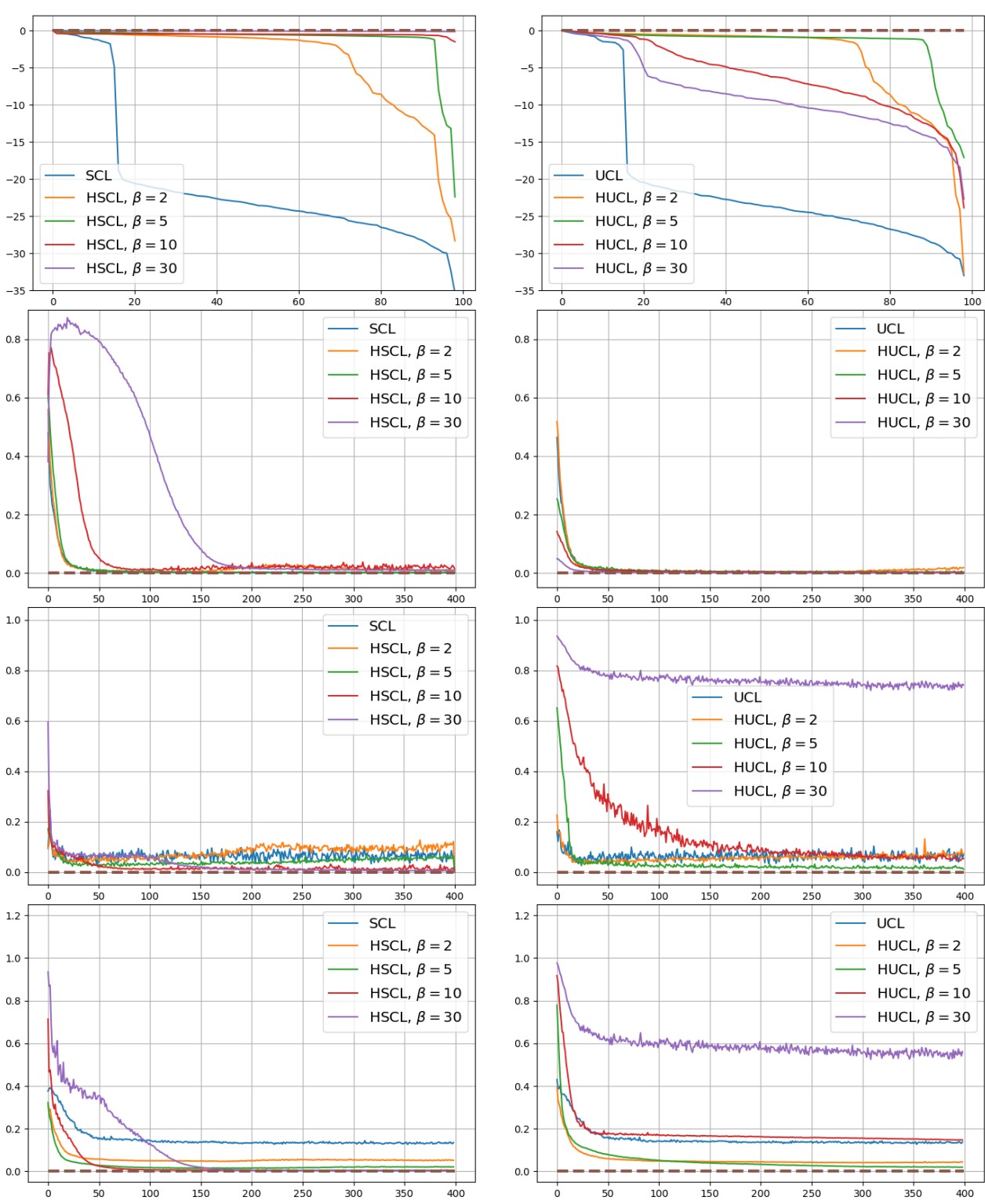

Figure 10: Results for CIFAR100 under supervised settings (SCL, HSCL, left column) and unsupervised settings (UCL, HUCL, right column) with unit-ball normalization and random initialization when batch size is equal to 64. Top row: sorted normalized singular values of the empirical covariance matrix of class means (in representation space) in the last epoch plotted in log-scale. Rows 2–4: Zero-sum metric, Unit-norm metric, and Equal inner-product metric, all plotted against the number of epochs.

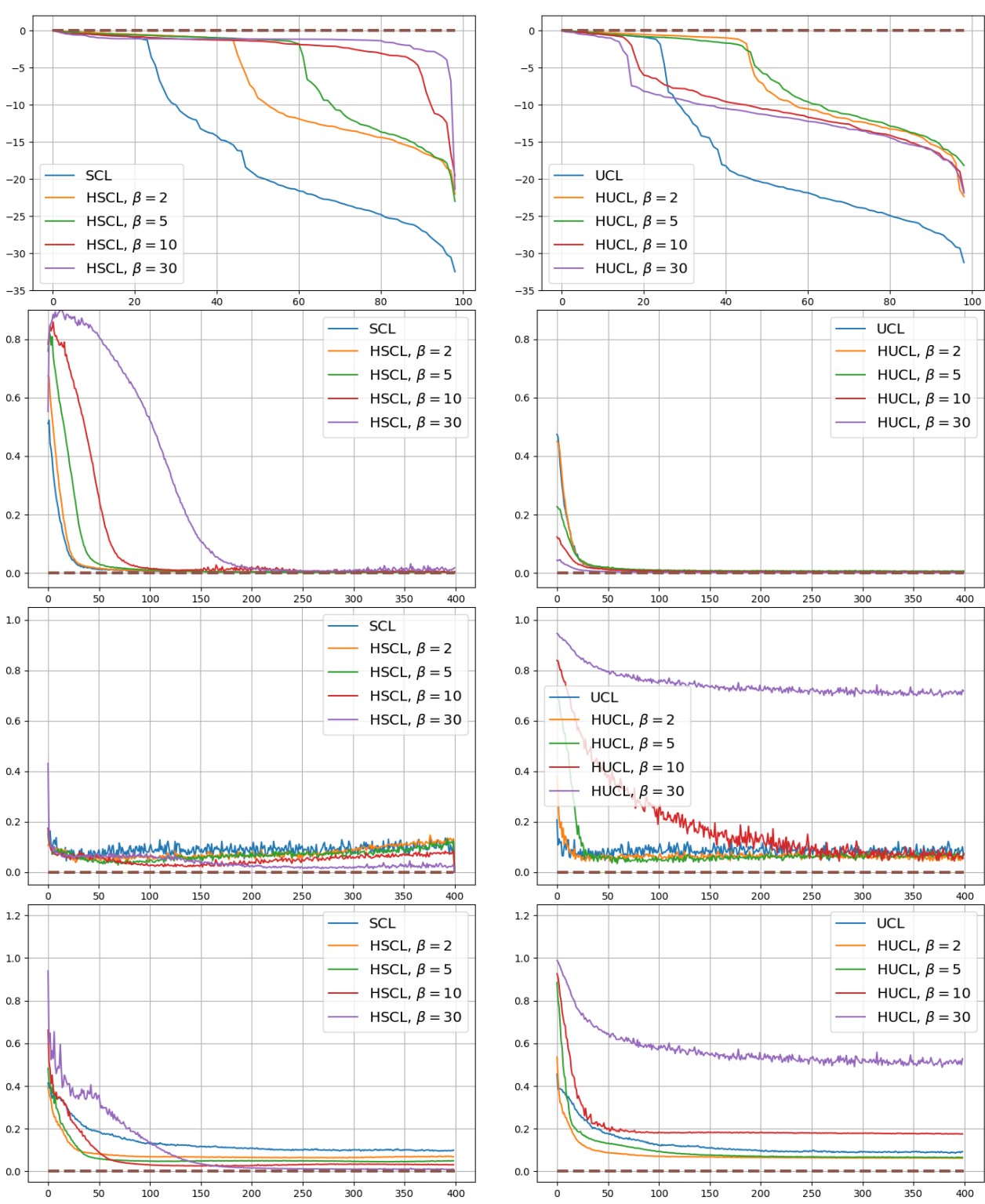

Figure 11: Results for CIFAR100 under supervised settings (SCL, HSCL, left column) and unsupervised settings (UCL, HUCL, right column) with unit-ball normalization and random initialization when batch size is equal to 32. Top row: sorted normalized singular values of the empirical covariance matrix of class means (in representation space) in the last epoch plotted in log-scale. Rows 2–4: Zero-sum metric, Unit-norm metric, and Equal inner-product metric, all plotted against the number of epochs.

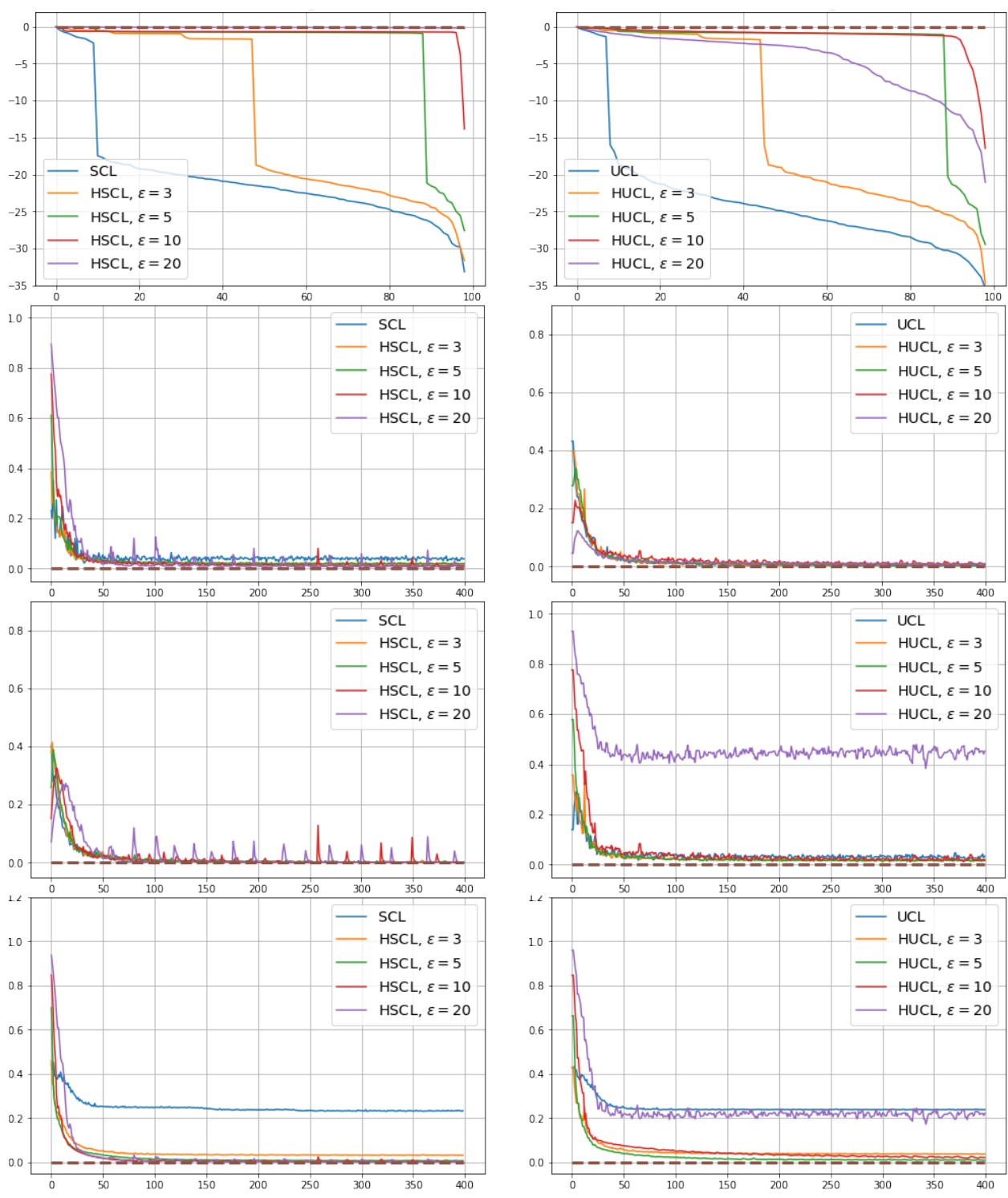

Figure 12: Results for CIFAR100 with polynomial hardening function, under supervised settings (SCL, HSCL, left column) and unsupervised settings (UCL, HUCL, right column) with unit-ball normalization and random initialization. Top row: sorted normalized singular values of the empirical covariance matrix of class means (in representation space) in the last epoch plotted in log-scale. Rows 2–4: Zero-sum metric, Unit-norm metric, and Equal inner-product metric, all plotted against the number of epochs.

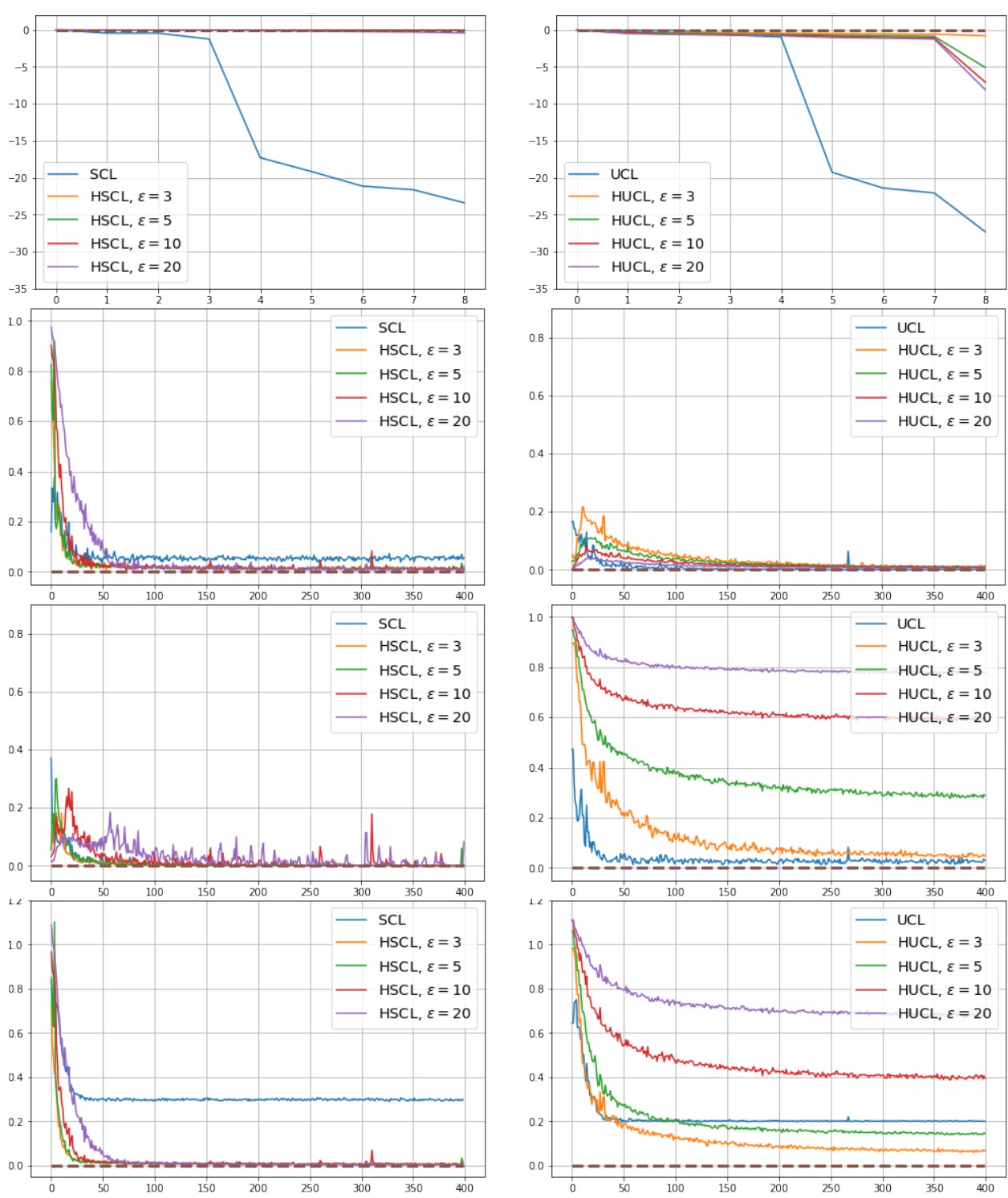

Figure 13: Results for CIFAR10 with polynomial hardening function, under supervised settings (SCL, HSCL, left column) and unsupervised settings (UCL, HUCL, right column) with unit-ball normalization and random initialization. Top row: sorted normalized singular values of the empirical covariance matrix of class means (in representation space) in the last epoch plotted in log-scale. Rows 2–4: Zero-sum metric, Unit-norm metric, and Equal inner-product metric, all plotted against the number of epochs.

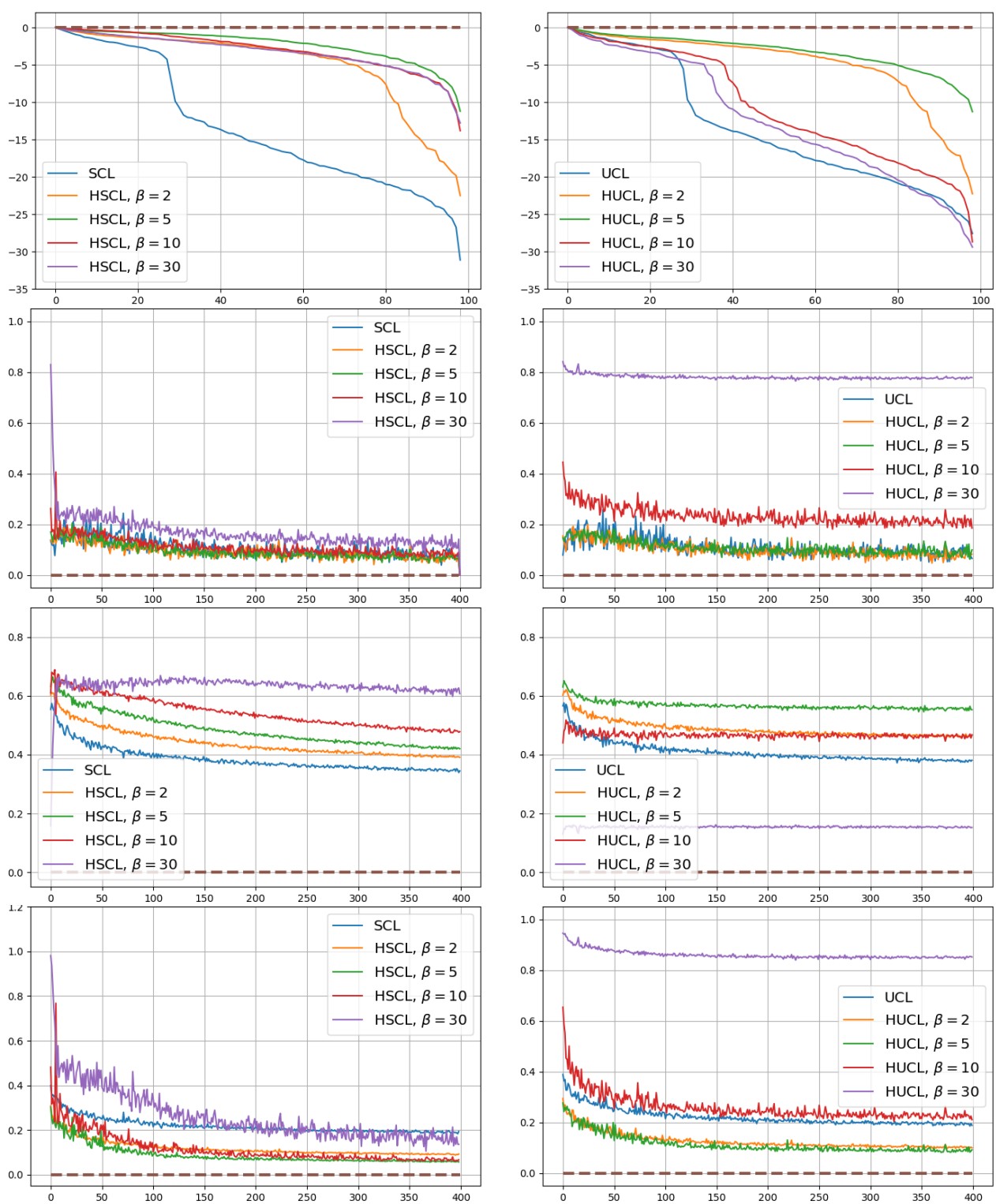

Figure 14: Results for CIFAR100 with SimCLR sampling with SimCLR loss (Eq.44), under supervised settings (SCL, HSCL, left column) and unsupervised settings (UCL, HUCL, right column) with unit-ball normalization and random initialization. Top row: sorted normalized singular values of the empirical covariance matrix of class means (in representation space) in the last epoch plotted in log-scale. Rows 2–4: Zero-sum metric, Unit-norm metric, and Equal inner-product metric, all plotted against the number of epochs.

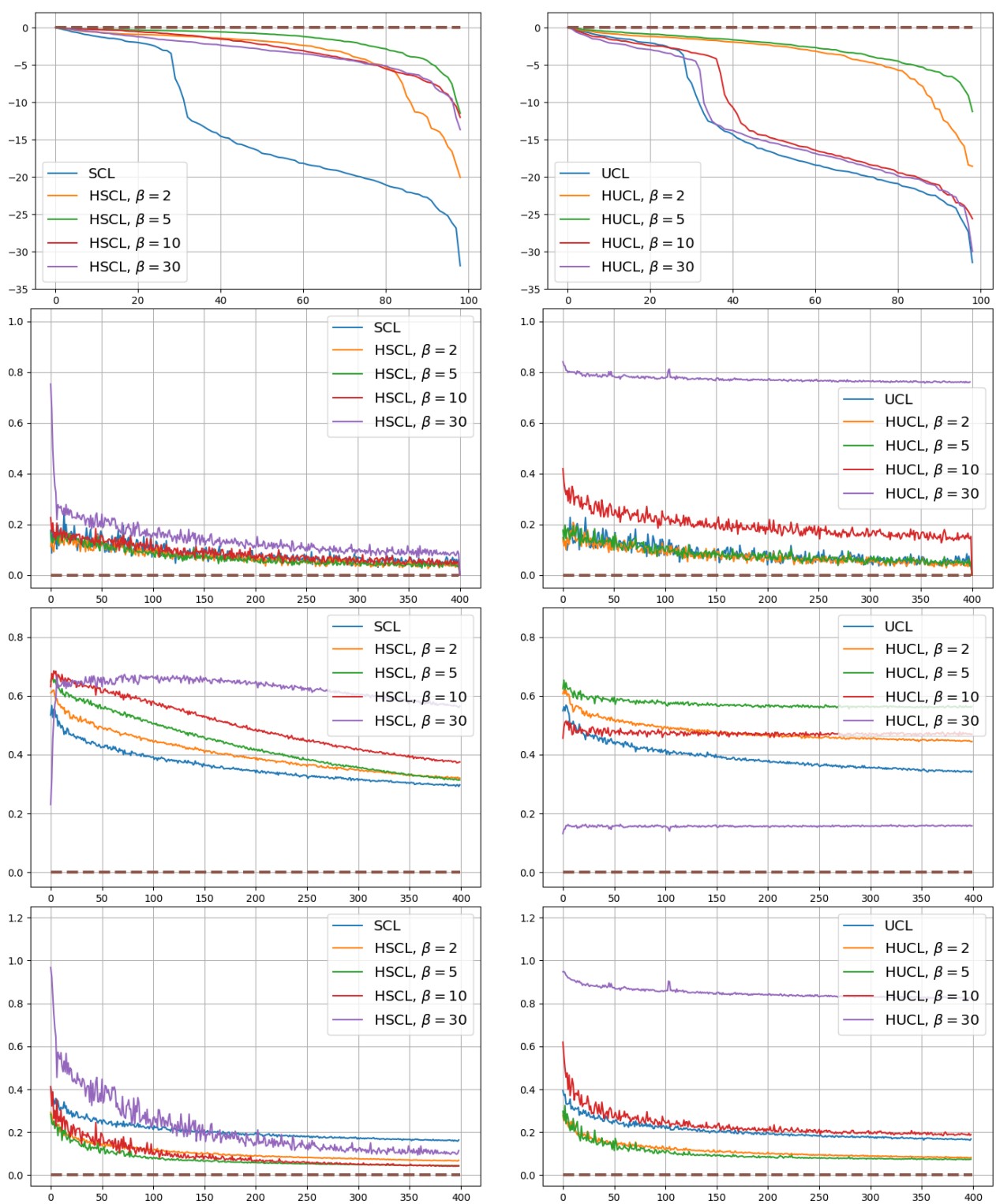

Figure 15: Results for CIFAR100 with SimCLR sampling using InfoNCE loss, under supervised settings (SCL, HSCL, left column) and unsupervised settings (UCL, HUCL, right column) with unit-ball normalization and random initialization. Top row: sorted normalized singular values of the empirical covariance matrix of class means (in representation space) in the last epoch plotted in log-scale. Rows 2–4: Zero-sum metric, Unit-norm metric, and Equal inner-product metric, all plotted against the number of epochs.

