# OpenReview forum: "Hard-Negative Sampling for Contrastive Learning: Optimal Representation Geometry and Neural- vs Dimensional-Collapse"
_TMLR — Accepted by TMLR_

### Review · Reviewer_C1Bz · 2024-12-13

**Summary Of Contributions:**

The paper studies the geometry of the loss landscape of contrastive learning with normalized features in both supervised and unsupervised setting as well as both hard negative sampling and standard negative sampling setting. On the theoretical front, authors first show that hard negative sampling is always at least as hard as the standard negative sampling in terms of the loss. Then they show, both in supervised and unsupervised setting that the loss of CL is lower-bounded by a value that is achieved by neural collapse. In case of strictly increasing and convex loss function neural collapse is the only solution achieving that loss. In SCL, the authors also show equivalence with the HSCL loss. The authors lift the class-conditional independence assumption from related work in their SCL analysis. On the empirical front, the authors first empirically back up their theoretical findings on synthetic dataset. Then, they analyze the convergence to NC solutions of CL in practice and show that the models sometimes exhibit dimensional collapse, which, however, can be partially avoided/remedied by using the hard negative sampling. They also study the effect of several design parameters and hyper-parameters on the emergence of NC and/or DC in CL.

**Audience:**

Yes

**Broader Impact Concerns:**

There are no broader impact concerns.

**Claims And Evidence:**

Yes

**Requested Changes:**

**Questions:**

-	In the paper you claim that the number of latent classes in UCL depends on the sampling procedure of the similar samples. How?

-	How is the sampling in hard negative sampling performed in practice? It seems to be computationally expensive to always evaluate the network on all the samples.

-	How does the model decide how many latent classes will there be in UCL and will it influence the optimal loss function?

-	In page 14 you write that you sample the negative samples from the entire batch and potentially including the anchors, positive examples and/or same-class examples. Doesn’t this undermine the difference between SCL and UCL and make all experiments effectively UCL? This question is further motivated by what you write on page 16 where you say that the NC is not exactly achieved when you use this procedure. Why using this procedure in the first place then?

-	Do you use weight decay in your practical experiments?

**Requested changes:**

-	Please add a detailed discussion on the effect of the number of latent classes on the loss in UCL and what does that mean for the incentives of the model and whether or not it introduces some biases, e.g. towards small or large number of latent classes.

-	Please add the discussion in the beginning of section 4.3 about the usage of your data distribution and loss as empirical distribution and empirical loss already to the point where you introduce those concepts. Furthermore, please always contextualize each theorem and argument to whether it refers to empirical or theoretical loss or both, to make the discussion more transparent.

-	Fix the typo in the beginning of page 14 where you refer to property “(d)” but it should be “(e)”.

**Strengths And Weaknesses:**

**Strengths:**

- The paper provides a very nice and neat unifying theory for both SCL and UCL and include the analysis for hard negative sampling. This can be regarded as a strong unified theory on neural collapse in contrastive learning.

- The experimental evaluation is structured and convincing and nicely complements the theoretical findings while also inviting a further research on the topic (such as deeper explanation on why hard negative sampling helps preventing the dimensional collapse in some cases).

- The paper is clearly written and easy to understand in most cases.

**Weaknesses:**

- The theoretical contribution, while unifying and structured, is not particularly novel. Theorems 1 and 3 are already partially covered by the related work with only minor improvements. The statement of theorem 2 is novel (to the best of my knowledge), however the proof techniques of both theorem 2 and 3 (such as applying the Jensen inequality to obtain the optimality of equidistant configurations) represent a rather routine way of proving NC both in CL, but also in standard training, because the considered loss can be understood as an easier version of the unconstrained features model used to explain NC in standard training (see for instance [1,2, 3]).

- Another issue is that the authors do not lift the conditional independence assumption in the UCL, which in my opinion is literally impossible to satisfy in practice.

- On the experimental front, I think it is not insightful that the hard negative sampling helps improve collapse because in this setting to speak about collapse is the same as to speak about training loss and thus if the related literature already established that hard negative sampling improves the training loss and/or the overall performance of the networks, this empirical observation doesn't bring any new insight.

- It is not entirely clear whether the the number of latent classes in UCL influences the loss function and if yes, what does the model do about it? In particular, it is the choice of the model how many latent classes will there be as long as it is at least the number of true classes. Won’t this produce some degenerate behavior form the model?

[1] Zhu, Zhihui, et al. "A geometric analysis of neural collapse with unconstrained features." Advances in Neural Information Processing Systems 34 (2021): 29820-29834.

[2] Zhou, Jinxin, et al. "Are all losses created equal: A neural collapse perspective." Advances in Neural Information Processing Systems 35 (2022): 31697-31710.

[3] Ji, Wenlong, et al. "An unconstrained layer-peeled perspective on neural collapse." arXiv preprint arXiv:2110.02796 (2021).

---

> ### Author Response · Authors · 2025-01-08
> **Author response to Reviewer C1Bz**
>
> **Comment**: The theoretical contribution, while unifying and structured, is not particularly novel. Theorems 1 and 3 are already partially covered by the related work with only minor improvements. The statement of theorem 2 is novel (to the best of my knowledge), however the proof techniques of both theorem 2 and 3 (such as applying the Jensen inequality to obtain the optimality of equidistant configurations) represent a rather routine way of proving NC both in CL, but also in standard training, because the considered loss can be understood as an easier version of the unconstrained features model used to explain NC in standard training (see for instance [1,2, 3]).
>
> *Response*: Regarding novelty over existing works we urge the reviewer to please go over the related work sections again. In particular, for SCL, we make use of label information for negative sampling as well, whereas existing works do not.
> Regarding the novelty of Theorem 1: We agree that a version of Theorem 1 was explored in [Theorem 3.1,Wu et al. (2020)]. However, the assumption in Wu et al. (2020) is pretty restricted. They assume a threshold on the exponent of the  inner product of positive and negative samples and their theorem only holds for a specific loss function. This restricted assumption immediately implied their claimed lower bound. In contrast, our Theorem 1 for UCL as well as SCL is more general and we avoid making any such assumption by creatively utilizing a comparison inequality, namely Harris’ inequality.
>
>
> Regarding the novelty of  Theorem 3, we believe that NC in UCL has not been shown yet in the literature.
> Theorem 2 may be regarded as partially novel in that NC for SCL has been discovered in previous work. However, there is an important difference between the SCL setting that we consider vs. related work, namely that we use label information for negative samples as well whereas previous work on NC for SCL do not. Furthermore, we show an additional novel result that NC also occurs in HSCL.
> Regarding the use of Jensen’s inequality: Yes, use of Jensen’s inequality is a standard and really indispensable technique in the analysis of convex functions. In the context of CL, the main difference from existing works is that here we utilized Jensen’s inequality for a generalized CL loss which includes the InfoNCE loss and the cross-entropy loss in the works referenced by the reviewer.
> In fact, we consider it is a strength (rather than a weakness) of our work that we can establish results of great generality using simple, well-known analytical tools.
>
>
> **Comment**: Another issue is that the authors do not lift the conditional independence assumption in the UCL, which in my opinion is literally impossible to satisfy in practice.
>
> *Response*: This seems to suggest that there should exist a version of Theorem 3 in which Neural Collapse (NC) will occur in UCL without assuming that the anchor and positive samples are conditionally independent given their common (latent) label. But perhaps this may be an impossible expectation? We certainly tried to and did not succeed. Moreover, empirically we did not observe NC in our SimCLR experiments where the conditional independence assumption is not satisfied.

---

> > ### Author Response · Authors · 2025-01-08
> > **Author response to Reviewer C1Bz (continue)**
> >
> > **Comment**: On the experimental front, I think it is not insightful that the hard negative sampling helps improve collapse because in this setting to speak about collapse is the same as to speak about training loss and thus if the related literature already established that hard negative sampling improves the training loss and/or the overall performance of the networks, this empirical observation doesn't bring any new insight.
> >
> > *Response*: The two losses, one using hard negatives and the other without, are very different and hence it does not make sense to claim that hard negative sampling improves the training loss. We do not make this claim and we believe that related literature also does not make this claim. Indeed, use of hard negative sampling is shown to learn better representations evidenced via its downstream performance (test accuracy).
> >
> > We want to emphasize that observing NC on the training data does NOT necessarily mean better performance in downstream tasks. Indeed, we do not claim NC can improve the learning performance and generalization ability of trained models. The experiments show that using hard-negative sampling helps attain NC when using iterative methods starting at a random initialization. A more important observation that the paper highlights is that even though hard-negative sampling may not completely aid the attainment of the NC solution (which is the optimal solution minimizing the training loss), it can prevent or mitigate dimension collapse (DC) and this is a novel insight. There is some empirical evidence that preventing DC may lead to better downstream performance as cited below:
> >
> > *Hassanpour, J., Srivastav, V., Mutter, D. and Padoy, N., 2024. Overcoming Dimensional Collapse in Self-supervised Contrastive Learning for Medical Image Segmentation. arXiv preprint arXiv:2402.14611.*
> >
> > *Jing, L., Vincent, P., LeCun, Y. and Tian, Y., 2021. Understanding dimensional collapse in contrastive self-supervised learning. arXiv preprint arXiv:2110.09348.*
> >
> > **Comment**: It is not entirely clear whether the the number of latent classes in UCL influences the loss function and if yes, what does the model do about it? In particular, it is the choice of the model how many latent classes will there be as long as it is at least the number of true classes. Won’t this produce some degenerate behavior form the model?
> >
> > *Response*: The lower bound of the UCL loss is shown to depend on the number of latent classes and therefore the number of latent classes will affect this minimum value.
> >
> > The number of latent classes in UCL is an unknown intrinsic property of the dataset and is not a tunable parameter of the neural network model used to learn a mapping from data space to representation space. We can, however, choose the dimension of the representation space of the model, and if we choose it to be large enough compared to the unknown number of latent classes, we may observe NC in UCL. If the dimension of the representation space is chosen to be too small, then the model may exhibit some degenerate behavior, e.g. dimension collapse. This aspect is hard to analyze theoretically.
> >
> > **Comment**: In the paper you claim that the number of latent classes in UCL depends on the sampling procedure of the similar samples. How?
> >
> > *Response*: The sampling procedure in UCL and the number of latent classes are inter-related in equation (4), Sec. 3.1 and the following paragraphs where we describe the latent model(s) in full detail:
> >
> > “Latent labels in UCL can be interpreted as indexing latent clusters. Suppose that there are $C$ latent clusters from which the anchor, positive, and the $k$ negative samples can be drawn from. Then the joint distribution of all samples and their latent labels can be described by the following equation:
> >
> > *Equation 4* - (we tried but could not successfully typed the equation 4 here)
> >
> > where $\lambda$ is the marginal distribution of the anchor's latent label, $r(\cdot)$ the marginal distribution of the latent labels of negative samples, and  $s(x^-|y^-)$ is the conditional probability distribution of any negative sample $x^-$ given its latent label $y^-$.”
> >
> > **Comment**: How is the sampling in hard negative sampling performed in practice? It seems to be computationally expensive to always evaluate the network on all the samples.
> >
> > *Response*: For a given batch (say of size 256), the hard negative samples are chosen from this batch, i.e. we treat the batch as data. This is not computationally expensive at all. This was described in Sec. 4.3 and the two paragraphs before Sec. 5.1.

---

> ### Author Response · Authors · 2025-01-08
> **Author response to Reviewer C1Bz (continue)**
>
> **Comment**: How does the model decide how many latent classes will there be in UCL and will it influence the optimal loss function?
>
> *Response*: As explained above, the number of latent classes is a property of the dataset and it is not the choice of our neural network model. The model, however, sets the dimension of the representation space. If the dimension of the representation space is large enough compared to the number of latent classes then we may observe NC.
>
> **Comment**: In page 14 you write that you sample the negative samples from the entire batch and potentially including the anchors, positive examples and/or same-class examples. Doesn’t this undermine the difference between SCL and UCL and make all experiments effectively UCL? This question is further motivated by what you write on page 16 where you say that the NC is not exactly achieved when you use this procedure. Why using this procedure in the first place then?
>
> *Response*: Thank you for catching this. We indeed made a typo in our writing. We only applied these settings (sampling the negative samples from the entire batch and potentially including the anchors, positive examples and/or same-class examples) for the SimCLR framework. We have revised the draft accordingly.
>
> **Comment**: Do you use weight decay in your practical experiments?
>
> *Response*: No, we do not use weight decay in our experiments.
>
> **Comment**: Please add a detailed discussion on the effect of the number of latent classes on the loss in UCL and what does that mean for the incentives of the model and whether or not it introduces some biases, e.g. towards small or large number of latent classes.
>
> *Response*: It is hard to precisely quantify the effect of the number of latent classes on the loss in UCL. Since the lower bound which is the optimal value of the loss depends on the number of latent classes, obviously, changing the value of the number of latent classes will change the optimal value of the loss. Indeed, from the lower bound, the larger the number of classes, the higher the  value of the lower bound. Also, if the number of classes is more than the dimension of the latent space plus one, then ETF is no longer the optimal solution, and the lower bound in our Theorem 3 is not achievable. In this case, we may need other approaches  based on convex optimization as described in [Thuan et al. (2024)] to answer this question.
>
>
> **Comment**: Please add the discussion in the beginning of section 4.3 about the usage of your data distribution and loss as empirical distribution and empirical loss already to the point where you introduce those concepts. Furthermore, please always contextualize each theorem and argument to whether it refers to empirical or theoretical loss or both, to make the discussion more transparent.
>
> *Response*: All results prior to Sec.4.3 pertain to the theoretical loss, not a practical loss with batch-size. In practice, due to the limited number of samples that can be handled in one batch, the practical loss is different and is sample-based loss. The purpose of Sec.4.3 is to demonstrate that results that were established in previous sections for the general theoretical loss also hold for the empirical and batched empirical losses. We have revised the first para of Sec.4.3 to make this point transparent.
>
>
> **Comment**: Fix the typo in the beginning of page 14 where you refer to property “(d)” but it should be “(e)”.
>
> *Response*: Thank you for catching this. We fixed this typo in the revision version.
>
> *All the changes in our revision version are highlighted in blue text.*
>
> *Thank you very much for your feedback!*
>
> *Sincerely,*
>
> *Authors,*

---

> > ### Comment · Reviewer_C1Bz · 2025-01-15
> > **Thank you for your response.**
> >
> > **Discussion point:** Novelty of the theoretical contribution.
> >
> > *Response:* Thank you for a detailed answer. I accept your answer about Theorem 1. Regarding Theorems 2&3, I personally consider the results by [1] as a form of UCL analysis (precisely for the reasons you outline about not using the negative label information), thus I would say that your Theorem 2 is indeed novel while the Theorem 3 is partially covered by [1]. Either way, one of the theorems is novel while the other is partially covered. Regarding the proof, yes, generalizing the analysis to as general of a loss as possible can be considered novelty, I was just remarking that on the technical level, the proof follows already existing proof patterns in this field, but I agree that overall, this level of abstractness is novel.
> >
> > **Discussion point:** Not lifting the conditional independence assumption in the UCL.
> >
> > *Response:* What I tried to emphasize by this comment is that I don’t see a way how to make the anchor and the positive sample independent in UCL in practice. Can you please elaborate how do you sample the anchor; positive sample pair in your UCL experiments to make them independent?
> >
> > **Discussion point:** All the discussion regarding the number of latent classes in UCL, in particular the authors’ claim that “The number of latent classes in UCL is an unknown intrinsic property of the dataset and is not a tunable parameter of the neural network model used to learn a mapping from data space to representation space.”
> >
> > *Response:* I disagree with your claim that the number of latent classes is an intrinsic property of the dataset and is not influenced by the model. The latent classes are created by the model’s representation as clusters in the representation/feature space. The model could collapse all the samples into a single vector, thus creating one latent class, or to the other extreme, could represent each training sample differently, thus effectively assigning each training example a different latent class. The number of latent classes is a variable that has to be carefully tuned by the optimization algorithm so that it well agrees with the number of true classes or the number of distinctive feature combinations.
> > 	Let me demonstrate this on an example: assume we have ten animal species and in each sample, two different animals are shown, one in foreground, one in background. Since there are no labels, it is fully up to the model whether it decides to represent the training data in 10 latent classes (the foreground animal), 45 latent classes (animal pairs), 90 latent classes (ordered animal pairs) or even a different number (like the number of training samples or, less than 10).
> > Based on this, I ask the authors to either clarify what exactly they mean by latent classes in UCL, using, perhaps, my example from above; or correct and reformulate all the occurrences of this statement, whilst still including the related discussion I asked for in my review.
> >
> > **Discussion point:** “Indeed, from the lower bound, the larger the number of classes, the higher the value of the lower bound.”
> >
> > *Response:* How do you see this? Intuitively there are two opposing factors – on one hand the more clusters, the smaller the probability of the negative sample being from the same cluster as the positive pair, but on the other hand, the more clusters the smaller margin.
> >
> > **Discussion point:** Including the discussion about empirical and theoretical losses.
> >
> > *Response:* I still think these remarks should not only be in the beginning of section 4.3 (in fact I didn’t have any problem with that paragraph), but also in Section 3.1.
> >
> > **Discussion point:** Finding out that hard negative sampling improves collapse is not insightful…
> >
> > *Response:* Thank you for clarifying this, I agree with you now.
> >
> > [1] Graf, Florian, et al. "Dissecting supervised contrastive learning." International Conference on Machine Learning. PMLR, 2021.

---

> > > ### Author Response · Authors · 2025-01-22
> > > **Author response to Reviewer C1Bz**
> > >
> > > **Discussion point:** Novelty of the theoretical contribution.
> > >
> > > **Comment:** Thank you for a detailed answer. I accept your answer about Theorem 1 ... but I agree that overall, this level of abstractness is novel.
> > >
> > > *Response:* We appreciate the reassessment of the novelty by the reviewer.
> > >
> > > **Discussion point:** Not lifting the conditional independence assumption in the UCL.
> > >
> > > **Comment:** What I tried to emphasize by this comment is that I don’t see a way how to make the anchor and the positive sample independent in UCL in practice. Can you please elaborate how do you sample the anchor; positive sample pair in your UCL experiments to make them independent?
> > >
> > > *Response*: We too are not aware of any practical method which can guarantee conditional independence of the anchor and positive samples given the (unobserved) latent label within the UCL setting. This remains an open problem at our end.
> > >
> > > However, in our UCL experiments aimed at verifying the results of Theorem 3, we generated anchor and positive samples using label information as in SCL which ensures conditional independence. But unlike in SCL, the ***negative*** samples in our UCL experiments aimed to verify Theorem 3 do not use label information. Note that we also reported experimental results for UCL where the anchor and positive samples are generated without using label information (Section B5, Experiments with the SimCLR framework). In these experiments conditional independence is not guaranteed and we also do not observe neural collapse (Figure 12).
> > >
> > > **Discussion point:** All the discussion regarding the number of latent classes in UCL, in particular the authors’ claim that “The number of latent classes in UCL is an unknown intrinsic property of the dataset and is not a tunable parameter of the neural network model used to learn a mapping from data space to representation space.”
> > >
> > > **Comment:** I disagree with your claim that the number of latent classes is an intrinsic property of the dataset and is not influenced by the model. The latent classes are created by the model’s representation as clusters in the representation/feature space. The model could collapse all the samples into a single vector, thus creating one latent class, or to the other extreme, could represent each training sample differently, thus effectively assigning each training example a different latent class. The number of latent classes is a variable that has to be carefully tuned by the optimization algorithm so that it well agrees with the number of true classes or the number of distinctive feature combinations. Let me demonstrate this on an example: assume we have ten animal species and in each sample, two different animals are shown, one in foreground, one in background. Since there are no labels, it is fully up to the model whether it decides to represent the training data in 10 latent classes (the foreground animal), 45 latent classes (animal pairs), 90 latent classes (ordered animal pairs) or even a different number (like the number of training samples or, less than 10). Based on this, I ask the authors to either clarify what exactly they mean by latent classes in UCL, using, perhaps, my example from above; or correct and reformulate all the occurrences of this statement, whilst still including the related discussion I asked for in my review.
> > >
> > > *Response:* In our theoretical as well as practical implementational set-up, the latent classes in the data are completely modeled via the ***joint distribution of positive pairs***. Contrastive Learning learns representations that are similar for positive pairs and are different for negative samples.
> > >
> > > In the reviewer’s example, if the positive sampling distribution in UCL pairs images with the same foreground animal irrespective of the background animal, then the number of latent classes is 10. On the other hand if the positive sampling distribution always pairs images having the same pair of distinct animals in the foreground and background,  then the number of latent classes is 45, and so on. Of course, for a given positive sampling distribution generating the images, the underlying latent classes with respect to which the image-pairing is consistent may not be explicitly known or even obvious.
> > >
> > > In conclusion, the number of latent classes is ***not*** a tunable parameter of the model that maps the data to the representation space. It is an intrinsic property of the positive sampling distribution. One can also appreciate this by considering the unconstrained features model which would take the network (f) completely out of the picture and yet all the theoretical analysis and conclusions we presented hold.  The only tunable parameter related to observing the effect of the latent classes (e.g., neural or dimensional collapse) is the output dimension of the neural network. In practice, the specific network itself may play some role, but we believe that it is primarily in terms of its representation capacity.

---

> > > > ### Author Response · Authors · 2025-01-22
> > > > **Author response to Reviewer C1Bz (continue)**
> > > >
> > > > **Discussion point:** “Indeed, from the lower bound, the larger the number of classes, the higher the value of the lower bound.”
> > > >
> > > > **Comment:** How do you see this? Intuitively there are two opposing factors – on one hand the more clusters, the smaller the probability of the negative sample being from the same cluster as the positive pair, but on the other hand, the more clusters the smaller margin.
> > > >
> > > > *Response:* Theorem 2 for SCL and Theorem 3 for UCL provide lower bounds to the costs (see Equation (11) and Equation (30) ) and also conditions under which the lower bounds can be achieved with equality. The theorems also show that when the lower bounds are achieved, we will have neural collapse (then there is no intra-class variance of representations). The lower bounds are nondecreasing functions of the number of latent classes (because the $\psi$ function is nondecreasing).
> > > >
> > > > **Discussion point:** Including the discussion about empirical and theoretical losses.
> > > >
> > > > **Comment:** I still think these remarks should not only be in the beginning of section 4.3 (in fact I didn’t have any problem with that paragraph), but also in Section 3.1.
> > > >
> > > > *Response:* We will add these remarks in section 3.1 as suggested.
> > > >
> > > > **Discussion point:** Finding out that hard negative sampling improves collapse is not insightful…
> > > >
> > > > **Comment:** Thank you for clarifying this, I agree with you now.
> > > >
> > > > *Response:* We appreciate the reassessment by the reviewer on the utility of negative sampling.

---

> > > > > ### Comment · Reviewer_C1Bz · 2025-01-30
> > > > >
> > > > > **Discussion point:** Conditional independence in UCL.
> > > > >
> > > > > *Response:* But I find this an issue. If there is no way how to guarantee conditional independence of the positive pair, then your theorem in UCL is based on impossible assumptions. The fact that NC is not achieved without this is even more pessimistic. It suggests NC is not achieved in truly UCL regimes at all.
> > > > >
> > > > > **Discussion point:** The number of latent classes.
> > > > >
> > > > > *Response:* Unfortunately I still disagree with you. In UCL, there is no way how to guarantee sampling positive pairs from, for instance, the set of images with the same foreground animal. There simply does not exists such sampling procedure that would guarantee this. Instead, all samples in UCL are treated as a separate class. Then, it is upon the model *and* optimization mechanism to determine the number of clusters in the feature space. You can incentivize less or more clusters by tuning the optimization hyperparameters.
> > > > >
> > > > > **Discussion point:** The value of the lower bound in UCL is non-decreasing function of the number of classes.
> > > > >
> > > > > *Response:* I still don't believe this. Perhaps the $\phi$ is non-decreasing, but the issue is the probability of class clash also decreases, which should re-balance the first effect. Could you please try to numerically evaluate the expression 30 for a couple of popular choices of $\phi$? I am quite sure the lower-bound will be unimodal with a clear local minimum for some relatively small (probably one digit) number of latent classes.

---

> > > > > > ### Author Response · Authors · 2025-02-13
> > > > > > **Author response to Reviewer C1Bz**
> > > > > >
> > > > > > **Discussion point**: But I find this an issue. If there is no way how to guarantee conditional independence of the positive pair, then your theorem in UCL is based on impossible assumptions. The fact that NC is not achieved without this is even more pessimistic. It suggests NC is not achieved in truly UCL regimes at all.
> > > > > >
> > > > > > *Response to the last two sentences:* Please note that our UCL result, namely Theorem 3, shows that conditional independence is a ***sufficient*** condition for NC to be the optimal solution. We have not shown, and we do not claim, that NC cannot be achieved without conditional independence, that is, conditional independence may not be ***necessary***. So there may be cases where NC may still be achieved in UCL settings (this is an open question).
> > > > > >
> > > > > > *Response to the utility of Theorem 3:*  While the result does not provide a method to verify if conditional independence holds, the novelty of the result, in relation to existing theoretical works on UCL, is that it relates the structure of the sampling mechanism to the optimal value of the contrastive loss, albeit under the restrictive assumption of class-conditional independence of positive pairs. We believe this is an interesting property of UCL.
> > > > > > There are results in ML that are based on assumptions that are difficult or impossible to verify in practice, e.g., the IID assumption that is standard in all ML analyses.
> > > > > >
> > > > > > **Discussion point**: Unfortunately I still disagree with you. In UCL, there is no way how to guarantee sampling positive pairs from, for instance, the set of images with the same foreground animal. There simply does not exists such sampling procedure that would guarantee this. Instead, all samples in UCL are treated as a separate class. Then, it is upon the model and optimization mechanism to determine the number of clusters in the feature space. You can incentivize less or more clusters by tuning the optimization hyperparameters.
> > > > > >
> > > > > > *Response:* The reviewer had asked us to explain the number of latent classes in the specific scenarios where the class depended on the foreground, or background, or both foreground and background animals in the images.  Our previous response to these scenarios clarified this in detail under the conditions stated by the reviewer (with the sampling mechanism respecting the foreground-background constraints).
> > > > > >
> > > > > > To the reviewer’s current point regarding the non-existence of sampling mechanisms in UCL that respect these specific latent classes: Our main point is that a given sampling mechanism could respect the structure of some other latent classes and these may or may not be interpretable in a simple manner such as foreground/background animal(s). Our analysis is agnostic to the human-interpretability of the latent classes.
> > > > > >
> > > > > > *If* indeed the samples are generated according to a latent class probability distribution as explained in Section 2,  then all our results [lower bounds, NC, ETF, etc.] are independent of the neural network used for optimization *as long as the neural network model has the capacity to represent the mapping which achieves the lower bound*.
> > > > > >
> > > > > > On the other hand, of course, in practice, the training algorithm may not achieve a global optimum (this depends on the neural network model, the optimization algorithm, as well as the hyperparameters) and the local optimum to which it converges may correspond to a different number of latent classes. From our perspective, we distinguish between the true number of latent classes and the ones that are discovered by a suboptimal model/algorithm. Perhaps this is the point that the reviewer is making? If so, we are happy to revise the discussions at appropriate places.

---

> ### Author Response · Authors · 2025-02-13
> **Author response to Reviewer C1Bz (continue)**
>
> **Discussion point**: I still don't believe this. Perhaps the  is non-decreasing, but the issue is the probability of class clash also decreases, which should re-balance the first effect. Could you please try to numerically evaluate the expression 30 for a couple of popular choices of ? I am quite sure the lower-bound will be unimodal with a clear local minimum for some relatively small (probably one digit) number of latent classes.
>
> *Response:* Admittedly, we were not careful in our response to this question. We re-read all our prior responses and traced the source of potential confusion to our first response where we claimed monotonicity for both the SCL and UCL lower bounds without elaboration. We thank the reviewer for pushing us to revisit this point in our response.
>
> The lower bound for SCL (Eq. (11)):
>
> $\Psi_k(\frac{-C}{C-1}, \ldots, \frac{-C}{C-1})$
>
> is clearly monotonically increasing in C since $\Psi_k$ is argumentwise non-decreasing.
>
> Interestingly, the UCL lower bound in (Eq. (30)):
>
> $L_{UCL}^{(k)}(f)  \geq   \frac{1}{C^{k+1}} \sum_{y,y^-_{1:k} \in \Ycal} \psi_k \left(\tfrac{-C\,1(y^-_1 \neq y)}{(C-1)},\ldots,\tfrac{-C\,1(y^-_k \neq y)}{(C-1)}\right)$
>
> is also monotonic in C, but it is monotonically decreasing and not increasing in C. We agree this is not transparent from the equation. We include the proof of monotonicity for the case k = 1 in Appendix A of the revised paper. For larger values of k=1,2,3,4,5, we also provide in Appendix A two plots of the lower bound as a function of C for the InfoNCE loss and the triplet loss.
> As can be seen from the plots, the lower bound decreases with C, and for a given C, it increases with k.
>
> We added a discussion of this point in Appendix A in the revised paper as originally suggested by the reviewer (in the first round of reviews).

---

> > ### Comment · Reviewer_C1Bz · 2025-03-04
> > **good job**
> >
> > Thank you for executing this discussion point. I think your added discussion improves the paper.

---

> ### Comment · Reviewer_C1Bz · 2025-03-04
> **still not clear**
>
> Since the discussion period is approaching end, let me only follow-up on the second discussion point here, which I still find important-enough to clarify.
>
> Let me try another way of explaining my concern: based on the assumptions of theorem 3 and equation 4, in UCL in practice you basically sample all the samples including anchor, positive sample and negative samples IID from the training data. This means together with (4) that the latent classes as you see them, i.e. induced by the sampling mechanism, do not matter at all in the mechanism. On the other hand, assume that your model *represents* the training data into $C^\prime$ perfect clusters that form an ETF. Then, it is clearly visible from equation (1) that the loss would (owing to the IID sampling from the training dataset) depend on the number these clusters, but not on the latent classes and the $C$ in your lower bound would not be the $C$ that is defined as the number of latent classes (which does not have any effect on the loss whatsoever) but rather the $C^\prime$ that represents clusters induced by **model's representation.** Therefore, it is the number of clusters induced by model's representation what enters the bound in theorem 3 as the $C$ variable and not the number of latent classes.
>
> I hope this clarifies my concern which stays the same from the very beginning and is exactly described in the above paragraph.

---

> ### Author Response · Authors · 2025-03-05
> **Regarding the neural network model influencing the number of latent classes in Theorem 3**
>
> Thank you for your further clarification regarding this question and for your continued engagement with us.
>
> Firstly, Equation (4) which fully describes the joint distribution of the anchor, positive, and k negative samples does not imply that all the samples (anchor, positive, negatives) are IID. Equation (4) implies that all the negatives are independent of each other and the anchor and positive samples and furthermore the anchor and positive samples are conditionally IID given their latent class label (they are not unconditionally IID).
>
> The reviewer’s comment that “in practice you basically sample all the samples … IID from the training data” is not correct. In practice, i.e. for the simulations, we do not sample **all** samples in an IID manner from training data in any of our experiments. We have described this in detail in the first 5 paragraphs of Section 5 and we summarize the key points here. We sample **negative** samples uniformly from among all the training samples (and their augmentations, if any). But for the positive pair, we report results for three different sampling mechanisms. In one, for the purpose of empirical verification of our theoretical results, we use latent class information to generate samples. This satisfies the assumptions on the sampling mechanism described in Theorem 3.  In particular, the positive samples are conditionally IID given their latent class label, but they are not IID (unconditionally). The second sampling mechanism for generating a pair of positive samples that we used is independent Gausian noise augmentations of a reference sample. If the noise level is not high, the two positive samples are likely to have the same latent class label as the reference and they would be conditionally IID given the label (but not unconditionally IID as implied by the reviewer). The third sampling mechanism for positive pairs that we reported results for in Section 5.8 and Appendix B.5 is the SimCLR framework for which the assumptions of Theorem 3 cannot be guaranteed and we have discussed this very point in the paper draft. So the reviewer’s comment that “the latent classes…do not matter at all in the mechanism” is not correct..
>
> Next, the conclusions drawn by the reviewer for the hypothetical scenario in which the model already maps training data into C’ clusters with representations that form an ETF are incorrect since the positive pairs are not independent in all 3 sampling mechanisms we have reported results for. If the **positive sampling mechanism is compatible with C latent classes**, but the model is initially mapping training data into C’ “clusters” that form an ETF, **but the C’ clusters are not compatible with the C latent classes** (meaning that samples from the same latent class appear in different clusters), then during training, the model will attempt to map training samples to an ETF which is **compatible with C latent classes**. Of course, for this to succeed, (i) the model must have the capacity to represent the mapping corresponding to an ETF compatible with C latent classes and (ii) it must be able to reach the global minimum. We had explained this in our previous response.

---

> > ### Comment · Reviewer_C1Bz · 2025-03-05
> > **"in practice" meant something else**
> >
> > Thank you for your prompt answer. Unfortunately, I did not choose the right word. By "in practice" I did not mean in "real practice" when you actually train and evaluate your models. What I meant is "in practice with regard to assumptions of Theorem 3", i.e. having the empirical loss. Based on assumptions of theorem 3, all samples are IID and "in practice" sampled from fixed training data. Therefore, in this case my previous analysis holds and it is indeed the C' parameter and not C that matters for the loss.
> >
> > I hope this clarifies the discussion.

---

> > > ### Author Response · Authors · 2025-03-05
> > > **Independence relationships between anchor, positive and k negatives**
> > >
> > > It is unfortunate the reviewer said “in practice” when they really meant “in theory” which is not how the phrase is usually interpreted. Even now, it is unclear whether the reviewer’s question is really about theory (Theorem 3) or practice (experiments). This is muddling the discussion. Even so, we had responded to both aspects (theory and practice) in our previous response. It appears the reviewer’s understanding of the independence relationships between various samples associated with the assumptions of Theorem 3 is erroneous and therefore the ensuing analysis by the reviewer falls apart completely.
> > >
> > > In the very first paragraph of our previous response, we had already discussed the assumptions of Theorem 3 and had explicitly pointed out that **the samples are not IID.** For completeness, we explain the key points here, once again, but in more detail. The very first line of Theorem 3 states that the joint distribution of the anchor sample, the positive sample, the k negative samples, and their (k+1) latent labels (the anchor and positive samples share a common latent label) is as per Equation (4). The joint  distribution given by Equation (4) does not imply that the (k+2) samples (anchor, positive, k negatives) are IID, even with the additional assumptions in Theorem 3. The anchor-positive-pair is independent of the k negatives. The k negatives are IID. But **the anchor sample and the positive sample are dependent through their common shared latent class label.** Per additional assumption (e) of Theorem 3, **the anchor and positive samples are conditionally IID given their shared label, but conditional IID does not make them (unconditionally) IID.** In fact, their dependence through their shared label is crucial.
> > >
> > > We hope this explanation helps clear up any confusion the reviewer may have about the assumptions of Theorem 3 and associated independence relationships between the samples.

---

> > > > ### Comment · Reviewer_C1Bz · 2025-03-05
> > > > **you are right**
> > > >
> > > > Thank you for your detailed explanations. I admit I made a mistake and I apologize for a needlessly prolonged discussion. You are right that it is indeed the number of latent classes and not the number of clusters that the model represents the samples with which matters in theorem 3. The source of my fallacy was that I have always subconsciously thought of the anchor and positive sample in UCL as identical (or different augmentation of the same sample that are implicitly guaranteed to be represented the same way by the optimal model). It is indeed true that this is *not* in accordance with the assumptions of theorem 3 where the positive sample is re-sampled from the pool (conditionally on the same latent label as the anchor). This indeed makes the number of latent classes the key quantity. While I don't want to defend myself too much, it is still hard to imagine that the latent classes in UCL in practice, given a fixed training set, would be anything else *but* the individual samples (singletons) or their augmentations. In particular, if the latent classes were anything else, one could infer them with high probability by sampling enough times and observing the anchor-positive sample pairs and forming clusters of these pairings, which would turn it into SCL.
> > > >
> > > > Anyway, given that this point is resolved, I can proceed with the official recommendation, which will be accept.

---

### Review · Reviewer_BYJa · 2024-12-15

**Summary Of Contributions:**

This paper proposes a lower bound analysis for a family of contrastive learning loss function, for which these lower bounds are attainable by the Neural-Collaspe solution. Their analysis relies on the assumption that each data sample is assigned to one latent class label. An `equal inner-product class means` property (13) is proposed and the anaylsis studies its connection to the global optimum of contrastive losses, for example claiming (13) as a necessary condition of optimality for strictly convex loss function $\psi$. Experimental results are provided as an attempt to observe the Neural-Collaspe phenomenon under supervised/unsupervised losses with different hard-negative sampling strategies.

**Audience:**

Yes

**Claims And Evidence:**

No

**Requested Changes:**

### Typos
- $f(x)_1^-$ should be $f(x_1^-)$ above equation (14).
- Typo in Section 5: `all samples that share the same label ...`. "all" should be capitalized.

### Presentation
- The losses in Theorem 1 are only implicitly defined through the definition of $L_{CL}^{(k)}(f)$ in Definition 1. I suggest the authors to generalize the notation after Definition 1 and define $L_{HUCL}, L_{UCL}, L_{HSCL}, L_{SCL}$ properly, for example denote $L_{HUCL}^{(k)}(f) = L_{CL}^{(k)}(f, p_{HUCL})$.
- The result of Theorem 1 is not discussed. What are the implications from to the proved inequalities?
- The experiment results can be presented more clearly. I suggest the authors to pick out some representative figures to be shown nearby in the main text, rather than showing numerous figures in a single page that is far from the discussion of those results.



### References

- I suggest the authors to include a reference to a recent work on hard-negative sampling [2]. Since it is mentioned in Section 6 that the training dynamic of hard-negative sampling is an open question, we can see [2] explains the optimization trajectory of hard-negative samples as obtaining unbiased gradient estimation of the global contrastive loss. On the other hand, it would also be interesting to see if the minimizer of the global contrastive loss also exhibits Neural-Collaspe by a similar experiment setup in Section 5.

[1] Alec Radford, Jong Wook Kim, Chris Hallacy, Aditya Ramesh, Gabriel Goh, Sandhini Agarwal, Girish Sastry, Amanda Askell, Pamela Mishkin, Jack Clark, Gretchen Krueger and Ilya Sutskever. Learning Transferable Visual Models From Natural Language Supervision. ICML, 2021.

[2] Chung-Yiu Yau, Hoi-To Wai, Parameswaran Raman, Soumajyoti Sarkar and Mingyi Hong. EMC$^2$: Efficient MCMC Negative Sampling for Contrastive Learning with Global Convergence. ICML, 2024.

**Strengths And Weaknesses:**

- The assumption that each data sample belongs to one class is very restrictive for application. For instance, real-world data samples often follow a hierarchical class attributes and state-of-the-art contrastive learning models such as CLIP [1] are often considered as learning such hierarachical structure.

- In SimCLR, the InfoNCE loss is adopted with $\psi_k(t_{1:k}) = \log(\alpha + \sum_{i=1}^k e^{t_i})$ and $\alpha = 0$. This could be mentioned in Section 5.8 to explain why SimCLR does not exhitbit Neural-Collaspe, as it determines the strict convexity of $\psi$ which is a condition considered in Theorem 3.

- In page 9 `(iv) Proof that additional condition (13) implies zero within-class variance`, it is not clear why (22) $\Leftrightarrow f(x) = \mu_y$ given $y$. For instance, it is possible that for $\bar{x},\tilde{x} \sim s(x | y)$, (22) $\Rightarrow \| f(\bar{x}) \| = \| f(\tilde{x}) \| = 1$ but $f(\bar{x}) \neq f(\tilde{x})$.

---

> ### Author Response · Authors · 2025-01-08
> **Author response to Reviewer BYJa**
>
> **Comment**: The assumption that each data sample belongs to one class is very restrictive for application. For instance, real-world data samples often follow a hierarchical class attributes and state-of-the-art contrastive learning models such as CLIP [1] are often considered as learning such hierarachical structure.
>
> *Response*: This theoretical work focuses on the single-label problem common to a multitude of datasets such as CIFAR 10 and CIFAR 100. Multi-label, hierarchical-label, and more generally, structure-learning problems are certainly important in suitable real-world settings, but they are not the focus of this work. To the best of our knowledge, all theoretical studies of neural collapse within the contrastive learning setting are confined to the single-label scenario. We would like to clarify that our theoretical model for data considers the agnostic (stochastic) case where the label is not a deterministic function of the data. But specific theoretical results related to neural collapse in this work only hold for deterministic labels.
>
> **Comment**: In SimCLR, the InfoNCE loss is adopted with (latex of loss function). This could be mentioned in Section 5.8 to explain why SimCLR does not exhitbit Neural-Collaspe, as it determines the strict convexity of which is a condition considered in Theorem 3.
>
> *Response*: For our “SimCLR” experiments we use the loss indicated in Equation (42), Section B.5 and the choice of alpha there is equal to 1. Therefore, the loss function is strictly convex. The main reason we don’t see NC in our SimCLR setting (as explained in section 5.8) is because the augmentation and negative sampling set-ups do not use hard label information (in contrast to Theorem 2 which uses hard label information for negative sampling) and nor does it satisfy the conditional independence assumption (used in Theorem 3). The loss in the original SimCLR work does indeed have alpha equal to zero. But the main idea in the original SimCLR work was to use random independent augmentations of the anchor to create positive pairs. In our SimCLR setting this is what is done but with a slightly different choice of the loss. Note that due to this augmentation mechanism, the conditional independence of the positive pairs is lost, which is what we wanted to highlight for the SimCLR setting.
>
> **Comment**: In page 9 (iv) Proof that additional condition (13) implies zero within-class variance, it is not clear why (22) given … For instance, it is possible that …
>
> *Response*: Here, we applied Jensen’s inequality to the variable f(x), not |f(x)|. Therefore, to achieve equality, it must be the case that f(x) has the same value (with probability one) for any x in the same class which is also the class mean value.
>
> **Comment**: Typos: above equation (14). Typo in Section 5: all samples that share the same label .... "all" should be capitalized.
>
> *Response*: Thank you very much for pointing these typos out. We have corrected them in the revised version.
>
> **Comment**: The losses in Theorem 1 are only implicitly defined through the definition of $L_{CL}^{(k)}(f)$ in Definition 1. I suggest the authors to generalize the notation after Definition 1 and define $L_{HUCL}, L_{UCL}, L_{HSCL}, L_{SCL}$ properly, for example denote  $L_{HUCL}^{(k)}(f)=L_{CL}^{(k)}(f, p_{HUCL})$
>
> *Response*: In addition to ease of reading and simplicity, our decision to intentionally suppress the underlying probability distribution in the loss function notation was based on two considerations. Firstly, the only *primary* variable of optimization in the loss function during training is the representation map $f$. The underlying probability distribution depends on $f$ only in the hard-negative cases. Secondly, the underlying probability distribution is at once apparent from the subscript of the loss function itself, e.g., $L_{HUCL}^{(k)}(f)$ has $p_{HUCL}$ as the underlying distribution. Thus it seems unnecessary to explicitly include $p_{HUCL}$ as an additional argument. If the reviewer still feels strongly about this point, we are happy to expand the notation. But we would prefer to retain the simpler notation.
>
> **Comment**: The result of Theorem 1 is not discussed. What are the implications from to the proved inequalities?
>
> *Response*: The implications of Theorem 1 are discussed later in Theorem 2 where we showed that both SCL and HSCL achieve NC. We added this point in the revised draft right after the proof of Theorem 1.

---

> > ### Comment · Reviewer_BYJa · 2025-02-14
> > **Discussion on Proof (iv) of Theorem 2**
> >
> > I would like to re-emphasize our discussion on the Proof (iv) of Theorem 2 again. I agree that (22) is correct according to Jensen's inequality. However, the tight bound shown in (22), i.e., $\mathbb{E}_{x \sim s(x | y)} [\\| f(x)\\|^2 ] = 1$, only ensures that the mean of feature squared norms equals to 1. It is not clear how does it implies each $f(x)$ has the same value $\mu_y$ for each $x$ from the same class, i.e., within class neural-collapse.

---

> ### Author Response · Authors · 2025-01-08
> **Author response to Reviewer BYJa (continue)**
>
> **Comment**: The experiment results can be presented more clearly. I suggest the authors to pick out some representative figures to be shown nearby in the main text, rather than showing numerous figures in a single page that is far from the discussion of those results.
>
> *Response*: We had tried the arrangement suggested by the reviewer before but decided to keep the current format since keeping all figures from one experiment together allows us to keep a consistent order throughout different experiments and clearly bring out all the points in a concise and transparent manner. We will still try to see if this can be done in a better way without sacrificing the quality of the figures and the discussions and revise the draft accordingly.
>
> **Comment**: I suggest the authors to include a reference to a recent work on hard-negative sampling [2]. Since it is mentioned in Section 6 that the training dynamic of hard-negative sampling is an open question, we can see [2] explains the optimization trajectory of hard-negative samples as obtaining unbiased gradient estimation of the global contrastive loss. On the other hand, it would also be interesting to see if the minimizer of the global contrastive loss also exhibits Neural-Collaspe by a similar experiment setup in Section 5.
>
> [1] Alec Radford, Jong Wook Kim, Chris Hallacy, Aditya Ramesh, Gabriel Goh, Sandhini Agarwal, Girish Sastry, Amanda Askell, Pamela Mishkin, Jack Clark, Gretchen Krueger and Ilya Sutskever. Learning Transferable Visual Models From Natural Language Supervision. ICML, 2021.
>
> [2] Chung-Yiu Yau, Hoi-To Wai, Parameswaran Raman, Soumajyoti Sarkar and Mingyi Hong. EMC Efficient MCMC Negative Sampling for Contrastive Learning with Global Convergence. ICML, 2024.
>
> *Response*: We appreciate these references. At first read the second reference seems to be regarding making the sampling of negative samples more efficient from a large dataset, which is relevant to large scale implementation of CL methods. It does not seem to be related to the dynamics of the iterative methods used for minimizing the CL loss itself under hard negative sampling. If the reviewer can please provide further insights on how this work is related to the open problem posed in our paper we will be happy to cite it appropriately in the revision.
>
> *All the changes in our revision version are highlighted in blue text.*
>
> *Thank you very much for your feedback!*
>
> *Sincerely,*
>
> *Authors,*

---

> ### Author Response · Authors · 2025-02-19
> **Author response to Reviewer BYJa**
>
> **Discussion on Proof (iv) of Theorem 2**
>
> *Comment:* I would like to re-emphasize our discussion on the Proof (iv) of Theorem 2 again. I agree that (22) is correct according to Jensen's inequality. However, the tight bound shown in (22), i.e., $E_{x \sim s(x|y)}[||f(x)||^2]=1$, only ensures that the mean of feature squared norms equals to 1. It is not clear how does it implies each f(x) has the same value μy for each x from the same class, i.e., within class neural-collapse.
>
> *Response*: Thank you. We recall the following well-known result concerning conditions for equality in Jensen’s inequality for random vectors and a strictly convex function, e.g., see these references:
>
> *FERGUSON, T. S. (1967). Mathematical Statistics. Academic Press, New York.*
>
> *LOEVE, M. (1963). Probability Theory, 3rd ed. Van Nostrand, Princeton, NJ*
>
> *MICHAEL D. PERLMAN, Jensen’s Inequality for a Convex Vector-Valued Function on an Infinite-Dimensional Space, JOURNAL OF MULTIVARIATE ANALYSIS 4, 52-65 (1974)*
>
> If $g(z)$ is a strictly convex function of $z$ and we have equality in Jensen’s inequality for random vectors, i.e., $E[g(Z)] = g(E[Z])$, then $Z = constant$ with probability one according to the underlying probability measure.
>
> We used this well-known result in our proof as follows:
>
> $Z = f(X)$, $g(Z) = ||Z||^2$ (strictly convex function of $Z$), and the probability measure is the conditional probability distribution of $Z$ given $Y =  y$ whose mean is equal to $\mu_y$.
>
> Equation (22) in our paper draft shows that we have equality in Jensen’s inequality with, $Z$, $g(Z)$, and the probability measure as defined above. Therefore, $Z|(Y=y) = constant$ with probability one. This constant is the expected value of $Z |(Y=y)$, i.e., $\mu_y$.
>
> We hope the above explanation resolves the reviewer’s confusion.

---

### Review · Reviewer_PVUi · 2025-02-14

**Summary Of Contributions:**

The paper provides a theoretical and empirical investigation into the impact of hard-negative sampling in contrastive learning (CL). The authors establish that supervised and unsupervised contrastive learning losses are minimized by representations that exhibit Neural-Collapse (NC), where class means form an Equiangular Tight Frame (ETF). The paper introduces Hard-Supervised Contrastive Learning (HSCL) and Hard-Unsupervised Contrastive Learning (HUCL), proving that their losses are lower-bounded by the corresponding standard CL losses. Through rigorous mathematical proofs and experiments on synthetic and real datasets, the paper demonstrates that feature normalization and hard-negative sampling mitigate Dimensional-Collapse (DC), a phenomenon where learned representations degenerate to a low-rank subspace.

**Audience:**

Yes

**Broader Impact Concerns:**

No Broader Impact Concerns

**Claims And Evidence:**

Yes

**Requested Changes:**

1. While the paper is generally well-written, some definition(especially for the hard-negative sampling part), statements and notations could be streamlined for better readability.
2. Equal inner-product class assumption directly enforces Neural Collapse on class means. While it simplifies theoretical analysis, it bypasses the question of whether this collapse naturally occurs during training in contrastive learning settings. A more relaxed assumption would allow class means to emerge as an ETF rather than explicitly requiring it.

**Strengths And Weaknesses:**

**Strengths**：
1. Theoretical Contributions：The paper provides formal proofs that hard-negative contrastive learning converges to Neural-Collapse, extending existing results beyond the class-conditional independence assumption. Moreover, It establishes that HSCL and HUCL losses are greater than or equal to SCL and UCL losses, respectively. Finally, The results apply to a broad class of loss functions, including InfoNCE and triplet loss.
2. Practical Relevance: The study provides insights into choosing the hardness level to achieve better feature separation.The results highlight the necessity of feature normalization and hard-negative sampling to avoid Dimensional-Collapse.

**Weakness**:
1. Clarity: The section 3.2 "Hard-negative sampling" is unclear for the reader to understand. For the harding function (definition 2), while an exponential tilting hardening function is provided as example, however, it is unclear how it is used in the practice. Moreover, HSCL and HUCL is hard to understand. It would be better to provide some examples for intuition.
2. Strong assumption: For theorem 2, the authors assume that the function f satisfy the equal inner-product class, which actually directly assume the simple ETF structure of the class-mean centers, which might  holds some some special case of losses function, such as InfoNCE based on previous literature, however for this assumption holds for other losses such as triple loss is unknow.

**Questions**:
1. Hardness Selection Heuristics: The study empirically explores various hardness levels but does not provide a principled method for selecting the optimal hardness level in practical settings. Is there a principled way to determine the optimal hardness level for hard-negative sampling in contrastive learning? Could an adaptive hardness function be developed?
2. Role of Feature Normalization: Why is feature normalization so crucial for avoiding Dimensional-Collapse? Could there be other approaches to address Dimensional-Collapse without normalization?
3. How does the choice of batch size affect the convergence to Neural-Collapse? Would using larger batches inherently lead to more stable results?

---

> ### Author Response · Authors · 2025-02-24
> **Author response to Reviewer PVUi**
>
> **Discussion point:**
>
> "The section 3.2 "Hard-negative sampling" is unclear for the reader to understand. For the harding function (definition 2), while an exponential tilting hardening function is provided as example, however, it is unclear how it is used in the practice. Moreover, HSCL and HUCL is hard to understand. It would be better to provide some examples for intuition."
>
> and
>
> "While the paper is generally well-written, some definition(especially for the hard-negative sampling part), statements and notations could be streamlined for better readability."
>
>
> **Response:** Definitions 3 and 4, respectively, provide a precise technical explanation of hard-negative sampling for SCL and UCL. The paragraphs just below those definitions provide a more intuitive explanation of hardening, namely that negative samples that are more aligned with the anchor in the representation space, i.e., $f (x)^⊤ f (x^−)$ is large, are sampled relatively more often in hard negative sampling compared to regular sampling. Appendix C.4 provides another example of a hardening function and associated experimental results.
>
> If the confusion is about how hard negative sampling is implemented in practice, within each minibatch, when we sample negatives, we use the hardening function to increase the probability of selecting negative samples that have a higher value of $f (x)^⊤ f (x^−)$ by a multiplicative factor equal to the hardening function value at $f (x)^⊤ f (x^−)$. We can provide more details in Section 5 or the Appendix in the revised draft.
>
> **Discussion point:**
>
> "Strong assumption: For theorem 2, the authors assume that the function f satisfy the equal inner-product class, which actually directly assume the simple ETF structure of the class-mean centers, which might holds some some special case of losses function, such as InfoNCE based on previous literature, however for this assumption holds for other losses such as triple loss is unknow."
>
> and
>
> "Equal inner-product class assumption directly enforces Neural Collapse on class means. While it simplifies theoretical analysis, it bypasses the question of whether this collapse naturally occurs during training in contrastive learning settings. A more relaxed assumption would allow class means to emerge as an ETF rather than explicitly requiring it."
>
> **Response:** We believe this is a misunderstanding of Theorem 2:  The equal inner products condition (13), i.e., the existence of an $f$ satisfying (13), is a condition on the family of representation maps $f$ and the dataset and not on the loss function. The results of Theorem 2 hold for *any loss function that is convex and argument-wise non-decreasing* (Definition 1), which includes the triplet loss, and is not confined to just the InfoNCE and similar loss functions. We would also like to draw attention to the “Empirical SCL” loss para of Section 4.3. As we discuss there, condition (13) can be guaranteed provided the family of representation maps (defined by the neural network) has sufficiently high capacity. This holds irrespective of the loss function.
>
> Theorem 2 has a necessary part and a sufficient part. The sufficient part states that **if** there is an $f$ such that the equal angles condition (13) holds (and note that this is a property of the family of $f$’s and the dataset and not the loss function), then the lower bound (11) will be achieved for any general $\Psi$ (convex and argument-wise non-decreasing) and we will have other properties (i)--(vi) as well. The necessary part of Theorem 2 (the last sentence) states that the only way we can get equality in the lower bound is to have the equal angles condition (13). So Theorem 2 is **not limited** to one particular loss function like InfoNCE, but rather any $\Psi$ satisfying Definition 1 (including triplet loss).

---

> > ### Author Response · Authors · 2025-02-24
> > **Author response to Reviewer PVUi (continue)**
> >
> > **Discussion point:** "Hardness Selection Heuristics: The study empirically explores various hardness levels but does not provide a principled method for selecting the optimal hardness level in practical settings. Is there a principled way to determine the optimal hardness level for hard-negative sampling in contrastive learning? Could an adaptive hardness function be developed?"
> >
> >
> > **Response:** Even though the advantages of using hard-negative sampling have been demonstrated in practice, to the best of our knowledge, finding an optimal hardness level in a principled way is still an open problem as we mentioned in Section 6 titled Conclusion and Open Questions. It may be possible to adaptively change the hardening function, but that setting may require a different set-up that allows for some feedback on the performance of the current hardness level to enable updates on the hardening function. Indeed, this is an interesting direction for future work, but is beyond the scope of the present paper.
> >
> > **Discussion point:** "Role of Feature Normalization: Why is feature normalization so crucial for avoiding Dimensional-Collapse? Could there be other approaches to address Dimensional-Collapse without normalization?"
> >
> > **Response:** As we discussed in the paragraph following Definition 7, one way to avoid Dimensional-Collapse is to achieve Neural Collapse. As shown in our Theorem 2 and Theorem 3 (and Remarks 1 and 2 following Theorem 2), one condition for Neural Collapse is unit-ball or unit-sphere normalization of feature vectors. This may partially explain why feature normalization helps prevent or mitigate dimensional collapse. On the other hand, it is not necessary to achieve Neural Collapse to avoid Dimensional-Collapse (this is also discussed in the paragraph following Definition 7). We leave it to future work to study more general conditions that help avoid Dimensional-Collapse.
> >
> > **Discussion point:** "How does the choice of batch size affect the convergence to Neural-Collapse? Would using larger batches inherently lead to more stable results?"
> >
> > **Response:** We showed that Neural Collapse can occur in Contrastive Learning with any value of batch size if certain conditions are satisfied (see Section 4.3). Thus, in theory, Neural Collapse could be observed regardless of the value of the batch size. In practice, however, it may be harder to achieve Neural Collapse with a very small or a very large batch size as we explain below.
> >
> > Small batch size effects: in our simulations, we observed that Neural Collapse is consistently achieved when the batch size is 64 or larger (up to 512). However, when the batch size is reduced to 32, Neural Collapse is no longer observed. One potential reason for this is the presence of 100 distinct classes in our dataset. With a batch size of 32, the anchor can be compared with samples only from very limited classes, reducing the diversity of negative pairs. This suggests that using larger batch sizes can contribute to more stable and robust results in the context of high class-cardinality.
> >
> > Large batch size effects: When the batch size is too large, the neural network may require a greater representation capacity to map a larger number of samples from the same class into the same point (class-collapse). This could be solved by increasing the capacity of the neural network by expanding its size. In our experimental results, however, we only used standard architectures and observed Neural Collapse for all batch sizes ranging from 64 to 512.

---

### Decision · Action_Editor_fuLF · 2025-04-11

**Recommendation:** Accept as is

**Comment:**

This long submission received extensive reviewer feedback regarding its presentation, experiments, and theoretical contributions—including its improvements over prior work and claims about the number of latent classes. The authors addressed these comments through additional discussions and experiments.  With thorough back-and-forth discussions, all comments have been satisfactorily addressed, and all three reviewers now recommend acceptance.

**Audience:**

This work will be of interest to researchers focused on contrastive learning, representation learning, and the theoretical foundations of deep learning.

**Claims And Evidence:**

The paper investigates contrastive learning with normalized features across a range of settings, including both supervised and unsupervised learning, as well as hard negative sampling and standard negative sampling. The authors analyze the geometry of the optimization landscape and show that the global minimum exhibits the neural collapse phenomenon in both supervised and unsupervised settings, thereby improving upon existing work that relies on the class-conditional independence assumption. Experiments on both synthetic and real datasets are provided to support the theoretical findings. Additionally, the paper demonstrates that feature normalization and hard negative sampling help mitigate dimensional collapse—a phenomenon where learned representations degenerate into a low-rank subspace.